# Modern Carbon–Based Materials for Adsorptive Removal of Organic and Inorganic Pollutants from Water and Wastewater

**DOI:** 10.3390/molecules26216628

**Published:** 2021-11-01

**Authors:** Vera I. Isaeva, Marina D. Vedenyapina, Alexandra Yu. Kurmysheva, Dirk Weichgrebe, Rahul Ramesh Nair, Ngoc Phuong Thanh Nguyen, Leonid M. Kustov

**Affiliations:** 1N. D. Zelinsky Institute of Organic Chemistry, Russian Academy of Sciences, Leninsky Prospect 47, 119991 Moscow, Russia; ankrepta@gmail.com; 2Institute for Sanitary Engineering and Waste Management, Leibniz University Hannover, Welfengarten 1, D-30167 Hannover, Germany; weichgrebe@isah.uni-hannover.de (D.W.); nair@isah.uni-hannover.de (R.R.N.); thanh.nguyen@isah.uni-hannover.de (N.P.T.N.); 3Chemistry Department, Moscow State University, Leninskie Gory 1, Bldg. 3, 119992 Moscow, Russia

**Keywords:** water remediation, pollutant removal, adsorption mechanism, carbon-based nanoadsorbents, carbon nanotubes, graphene oxide, MOF-carbon composites, MOF-derived carbons, bioadsorbents, biochar

## Abstract

Currently, a serious threat for living organisms and human life in particular, is water contamination with persistent organic and inorganic pollutants. To date, several techniques have been adopted to remove/treat organics and toxic contaminants. Adsorption is one of the most effective and economical methods for this purpose. Generally, porous materials are considered as appropriate adsorbents for water purification. Conventional adsorbents such as activated carbons have a limited possibility of surface modification (texture and functionality), and their adsorption capacity is difficult to control. Therefore, despite the significant progress achieved in the development of the systems for water remediation, there is still a need for novel adsorptive materials with tunable functional characteristics. This review addresses the new trends in the development of new adsorbent materials. Herein, modern carbon-based materials, such as graphene, oxidized carbon, carbon nanotubes, biomass-derived carbonaceous matrices—biochars as well as their composites with metal-organic frameworks (MOFs) and MOF-derived highly-ordered carbons are considered as advanced adsorbents for removal of hazardous organics from drinking water, process water, and leachate. The review is focused on the preparation and modification of these next-generation carbon-based adsorbents and analysis of their adsorption performance including possible adsorption mechanisms. Simultaneously, some weak points of modern carbon-based adsorbents are analyzed as well as the routes to conquer them. For instance, for removal of large quantities of pollutants, the combination of adsorption and other methods, like sedimentation may be recommended. A number of efficient strategies for further enhancing the adsorption performance of the carbon-based adsorbents, in particular, integrating approaches and further rational functionalization, including composing these adsorbents (of two or even three types) can be recommended. The cost reduction and efficient regeneration must also be in the focus of future research endeavors. The targeted optimization of the discussed carbon-based adsorbents associated with detailed studies of the adsorption process, especially, for multicomponent adsorbate solution, will pave a bright avenue for efficient water remediation.

## 1. Introduction

Pollutants from various sources such as industrial wastewater and burning of fossil fuels are one of the current environmental concerns. Water contamination by inorganic pollutants such as heavy metal ions and organic pollutants due to anthropogenic or geological reasons [1], presents a serious threat for all living organisms. As to the anthropogenic factors, the release of toxic pollutants into water is the result of increasing agricultural, domestic, industrial, and even social activities.

The common pollutants released into the water are pharmaceuticals, personal care products (PPCPs), artificial sweeteners (ASs) [2], dyes [3], pesticides/insecticides/fungicides/herbicides [4], organic aromatic compounds, including polycyclic aromatic hydrocarbons and polychlorinated aromatic compounds [5,6], detergents, organoarsenic compounds [7] and phthalates [8] as well as heavy metal ions (Cr(VI), Pb(II), Hg(II), etc.). The type of pollutants depends mainly on the kind of human activity. In the last decades, a rise in consumption of pharmaceuticals and PPCPs on a global scale resulted in their release into environment [9,10].

The main sources of the infiltration of organic and inorganic pollutants into the natural ecosystems are wastewater from agriculture plants (non-point sources), city-wide treatment facilities, sanitary, cosmetics, and pharmaceutical industries (point sources). Additionally, the groundwater (water accumulated underground in the cracks and voids in soils, sand, and rocks) contamination may be caused by contacts of rivers and/or sewage networks as well as leaching of the agricultural fields [10].

The diverse organic and inorganic compounds and species present in water resources are very harmful for ecological systems and aquatic animals [11]. Most of the organic pollutants are hydrophobic. Therefore, they accumulate in water and then penetrate the tissues of various aquatic organisms, as well as humans [12]. So, even trace concentrations of toxic pollutants in water may lead to serious health problems. For instance, most organic pollutants have a carcinogenic nature or are regarded as endocrine disruptors due to the tendency to interfere with the function of the natural hormones [13]. Internationally, it has been agreed to act against persistent organic pollutants (POPs) in particular [14,15].

Therefore, there is a serious and urgent need in the search and design of efficient techniques for removing these harmful materials from water. To date, a wide range of treatment strategies has been investigated for the environmental remediation in water by removal/degradation of organic toxic contaminants [16]. They include advanced oxidation processes, biological oxidation, photodegradation, biodegradation, and chlorination [17]. Conventional methods, such as membrane separation, coagulation, and flocculation, reagent purification, etc. demonstrate certain drawbacks. They are time-consuming, laborious, expensive, and even produce undesired (more harmful) by-products. Moreover, POPs are highly resistant to naturally occurring processes of biodegradation and photolysis [18].

For instance, advanced oxidation processes (AOPs) are mostly energy-intensive, and moreover, some undesired by-products are generated in the case of ozonation [19]. In turn, the coagulation method for removal of non-settleable particles is based on the formation of suspended solid particles with coagulation chemicals such as aluminum and iron coagulants, for instance, ferric and aluminum sulfates [20]. These particles gather to form larger particles in the flocculation process using polyaluminum chloride, biopolymeric pectin, polyacrylamide, etc., [21]. Then, these larger particles are removed through the sedimentation process.

A drawback of this strategy is incomplete removal of the pollutants, as well as the formation of secondary pollution in the form of sludges. Other physical techniques, such as filtration, and reverse osmosis, are widely used [22]. However, reverse osmosis, for example, requires periodic membrane recovery and purification and is quite energy-intensive [23].

To date, the use of bioremediation with application of naturally occurring microorganisms, such as algae, bacteria, and fungi deposited on hydrogels, to degrade the organic pollutants is rather typical [24]. However, some of these contaminants are resistant to biodegradation. A promising way for effective water cleaning is a combination of different removal techniques, for instance, filtration with membranes associated with coagulation/flocculation [21].

To overcome the limitations of different water remediation methods, adsorption is considered as an alternative strategy. Adsorption is one of the most easy-handled, energy-saving, and effective methods for removing organic and inorganic pollutants from aquatic systems [24]. This process involves mild operating conditions by implementing mostly at room temperature, and robust cost-energy efficiency. Moreover, adsorption has been proposed to solve the challenging task of incomplete removal of pollutants during wastewater processing [25].

Efficient adsorptive removal of hazardous materials relies on different parameters of the process depending on the adsorption design, batch, or column. For column adsorption, these are the inlet ion concentration, flow rate, bed height, etc. For the batch adsorption, the key parameters are the contact time, pH, temperature, and adsorbent dose. Note that majority of the research papers discussed in this review deal with model adsorption processes that require small quantities of the adsorbents under batch conditions.

Other important factors contributing to the efficiency of the adsorption process include rapid uptake and easy regeneration of the used adsorbents [26]. So, a corner-stone of the adsorption process is an appropriate choice of the adsorbent material. The important properties of the adsorbents of choice are high porosity, large specific surface area, chemical stability, availability, processability and regenerability [27]. Among these characteristics, the pore geometry, favorable pore structure and surface chemistry are key factors affecting the adsorption characteristics of adsorbents. Adsorbents with a high surface area and large pores (mesopores/macropores) allow the adsorbates to diffuse without hindrances inside pores [28]. In this way, several possible interactions assisting the high adsorption efficiency may be involved. Other important characteristics are the reactivity and strong adsorption affinity to aqueous adsorptive removal of pollutants [29].

In the adsorption process, pollutant molecules are attracted from the solution volume onto the surfaces and then in the active pores of the adsorbent materials via diffusion [30]. Usually, the adsorption mechanism involves intermolecular forces of attraction, such as chemisorption, e.g., chemical bonding, acid-base interactions, coordination interactions, and physisorption, e.g., electrostatic interactions, hydrogen bonding, van der Waals forces, π-π interactions and diffusion [31].

Currently, various materials are used as adsorbents for the removal of organic and inorganic pollutants [32], such as AC [33,34], polymeric matrices [35], agro-industrial wastes [36] which are regarded as low-cost adsorbents and biosorbents, i.e., biomass and chitosan [37,38] as well as inorganic matrices including alumina [39], zeolites [40], soil/clays [41,42], mesoporous silicas [43], and hybrid nanoporous solids, i.e., MOFs [8].

Among various types of adsorbent materials, carbon-based materials including activated carbon (AC) [44] take a particular place. They are the most popular adsorbents, especially commercial and widely produced from biomass. The used raw material has a significant impact on the final product properties, including pore size distribution and volume, hardness, and purity [45]. The popularity of the activated carbon originates from its physical characteristics, such as a high surface area, up to 1100 m^2^/g, pore volumes up to 0.40 cm^3^/g, bulk density, low ash content and low conductivity [1]. AC has a micropore-dominant structure with a high external and internal or microporous surface area (the cumulative surface of the micropore walls) [46]. Additionally, surface functional groups provide binding sites for the efficient adsorption of pollutants in water [47]. AC exhibits excellent adsorption performance towards various chemicals ranging from hydrophobic organic contaminants to heavy metals [48]. However, AC has also a number of disadvantages such as irregular porous structure and tangled porous networks, as well as poor selectivity and ineffectiveness in the removal of some emerging organic pollutants from water due to unfavorable kinetics [49] and lack of complete removal at low concentrations of pollutants [50,51]. Moreover, the regeneration and recovery of the adsorbent are often challenging [52].

In general, AC [53] along with inorganic materials [54] have a limited possibility of surface modification (texture and functionality) for tuning the surface affinity of adsorbents to a specific pollutant, which is a challenging task, and their adsorption capacity is difficult to control.

Therefore, despite the significant progress in the development of the systems and methods for water remediation, there is still a need in the novel adsorptive materials with tunable functional characteristics. Nanotechnology provides a solution for this challenging problem concerning the treatment and purification of drinking water and/or industrial (leachate) water [55]. This strategy is based on design, fabrication, and utilization of advanced nanoadsorbents providing efficient pollutant removal [56]. To date, a variety of well-performing, eco-friendly, and cost-effective nanomaterials with versatile functionalities has been developed for decontamination of industrial effluents, surface water, ground water, and drinking water.

Nanoadsorbents can be divided in the following groups: metallic nanoparticles, nanostructured mixed oxides, carbonaceous nanomaterials and silicon nanomaterials including silicon nanotubes, silicon nanoparticles, silicon nanosheets as well as nanoclays, polymer-based nanomaterials, nanofibers, aerogels, and their composites with biochar [56].

These nano-engineered adsorbents show enhanced performance for water treatment and environmental remediation as compared to classical adsorbents [57]. However, their weak points are relatively high cost and often multistep synthesis.

Nanoadsorbents according to their morphology can be classified as nanotubes, nanorods, nanowires and nanosheets [58,59]. In particular, recent development on carbonaceous nanomaterials resulted in the exploitation of carbon nanotubes (CNTs, 1D), carbon nanoparticles (0D) and carbon nanosheets (2D) [60]. Factors controlling the nanoadsorbent properties are the particle and pore size, surface chemistry, agglomeration state, shape and fractal dimension, chemical composition, crystal structure and solubility [61].

Carbonaceous nano-engineered materials include CNTs, i.e., single-walled (SWCNTs) or multi-walled (MWCNTs), carbon nanofibers, fullerene, graphene (Gn) and its derivatives, and amorphous carbonaceous composites of other type matrices, such as zeolites [57,61]. A representative number of literary reports confirmed their efficiency and advanced performance for water remediation when incorporated in membranes, filters, or used as adsorbents [62].

The present literature shows that further search, design, and fabrication of novel carbon-based nanoadsorbents are topical. Note that, as a rule, most high-quality relevant reviews devoted to the carbon-based nanomaterials published to date consider mostly their separate kinds.

The scope of this review is restricted to Gn, graphene oxide (GO), rGO, CNTs, and their composites with other types of nanoporous materials, including MOFs as well as MC nanomaterials. Biochars and biochar-based composites, especially, MOF-biochar composites, are also the focus of this work (Figure 1). This is because, despite the differences between biochars and carbon-based nanoadsorbents, their composites open a new ecofriendly and scalable pathway for the utilization and recycling of different kinds of reuse biomass as low-cost adsorbents for water remediation with a potential for inheriting the high surface area of MOFs. The reason for combining these popular adsorbents known to date in one review helps to analyze and evaluate the most prospective search directions in the field of preparation and exploitation for the adsorptive removal of organic pollutants and heavy metal ions from aquatic media.

We intend to analyze and compare the peculiarities of adsorbent performance based on the specific adsorption mechanism realized using each type of carbon-based nanoadsorbents. The authors hope that this analysis will assist to evaluate the potential of the specific carbon-based nanomaterials in a predictable way for the most efficient adsorptive removal of organic pollutants from water and could contribute to their further development and exploitation.

## 2. Adsorption Isotherms

In order to understand the adsorption performance, the adsorption isotherm at a specified temperature can be analyzed. Several well-known models have been considered in the observation and evaluation of best adsorbate fitting model, namely Langmuir, Freundlich, Temkin, Redlich-Peterson isotherms, etc. The Langmuir isotherm (Equation (1)) explains adsorption based on the assumption of a mono-molecular layer, while the Freundlich model (Equation (2)) was developed from unlimited uptake adsorption with an uneven-heat-distribution surface. The mentioned isotherm models can be depicted as follows:(1)$$\frac{C_{e}}{q_{e}}=\frac{C_{e}}{Q_{0}}+\frac{1}{bQ_{0}}\;$$
(2)$$lnq_{e}=lnK_{f}+\frac{1}{n}lnC_{e}$$
where $C_{e\;}\left(\frac{\mathrm{mg}}{L}\right):$ the concentration at equilibrium, $q_{e}\;\left(\frac{\mathrm{mg}}{g}\right):$ the amount of adsorbate adsorbed per unit mass of the adsorbent, $Q_{0}\;\left(\frac{\mathrm{mg}}{g}\right),\;b\;\left(\frac{L}{\mathrm{mg}}\right):$ constants of the Langmuir model and $K_{f\;}\left(\frac{\mathrm{mg}}{g}\right)$, *n*: constants of the Freundlich model.

## 3. Carbon-Based Nanoadsorbents

The important advantages of modern carbon-based nanomaterials, i.e., CNTs, GN, GO, rGO as well their functional composites are unique physico-chemical characteristics, such as highly ordered structure, homogeneous pore size distribution, chemical and thermal stability, high specific surface area, high porosity and environment-friendly nature [63,64,65,66,67,68]. In addition, the surfaces of these materials can be functionalized to target specific pollutants via chemical or electrical interactions [57]. Moreover, the highly ordered structure of these carbon-based nanoadsorbents is the main difference between them and activated carbons.

Therefore, carbon-based nanoadsorbents have shown a particular potential for water remediation by removal of a large variety of toxic organic and inorganic pollutants from solution [69,70,71,72,73,74,75].

### 3.1. CNTs

CNTs together with Gn and GNs are the most widely studied carbon nano-allotropes [76]. Starting from their discovery in 1991, CNTs have been recognized as promising functional materials in many application fields, such as medicine and environment remediation [77,78,79,80]. In particular, CNTs have been studied for water purification and treatment including seawater desalination [81].

#### 3.1.1. Structural and Functional Properties of CNTs

The physical, chemical, electrical, and structural properties of CNTs are related to their unique structure [82,83,84,85,86]. CNTs are composed of one or several graphene sheets of hexagonally arranged carbon atoms with sp^2^ hybridization rolled into a tube forming SWCNTs with a diameter of ~0.4–2.0 nm or MWCNTs with diameters of 2–100 nm and a distance of 0.34–0.35 nm between sheets, respectively (Figure 1) [87,88]. Their uniform pore structure is organized as a system of 1D channels. A SWCNTs is one carbon atom thick [89].

Accordingly, the structure of MWCNTs contain two or more SWCNTs stacked on each other, so it can be represented as a concentric arrangement in which successive tubes increase in diameter. MWCNTs have a number of walls, which may vary from two to an unlimited upper number (Figure 1 right). Thus, the properties of each of the two types of CNTs are determined by the atomic configurations of the nanotube cylinders [81].

Moreover, CNTs’ intrinsic properties are their high aspect ratio (i.e., the length to width ratio) and extremely small pore size. CNT walls are not reactive, but their fullerene-like tips are known to be more reactive, so functionalization of CNTs is often used to obtain functional groups (e.g., –COOH, –OH, or –C=O) [91].

The chemical structure, chemical composition including functionalization, distribution of dimensions (uniformity of sample and aspect ratios), physicochemical properties of the surface, agglomeration state and presence of extraneous metallic nanoparticles (from the manufacturing processes) were identified as the most important parameters of CNTs [55].

In sum, CNTs are among the most promising carbon-based nanomaterials owing to their tunable properties that differentiate them from other solid materials in the relevant fields. In particular, the hexagonal arrays of carbon atoms in graphite sheets of the CNT surface have a strong interaction with other molecules or atoms, which contributes to the CNT efficiency for the adsorptive removal of organic pollutants from water as compared to conventional activated carbon adsorbents [55].

Actually, release of some species deposited in CNTs during the preparation process is an important issue in terms of additional contamination of aquatic media due to usage of the adsorbents themselves, a fact which draws particular attention to the relevant literature [92].

#### 3.1.2. CNTs for Adsorptive Removal of Organic Pollutants

The unique combination of the remarkable structural and functional properties of CNTs provides their high potential in environmental applications. Nanotechnologies based on CNTs are utilized in a number of water treatment processes, such as those that use sorbents, catalysts, filters, or membranes.

To date, CNTs have been recognized as the promising adsorbents for the removal of viruses, bacteria, and organic pollutants from water [93,94,95,96,97,98]. In the context of adsorptive removal of organic pollutants, benzene rings in graphene sheets contain sp^2^-hybridized carbon atoms with high polarizability [89] that makes CNTs hydrophobic materials [99], which can interact with aromatic contaminants through hydrophobic interactions and π-π-stacking interactions [100]. Moreover, CNTs have micropores and mesopores and different functional groups, such as –COOH, –OH, and/or –NH_2_ [101].

All these characteristics are important factors which contribute to the efficient adsorption of organic pollutants from different aquatic media (drinking water, leachate water, etc.) [102,103].

#### 3.1.3. Modification of CNTs

An important advantage of CNTs is the possibility of rational design allowing for control of textural characteristics, which can be further improved by activation and/or modification [104]. This possibility expands the application fields of CNTs including adsorption in liquid phase.

The modification techniques alter the characteristics of CNTs surface, e.g., functionality, as well as porosity, dispersion and hydrophobicity [90,105]. Various kinds of surface modification techniques are reported, which include acid treatment, metal impregnation and functional molecules/groups grafting [56]. In particular, acid treatment of CNTs is carried out by using different oxidants, including HNO_3_, KMnO_4_, H_2_O_2_, and H_2_SO_4_.

The introduction of additional functional groups onto the surface of CNTs is a widely used strategy to increase the adsorption capacity for the adsorptive removal of different water contaminants [106,107,108]. Figure 2 shows CNT surface modifications with different functional groups.

CNTs can be functionalized through covalent or non-covalent bonding techniques [107]. In particular, grafting functional molecules/groups on the surface of CNTs is an efficient way to improve their surface characteristics, solubility, dispersion, and adsorption capabilities due to improvement of their interaction modes with adsorbates [56].

Covalent functionalization results in the attachment of different functional groups to the outer walls of CNTs through covalent bonds [107,109]. In case of non-covalent functionalization, different biochemically active molecules can be attached to CNT walls through hydrogen bonds, Van-der-Waals forces, π-π-stacking interactions, hydrophobic or electrostatic interactions [110,111]. Non-covalent functionalization is considered as a more preferable modification because the structural characteristics of the pristine CNTs are retained in this case [109]. For instance, by using APTES, the covalent functionalization of carbon nanotubes can be realized, without any destruction of the CNT structure [112].

However, a possible release of the modifying agent, which could contribute to the additional water contamination, is an important problem in this case. So, a covalent modification of the CNT surface is preferable from this point of view.

Another possibility of CNT modification is the preparation of the functional composites on their basis. In particular, CNTs tend to aggregate in water because of their hydrophobicity and high length-to-diameter ratio [113]. A simple combination of CNTs with magnetite affording modified magnetic CNTs assists their dispersibility in water as well as reusability and recovery after adsorption [59]. Figure 3 shows the preparation route of the magnetite decorated MWCNTs preliminary modified with acid groups [114]. These Fe_3_O_4_/MWCNTs-COOH composites were utilized in the removal of Cu(II) from aqueous solutions.

An improvement of the adsorption performance of CNTs for organic pollutant removal can be achieved by preparation of their composites with ILs. The addition of ILs makes hydrophobic CNTs more hydrophilic, which may contribute to its homogeneous dispersion in water phases and enhance interactions between IL@CNT adsorbents and pollutants in water.

For the preparation of the IL@CNT composites, an impregnation method is commonly utilized (Figure 4) [115]. In IL@CNT composites, the potential release of ILs from the CNT surface should be evaluated to avoid the potential additional contamination with these modifying agents. However, some reported results demonstrate a good recovery and recyclability of IL@CNT adsorbents (ibid).

An increase of adsorption capacities has been shown for carbon nanotubes modified with 1-benzyl-3-hexylimidazolium and 1-benzyl-3-decahexylimidazolium ionic liquids as compared to the pristine CNTs for SMZ and KET [116]. The following order of Q_m_ values was reported for CNTs and IL@CNT composites as follows: CNT + KET/SMZ < CNT@ILs + SMZ < CNT@ILs + KET.

Kinetics, isotherm and computational studies were carried out to determine the adsorption efficiency and mechanism of SMZ and KET on IL@CNT composites. The adsorption of SMZ and KET on CNTs and modified CNTs was pH dependent, and it was best described by pseudo-second-order kinetics and the Freundlich adsorption isotherm. DFT analysis was used for better understanding the mechanism of adsorption at the molecular level through the estimation of electronic properties of the studied adsorbents. It was suggested that several mechanisms are responsible for the adsorption on pristine CNTs and IL@CNT composites in an agreement with reported in silico analysis for analogous systems [117]. So, in case of IL@CNT composites, the adsorption mechanism includes π-π interactions between π systems on CNT surfaces and benzene rings in organic molecules, hydrogen bonds involving functional groups on CNTs and ionic liquid modified CNTs surfaces, and electrostatic interactions.

In this context, the most innovative CNTs-based composites are obtained by a combination of CNTs with the new type of hybrid nanoporous materials, i.e., MOFs. This type of composites will be discussed in Section 4.

#### 3.1.4. Mechanism of Adsorption of Organic Contaminants onto CNTs

The adsorption mechanism of organic contaminants onto CNTs is determined by both CNT and adsorbates characteristics (polar and non-polar) as well as conditions of the adsorption process [95,118].

The adsorptive removal of organic pollutants from water on CNTs may be realized as physisorption and/or chemisorption, associated with their transport properties contributing to the intra-particle diffusion [119]. So, mechanisms of the organic pollutant adsorption on CNTs may include π-π interactions, H-bonding, electrostatic and hydrophobic interactions as well as their combinations [120]. The Figure 5 shows schematically different mechanisms realized for the adsorption of pharmaceuticals on MWCNTs [121].

The functional groups on the CNT surface may act as adsorption sites for the bulk organic molecules, while meso- and micropores in the CNT structures may serve as adsorption sites for small molecules and species [122]. Therefore, adsorptive removal of organic pollutants using CNTs is dependent on their physico-chemical properties including specific surface areas, surface charge and hydrophobicity as well as composition of a water solution and its pH value and ionic strength [123]. However, due to a number of these factors, e.g., adsorption activity of the CNT surface, an interpretation of the specific adsorption mechanism is rather complicated [124]. Especially, revealing the adsorption mechanism is difficult for the real water probes, which are usually multicomponent systems, so interactions of adsorbates and their competition for the adsorption sites associated with pore blocking by the bulkier pollutant molecules contribute to the adsorption process.

##### π-π Interactions

As a rule, the molecules of organic pollutants contain π-electrons, which may form π-π bonds with CNTs. Functional groups in benzene rings of organic molecules (in particular, pharmaceuticals) contribute to the strength of π-π bonds. However, π-π EDA interactions are often a dominant adsorption mechanism for organics pollutants, e.g., pharmaceuticals on CNTs [125]. π-π EDA interactions are realized mainly between electron-deficient aromatic rings and electron-rich aromatic moiety on the graphene surface of CNTs and adsorbates, respectively. It is suggested that the graphene structure contains both electron-donor functional groups (π-electrons)—oxygen-containing functional groups—and acceptors of π-electrons [126,127]. This property contributes to the facile bonding of several classes of aromatic compounds with strong π-electron-donor groups, such as OH groups in phenols [128] or π-acceptor groups, such as NO_2_ in nitrobenzene [129] with CNTs.

For instance, π-π interactions are a key adsorption mechanism for carbamazepine on CNTs [130]. Benzene rings of carbamazepine act as acceptors of π-electrons due to electron-acceptor ability of amide groups in the carbamazepine molecule. π-Electron-donor species on the graphene surface are located in aromatic rings and in carbonyl groups. π-π EDA interactions are also suggested as a mechanism of adsorption of carbamazepine on CNTs [131].

The next possibility is n-π EDA interaction (probably, in combination with π-π interactions), which involves a direct interaction of a lone electron pair of nitrogen and oxygen in the amino group, hydroxyl group, O- or N-heteroatoms as n-electron donors with π-electron-acceptor sites on the CNTs surface. Interactions of n-π EDA type may be reinforced by ionized amino- and hydroxyl groups at high pH values [132]. For instance, this mechanism was proposed as a dominating mechanism for thiamphenicol adsorption on CNTs [121].

##### Hydrogen Bonding

Oxygen-containing functional groups on the CNT surface may interact with –OH, –NH, and –NH_2_ groups organic molecules via hydrogen bonding that contributes to the organic pollutant adsorption on CNTs [133,134]. It was reported [121] that hydrogen bonding is one of the main mechanisms for the adsorption of different organic pollutants on CNTs. Actually, according to IR spectroscopy, there are hydrogen acceptors and donors on the CNT surface, while organic adsorbates have acceptor or donor groups for hydrogen bonding. In addition to hydrogen bonding, π-π interactions, hydrophobic interaction, and Lewis acid-base interaction participate in the adsorption process.

It was suggested [135] that hydrogen bonding is a dominating adsorption mechanism for acetaminophen on MWCNT. Similar results were obtained [136] for the adsorption of norfloxacin on CNTs. Actually, the norfloxacin molecule contains several fragments, which can participate in H-bonding as acceptors or donors, and thereby form H-bonds with functional groups on the adsorbent surface. Additionally, benzene rings on the adsorbent surface can act as H-bond donors and form hydrogen bonds with functional groups in the norfloxacin molecule. An additive effect of the multiple H-bonds between norfloxacin and CNTs may improve adsorption affinity [137]. A similar conclusion was reported recently [111]: it was suggested that diclofenac and triclosan adsorption on MWCNTs involved hydrogen interactions because the adsorbent and adsorbates had specific functional groups (–COOH, –OH), which could contribute in H-bonding. Moreover, π-π interactions, electrostatic interactions, and hydrophobic interactions may participate in adsorption [136,138].

##### Electrostatic Interactions

The appearance of a positive and negative charge on the CNT surface due to deviance from pH point of zero charge creates an electric potential on the adsorbent. Similarly, the organic pollutant surface becomes charged due to protonation and deprotonation at different pH values of the solution. These electrostatic charges both on the organic pollutant surface and CNT surface result in adsorption due to electrostatic interactions. The strength of these interactions changes at protonation/deprotonation of functional groups [132].

According to the specific surface charge determination by the measurement of streaming potential, SWCNT and MWCNT samples were negatively charged at pH values exceeding 3 (PZC of both samples was 3) [139]. The maximal adsorption value of triamterene at pH = 7 may be ascribed to the electrostatic attraction between the molecular form of triamterene and surfaces of MWCNTs and SWCNTs with a negative charge. The pKa of triamterene is 6.2 [140], and at pH = 7, the main fraction of triamterene exists in a molecular form. In its turn, amino-groups in the triamterene molecule can dissociate with the formation of H^+^ cations and corresponding anions. It was reported [141] that TC and ibuprofen (Figure 6) removal by CNTs proceeded through electrostatic interactions between them. Similar results were obtained elsewhere [142]. It was found that electrostatic interactions are responsible for the removal of diethyl phthalate and oxytetracycline on CNT-based adsorbents at pH = 7 and 25 °C.

According to [143], the maximal adsorption value for CIP (Figure 2) on CNTs was observed at pH = 6 and 25 °C. Note, at pH below 6.1, CIP exists in the cationic form in solution, and is attracts electrostatically to the negatively charged CNT surface according to CIP pKa and CNT pHPZC. Additionally, it was found that adsorption decreased at pH > 6 because in this case CIP is transformed into an anionic form. This results in its electrostatic repulsion with the negatively charged CNT surface. According to [143], electrostatic interaction is suggested as a dominating mechanism in this adsorption process. The analogous result was obtained by other authors [144,145]. The maximal adsorption capacity for caffeine and diclofenac was observed at pH = 7 and 25 °C [145], where the surface charge of MWCNT was negative, and adsorbate molecules had a positive charge. However, the experimental results show that hydrophobic and π-π interactions are also realized [144,145,146].

##### Hydrophobic Interactions

Hydrophobic interactions or the hydrophobic effect is an interaction between non-polar particles in water or other polar solvents, which is caused by the thermodynamic disadvantage of a water contact with non-polar species [147]. The role of the hydrophobic effect at organic pollutant adsorption on carbon-based adsorbents, including CNTs and graphene oxide, is evaluated with the single-point distribution coefficient *K_d_* [130]:(3)$$K_{d}=\frac{q_{e}}{C_{e}}$$

If hydrophobic interactions are the main adsorption mechanism, *K_d_* may correlate with the molecular descriptor of organic phase separation, for instance, separation coefficient of the mixture octanol-water (*K_ow_*) [130].

It was reported [148] that hydrophobic interactions are a dominating adsorption mechanism for the removal of sulfonamides. It was suggested [149] that hydrophobic interactions play a decisive role in chlorophenol adsorption on SWCNTs. A linear correlation between log*K_ow_* and log*K_d_* indicates that adsorption ability correlates positively with hydrophobicity. Moreover, the benzene ring in chlorophenol may trigger π-π EDA interaction with the SWCNT surface [146,150], because a strong electronegativity of the Cl atom decreases the electron density of the benzene ring and makes chlorophenol a π-electron-acceptor.

Hydrophobic interactions play an important role in the adsorption of four drugs, i.e., KET, NAP, diclofenac, and ibuprofen (Figure 2) on hydrophobic areas homogeneously distributed on the MWCNT surface (Table 1, entry 2) [151]. However, the absence of the correlation between hydrophobicity of pharmaceuticals and their adsorption affinity towards MWCNT indicates the existence of other types of interactions: π-π and n-π EDA interactions. This behavior is well known for interactions of organic molecules and CNTs [134]. The main mechanisms of the adsorption of carbamazepine and dorzolamide may be also hydrophobic bonds and π-π EDA interactions [152].

The Q_m_ values of the different CNT materials for specific organic pollutants, impact of activation/modification of the adsorption efficiency, dominating adsorption mechanism and effect of conditions of the removal process are listed in Table 1.

### 3.2. Gn-Based Nanoadsorbents

Gn is one of the carbon allotropes with outstanding features, which provide the potential for several environmental applications. It is a single two-dimensional (2D) one-atom-thick sheet comprised of sp^2^ hybridized carbon atoms arranged as a honeycomb lattice.

Gn demonstrates minimal solubility in water and organic solvents. Methods of graphene synthesis involve two main strategies [156]. The first approach includes a structural decomposition of carbon materials through various techniques, such as lithographic cutting of graphite sheets, unpacking of CNTs or sonochemical extraction of graphene sheets [157].

The second strategy involves Gn production from simple constituents, such as methane and ethanol or other liquid precursors [158]. The first method is widely used due to a high yield of the final product and low cost. However, it may result in the preparation of thin-filmed nanostructured graphene with a low content of defects [159].

Due to their unique physico-chemical characteristics, such as high electron conductivity, mechanical robustness, thermal stability, Gn-based materials have applications in many fields [160,161,162]. In particular, graphene-based nanomaterials have been studied for the potential application in water purification including adsorptive removal of organic pollutants. For instance, the NiO/Gn nanosheets composite has shown the CR adsorption capacity of 123.89 mg/g (pH = 7.0, 25 °C) [163].

#### 3.2.1. GO

The original Gn is an inert and hydrophobic material, so it interacts with different pollutants extremely hard [164]. Therefore, its chemical modification is needed for practical exploitation [165]. An example is graphene oxide preparation. GO is an oxidized derivative of graphene and precursor for the fabrication of GO derivatives (Figure 2) [166].

GO is a carbon nano-material with a two-dimensional structure produced by oxidation of a graphite layer [56]. GO has a few intrinsic features as compared to other nanomaterials, such as CNTs. A single layer GO has two-dimensional basal planes available for the maximum adsorption of toxic pollutants. It can be obtained through a simple synthesis process, which can be carried out by chemical exfoliation of graphite without using any metallic catalyst and complicated instruments (Figure 7) [167].

The synthesis of GO from Gn allows one to increase the distance between the sheets and change its surface functional properties. This improves significantly GO adsorption characteristics [168].

#### 3.2.2. Structural and Functional Properties of GO

The basal plane of GO is occupied by epoxides and hydroxyls, whereas carboxyl, ketone, and aldehyde groups are located, mainly, at the edge plane of GO [169]. Thus, oxygen-containing functional groups introduced by graphene oxidation assist in the proper distribution of formed GO in water or organic solvent as well as interact with different inorganic and organic contaminants in the adsorption process [170,171].

Numerous oxygen-containing functional groups in GO decrease its electroconductivity, this is undesirable for several catalytic reactions [172,173]. In order to solve this problem, oxygen loading in GO is decreased using several techniques [174].

GO is a promising adsorbent for the removal of metal ions, dyes and organic micro-pollutants due to its high surface area, mechanical strength, light weight, flexibility, and chemical stability. Moreover, hydrophilic functional groups on the surface of GO assist in the adsorption process [57] In particular, the presence of hydroxyl and carboxyl groups in GO increases the adsorption of toxic pollutants. An example is utilization of graphene-based nanomaterials for the adsorption of heavy metals, such as Cr(VI), Pb(II), Hg(II) from wastewater [57,166].

Note, that inevitable release of GO particles and their transformation in the aquatic media under light exposure as well as under dark conditions is an important issue. To date, a few comprehensive reviews evaluate the environmental fate and potential toxicity of GO for aquatic organisms [175].

In order to optimize the surface properties of GO, its further modification with functional groups may result in triggering weak Van-der-Waals forces, π-π-stacking and electrostatic interactions (Figure 8) [176]. For instance, GO could effectively adsorb metal ions via electrostatic attraction, coordination interactions and other interaction forces [166].

In several cases, the reduction of GO into rGO is demanded for the optimization of its surface properties and tuning hydrophobicity. This reduction process allows one to recover the hexagonal structure of a graphene honeycomb lattice and sp^2^-bonds disturbed in the oxidation process [177]. rGO is obtained by oxidation and exfoliation of GO followed by chemical reduction with different reducing agents, such as sodium borohydride (NaBH_4_), hydrazine (N_2_H_4_·H_2_O), hydrogen iodide acid (HI), ascorbic acid etc. [160]. Obtained rGO is rather similar to pure graphene with exceptions of some variations [178]. The π-conjugated bonds prevail in the rGO structure after removal of oxygen-containing functional groups [179]. The rGO electroconductivity is increased after reducing the sp^2^-conjugated graphene structure [180]. However, removal of oxygen-containing functional groups in rGO decreases the distance between interlayer sheets due to weaker repulsion forces between them [181]. The number of adsorption sites may be decreased, because rGO tends to aggregate due to strong Van-der-Waals and π-π interactions between rGO layers [182].

GNs have been studied for potential applications in water purification. Starting from their first usage as an adsorbent in 2004 [183], GO and rGO are widely used for adsorption of different hazardous substances in water, such as pigments [184,185,186,187], antibiotics [188,189,190,191], phenol and its derivatives [192,193,194], oils [195,196], and pesticides [197].

Moreover, the oxygen-rich graphene layers with high hydrophilicity in GO can be used to fabricate various nanocomposites with enhanced adsorption properties [198]. For instance, Mg(OH)_2_–containing hexagonal nanosheet-graphene oxide composites were utilized for CR removal with enhanced maximal adsorption capacity as high as ~117.5 mg/g with 400 mg/L of the adsorbent loading and 20–50 mg/L of the CR loading (25 °C). It was found that adsorption was based on electrostatic attraction (pH = 7) [199]. Du et al. have fabricated the graphene oxide/chitosan fiber with the maximum CR adsorption capacity of 294.12 mg/g using the 700 mg/L adsorbent loading in 20 to 200 mg/L CR solution at pH = 5.0 and 20 °C [200]. The functional GO composites with MOF materials will be discussed in Section 4 below.

#### 3.2.3. Mechanisms of the Organic Pollutants Adsorption onto GO

The adsorption mechanism for removal of organic pollutants using GO and rGO will be discussed taking pharmaceuticals as adsorbate examples. The adsorptive removal of organic pollutants from water on GO adsorbents is determined significantly by the unique graphene structure. The aromatic rings in organic pollutants may interact with graphene through the implementation of different mechanisms. There are four possible kinds of adsorptive interactions between drugs and GO [201,202,203,204] including π-π interaction [205], hydrogen bonding of organic molecules with hydrogen in carboxyl or hydroxyl groups on the GO surface [205,206], hydrophobic interaction of organic molecules with hydrophobic groups on GO [207,208,209,210] and electrostatic interactions [201]. Thus, these mechanisms provide an efficient adsorption for organic contaminant removal using GO.

The role of specific mechanisms will be discussed below.

##### π-π Interactions

Due to the numerous oxygen-containing functional groups and aromatic rings on the GO surface, it demonstrates acceptor or donor properties towards π-electrons [211,212,213]. The mechanism of π-π interactions explains better physical adsorption of carbamazepine on the GO surface [130]. Similar to numerous aromatic organic compounds [214,215], carbamazepine has a plane conjugated π-electron system, which may overlap with surface π-electron orbitals of the GO aromatic moiety. Metformin adsorption on GO [216] proceeds mainly through π-π interactions, because metformin can interact with aromatic rings of GO through π-π interactions [213,217]. However, hydrogen bonds can also appear between amino groups in metformin and oxygen-containing groups of GO [216].

π-π Interactions play an important role in the adsorption of sulfamethoxazole over GO [218,219]. It was shown [220] that aromatic rings of sulfamethoxazole were efficient acceptors of π-electrons, because electron-acceptor sulfur-containing groups of sulfamethoxazole molecules may attract electrons. Simultaneously, oxygen-containing functional groups of GO feature a high electronegativity. Analogously, paracetamol adsorption on GO proceeds mainly due to π-π EDA interaction [220].

It was suggested [221] that electrons restricted by aromatic rings may provide stronger π-π interactions than those delocalized in aromatic systems. Based on this suggestion [222], the quantitative criterion was formulated, which characterized a capability of aromatic molecules to π-π interactions, i.e., LOLIPOP index. According to [222], the benzene ring with a lower LOLIPOP value has stronger π-stacking-ability. It was shown [223] that an order of the LOLIPOP value for salicylic acid corresponded well to the order of adsorption ability. Therefore, π-π interaction is a main factor in the regulation of adsorption ability.

##### Hydrogen Bonding

Two types of hydrogen bonds are formed between organic pollutants and functional groups on GO during adsorption: direct hydrogen bonds between organics molecules and GO and bridging hydrogen bonds between organics molecules, water, and GO. Taking into account the fact that it is impossible to separate an energy of interaction of H-bond from the total interaction energy including a hydrogen bond, π-π-stacking interaction etc., the direct hydrogen bond between GO and aromatic compounds, e.g., bisphenol A, nitrobenzene, phenol, benzoic acid, and salicylic acid, is not a dominating factor, which determines an adsorption ability [223,224].

For instance, it was found [225] that the mechanisms of diclofenac adsorption on GO included not only hydrogen-bonding but π–π-stacking and hydrophobic interactions. According to [225,226,227], diclofenac is a weak acid (pKa = 4.2). The efficiency of the adsorptive removal for diclofenac reached ~96% at pH < pKa, then slightly decreased up to ~81% along with the pH increase up to 9. This high adsorption value at pH < pKa was not caused by electrostatic attraction, because diclofenac existed in its neutral form, while rGO adsorbent was positively charged (pHPZC = 6.3). Therefore, the improved diclofenac adsorption in a highly acidic medium was related to other adsorption mechanisms, such as hydrogen bonding, hydrophobic interaction and π-π interaction [228,229]. Diclofenac dissociation proceeds at pH > pKa, which results in its negative charging. Therefore, electrostatic attraction plays the main role in diclofenac adsorption on the positively charged rGO surface in the range of pH values between 4.2 (pKa diclofenac) and 6.3 (pHPZC rGO). At pH > 6.3, the rGO surface becomes negatively charged, this results in a decrease of diclofenac adsorption, because electrostatic repulsion between anionic diclofenac and negatively charged rGO becomes dominating. However, the adsorption value of diclofenac removal is rather high at pH values ranging from 6.3 till 9.0, i.e., a slight decrease from ~89 till 81% was observed. This observation shows that non-electrostatic adsorption mechanisms, such as hydrogen bonding and hydrophobic interaction, may contribute to the diclofenac removal in an alkali solution [230].

##### Electrostatic Interactions

Due to charged functional groups in organic molecules and on the GO surface, an electrostatic interaction may play an important role in the relevant adsorption process. For instance, ciprofloxacin molecules are adsorbed on graphene oxide through electrostatic interactions [219]. This suggestion is confirmed by the impact of pH values and ionic strength of the solution on the GO adsorption capacity for ciprofloxacin. According to [231], the rGO surface charge of PG sample as well as protonation/deprotonation of surface functional groups at different pH values play a decisive role in the adsorption process. As it is pointed above, rGO has carboxyl (–COOH), hydroxyl (–OH) and epoxide (–C–O–C–) functional groups on its surface and inside pores.

Pharmaceuticals, such as atenolol and carbamazepine contain NH_2_-groups, while ciprofloxacin and diclofenac contain NH–groups, so these pharmaceuticals may interact with rGO according to following reaction equations [231]:(4)$$\mathrm{PG}-{\mathrm{COO}}^{-}+\mathrm{Drug}-{\mathrm{NH}}_{3}^{+}\overset{\mathrm{Electrostatic}\;\mathrm{forces}}{\to }\mathrm{PG}-{\mathrm{COO}}^{-}{\mathrm{NH}}_{3}^{+}-\mathrm{Drug}$$
(5)$$\mathrm{PG}-\mathrm{OH}+\mathrm{Drug}-{\mathrm{NH}}_{2}\overset{\mathrm{Hydrogen}\;\mathrm{bonding}}{\to }\mathrm{PG}-\mathrm{OH}\cdots {\mathrm{NH}}_{2}-\mathrm{Drug}$$
(6)$$\mathrm{PG}-\mathrm{OH}+\mathrm{Drug}-\mathrm{NH}\overset{\mathrm{Hydrogen}\;\mathrm{bonding}}{\to }\mathrm{PG}-O^{H}\cdots {\mathrm{HN}}^{H}-\mathrm{Drug}$$
(7)$$\mathrm{PG}-\mathrm{OH}+\mathrm{Drug}-\mathrm{NH}\overset{\mathrm{Hydrogen}\;\mathrm{bonding}}{\to }\mathrm{PG}-\mathrm{OH}\cdots \mathrm{NH}-\mathrm{Drug}$$
(8)$$\mathrm{PG}-\mathrm{OH}+\mathrm{Drug}-\mathrm{NH}\overset{\mathrm{Hydrogen}\;\mathrm{bonding}}{\to }\mathrm{PG}-O^{H}\cdots \mathrm{HN}-\mathrm{Drug}$$

Electrostatic interactions may be realized, mainly, between protonated amino groups NH_3_^+^ and negatively charged groups of GO (Equation (4)). However, hydrogen bonds may appear between hydrogen atoms in GO hydroxyl groups and nitrogen atoms in amino groups of pharmaceuticals (Equations (5)–(8)). Hydrogen bonding may also take place between a pharmaceutical molecule and GO in the following way (Equations (9) and (10)):(9)$$\mathrm{PG}-\mathrm{COOH}+\mathrm{Drug}=O\overset{\mathrm{Hydrogen}\;\mathrm{bonding}}{\to }\mathrm{PG}-\mathrm{COOH}\cdots O=\mathrm{Drug}$$
(10)$$\mathrm{PG}-\mathrm{OH}+\mathrm{Drug}=O\overset{\mathrm{Hydrogen}\;\mathrm{bonding}}{\to }\mathrm{PG}-\mathrm{OH}\cdots O=\mathrm{Drug}$$

Moreover, hydrophobic and π-π interactions between localized and π-electrons of GO aromatic rings and drug molecules may contribute to the adsorption process [231].

It was reported [232] that maximal adsorption values for atenolol and propranolol were observed at pH < 4. Note, pKa values of atenolol and propranolol exceeded 9.0. In this case, the cumulative surface charge was negative, while drug molecules had a positive charge. Therefore, the electrostatic forces between negatively charged GO functional groups (–COO etc.) and positively charged functional groups in drug molecules (mainly, protonated amino groups NH^3+^) may play a dominating role in the adsorption process under strong acidic conditions:(11)$$\mathrm{GO}-{\mathrm{COO}}^{-}-{\mathrm{NH}}_{3}^{+}\overset{\mathrm{Electrostatic}\;\mathrm{forces}}{\to }\mathrm{GO}-{\mathrm{COO}}^{-}{\mathrm{NH}}_{3}^{+}-\mathrm{Drug}$$

Under alkaline conditions (pH > 8), drug adsorption is affected by the electrostatic attraction between negatively charged GO functional groups, such as –COO– and positively charged drug molecules (–NH_3_^+^). Moreover, at pH > 8, the adsorption is also implemented due to hydrogen bonds between: (1) H atoms in hydroxyl groups of GO and N atoms of amino groups in drugs; (2) oxygen atoms in hydroxyl groups of GO and H atoms of amino groups in drugs:(12)$$\mathrm{GO}-\mathrm{OH}+\mathrm{Drug}-{\mathrm{NH}}_{2}\overset{\mathrm{Hydrogen}\;\mathrm{bonding}}{\to }\mathrm{CO}-\mathrm{OH}\cdots {\mathrm{NH}}_{2}-\mathrm{Drug}$$

In addition to the interactions mentioned above, other forces, such as π-π interactions between localized π-electrons in conjugated aromatic rings of GO and drug molecules may contribute in the adsorption mechanism [232]. It was reported [228,233] about analogous effects and interactions realized in the adsorption of different drugs onto GO. For instance, a higher adsorption value was observed for diclofenac (pKa 4.15) and sulfamethoxazole (pKa 5.7) on GO under acidic conditions (pH < pKa), but not alkaline conditions (pH > pKa), so it may be suggested that the adsorptive removal of these drugs depends mainly on hydrophobic and π-π interactions [228]. Adsorption of tetracycline doxycycline and ciprofloxacin on GO is realized mainly due to electrostatic interactions, as confirmed by relevant investigations [234]. However, π-π interactions between drug molecules and GO may contribute to the adsorption process.

##### Hydrophobic Interactions

Taking into account a consideration that the aromatic moiety of pharmaceuticals assists in their adsorption on the GO surface, it was suggested [235] that hydrophobic interactions played an important role in adsorption of PAE on GO and rGO. PAE are used widely as plasticizers in the production of various polymeric products for industrial, domestic, food, and medical purposes, from which they are easily eliminated by heating [236,237].

It was reported [235] that rGO had a more hydrophobic surface, and, therefore, it had a higher adsorption ability towards PAE. It is found that PAE molecules may act as electron acceptors due to the ester functionality [238]. Therefore, π-π interactions may play an important role in the adsorption process between PAE and GO or rGO as π-donors. Hydrogen bonds between –OH or –COOH groups on the GO surface and functional groups in the adsorbate molecules may contribute to the PAE adsorption [239,240]. Hydrophobic interactions take place in SMZ adsorption on the hydrophobic surface of a GO sample modified with perfluorinated aromatic compounds [220].

Thus, the contemporary literature shows that four possible interactions may be involved in the adsorption of organic pollutants, such as pharmaceuticals, onto carbon-based nanoadsorbents, i.e., GO and CNTs: (1) π-π interactions of adsorbate molecules with π-electrons of the aromatic moiety of the adsorbent; (2) hydrogen bonding with an H^+^ ion from carboxyl or hydroxyl on the adsorbent surface; (3) hydrophobic interactions of adsorbate molecules with a hydrophobic moiety on the adsorbent surface; (4) electrostatic interactions of functional groups in the adsorbate molecules with those on the adsorbent surface administered by solution pH changing. So, the adsorption mechanisms mentioned above may be utilized as a base for the explanation of the reaction of the molecules of organic pollutants with modern carbon-based nanoadsorbents.

However, these main four adsorption mechanisms do not provide unambiguous insight into the efficiency of the utilization of modern carbon-based nanomaterials for adsorptive removal of organic pollutants from aqueous media. Moreover, GO and CNTs do not always show adsorption capacities towards organic pollutants, which are sufficient for industrial purification processes. This rather low adsorption capacity of CNTs and GO towards organic pollutants may be originated from their moderate or even small specific surface area (BET) as well as their capability to aggregate in the aquatic media due to the hydrophobic surface of these materials.

Therefore, the development of novel adsorbents based on GO and CNTs with high surface areas and pore volumes combined with hydrophilic functionality of their surface is a big challenge. One of the most promising strategies is the preparation of functional composites based on GO/CNTs and other types of materials, such as MOFs.

Table 2 shows the maximal adsorption values for different GO adsorbents measured for various drugs, the influence of GO activation/modification on the adsorption efficiency, dominating adsorption mechanisms and impact of process parameters.

## 4. MOF-Carbon Composites

An efficient way to improve adsorption properties of carbon-based nanoadsorbents is to combine them with other types of materials. In this case, the adsorption performance enhancement may be related to a modification of the textural characteristics, imparting and modulating hydrophilic properties and surface functionality of the formed composites and thereby an enrichment of the possible adsorption mechanisms realized by additional interactions occurring due to another composite component.

As it was mentioned above in the previous sections, there are numerous examples of the relevant literature on the preparation and analysis of adsorption properties of carbon-based composites with different kinds of materials, such as quantum dots, cyclodextrins, magnetic oxides, and silica [242,243,244].

However, the recent literature shows that the most promising components for the preparation and further exploitation of carbon-based composites in adsorptive removal of organic pollutants are MOFs [245]. MOF matrices are assembled of inorganic metal ions or metal oxide clusters as nodes connected by multifunctional organic molecules (linkers) [246,247] resulting in both organic and inorganic sites within the pores [248]. So, MOF building blocks can themselves be functional, which is different from the other nanomaterials. Their structure and composition can easily be tuned through the almost infinite possible combination of metal inorganic subunits (clusters, chains or layers of transition metals, 3p, lanthanides, etc.) and/or organic ligands (carboxylates, phosphonates, azolates, etc.) leading to thousands or a million of MOF structures with unique features [249].

MOF materials [247,250,251] draw increased attention because of their potential applications, such as gas adsorption/storage [252,253], separation [254,255], catalysis [256,257,258], atmosphere [259] and water [260] remediation.

In context of the adsorptive removal of water contaminants, MOF materials show significant advantages as compared to traditional adsorbents, such as record-high surface areas and porosity with a large range of pore sizes (micro- or mesopores) and pore shapes (cages, channels, etc.) [249], periodic fully open porous structure accessible for the guest molecules (loading capability) and rather flexible frameworks (in many cases) [261], various functional groups (e.g., –NH_2_, –OH) in organic linkers, charge conductivity, and open metal coordination sites enabling strong interactions, such as electrostatic and hydrogen bonding interactions [11]. So, due to their unique structural, compositional, textural, and adsorption characteristics, MOF matrices are prospective adsorbents for the removal of organic pollutants from gas phase (air), water, and fuel.

Particularly, for the efficient purification of water and fuel [44,46,47], the physicochemical properties, i.e., pore shape, size, and hydrophobicity, of MOF materials can be tuned by using appropriate functionalization [262]. The polar/apolar character of MOF pore surfaces can be tuned through the introduction of corresponding organic functional groups by direct synthesis or post-synthesis modification of the organic linkers or metal sites [263]. Furthermore, an introduction of electron-rich or electron-deficient functional groups may provide an electrostatic charge on the pore surface. In its turn, OMSs in some MOF structures can serve as chemisorption centers by coordination with adsorbate molecules [264]. Therefore, compared to other classical porous materials like active carbons, mesoporous silicas, and zeolites, MOF matrices show better user-configuration including tailorable chemical affinities towards specific tasks in adsorptive removal of organic pollutants [265]. 

To date, MOF matrices have been studied in the aqueous-phase adsorption of a large variety of pollutants in water, such as heavy metals [266], organics dyes [267], organic arsenic acids [7], PPCPs (e.g., naproxen and clofibric acid) [12], drugs [268,269], PAE [8], pesticides [270,271,272], aromatic and nitroaromatic compounds [273].

Accordingly, several excellent reviews and research papers have justified the potential of MOF materials and their derivatives as a versatile multifunctional platform for hazardous organic removal and the establishment of the relationship between the structure of MOF-based materials and their adsorption performance has been published [260,265,274,275,276,277]. Note, in several works, the adsorptive performance of metal-organic frameworks is usually compared with those of activated carbons (as the standard) to estimate the practical potential of MOF materials in the adsorption processes [278].

To further rationalize the design and synthesis of MOF-based nanoadsorbents, it is important to determine the main factors affecting the adsorptive removal of organic pollutants from aquatic media using these materials. In particular, an important factor determining the adsorption performance is the porous structure of metal-organic frameworks.

Molecular modeling has shown that when the pore sizes of the MOF is larger than the size of the pollutant molecule, the guest molecule preferably resides in the pores of MOFs [260]. Alternatively, the guest molecule is adsorbed on the outside if its size exceeds the diameter of MOF pores. In this context, mesoporous MOFs have the pore size, which is comparable to or larger than molecular sizes of typical organic compounds, except polymers. Therefore, mesoporous MOFs cavity sizes changing in the range of 2–4 nm are particularly promising for adsorption of bulky molecules of organic contaminants, which usually feature high toxicity [279].

The flexibility of the metal-organic frameworks and their breathing properties, which are dynamic structural transformations and pore volume/opening triggered by temperature changes and/or the presence of specific adsorbate or solvent molecules, impact seriously on adsorption behavior of some MOF materials, like MIL-53 matrices [271].

To date, it has been recognized that understanding the nature and kind of interactions between an adsorbate and an adsorbent pave the way to the rational design of the advanced materials for the adsorptive removal of hazardous pollutants from water [280,281]. So, the key factor determining adsorption process efficiency is the mode of adsorption interactions realized in the specific process using specific MOF-based adsorbents.

A number of interactions have been observed to play a key role in different types of adsorption mechanisms. To date, many different mechanisms are defined that are observed during adsorptive removal of hazardous contaminants from water using MOF matrices, such as electrostatic interactions, acid-base interactions, hydrogen bonding, π-π stacking/interactions, adsorption onto a coordinatively unsaturated site or open-metal site, and hydrophobic interactions [281]. It has also been pointed out that in a particular adsorption process, multiple interactions may take place [282]. In this context, MOF matrices with rich functionality contribute to the realization of several adsorption mechanisms (Figure 9) involving physisorption and chemisorption interactions for effective removal of organic and inorganic pollutants from aqueous media [283].

The implementation of the specific mechanism depends on the specific MOF structure used as an adsorbent and is also defined by a specific organic pollutant or metal ion to be adsorbed. Moreover, the pH value and ionic strength of a water medium contribute to the realization of a specific adsorption mechanism.

Electrostatic interactions are the most frequently observed phenomena during the adsorptive removal of hazardous materials from water. The surface charge is the electric charge present at an interface and on dispersion in polar media such as water. In its turn, the net surface charge of MOF matrices changes depending on the water pH value [281].

H-bonding is regarded as an important mechanism to explain the adsorption of polar organics on functionalized MOFs [12]. For example, the adsorption of artificial sweeteners [2] is driven by H-bonding.

Acid-base interactions. As compared to other interactions, acid-base interactions are rather seldom realized in adsorptive removal of organic pollutants from water [281].

Effect of a metal node in the framework. Some specific metal ions, e.g., Zr^4+^, Fe^3+^, Cu^2+^, Cr^3+^, as nodes in the framework show a particular potential as adsorption sites through complexation with functional groups in organic pollutant molecules [284,285].

π-π Stacking interactions. For aqueous-phase adsorption of aromatic compounds, π-π stacking interactions plays a significant role for MOF adsorbents with linkers based on phenylene polycarboxylates [286] or substituted azolates.

Pore/size-selective adsorption. One of the unique MOF characteristics is the ability to tune their pore size without altering the structural properties that assist in size-selective (or shape-selective) adsorption [281].

Hydrophobic interactions. Hydrophobic compounds are generally nonpolar, poorly soluble in water, and usually have long carbon chains. Hydrophobic interactions are often observed during adsorption of organics from aqueous media [281].

Thus, the information and understanding about the mechanism provide an appropriate choice of a contaminant to be adsorbed from an aquatic medium. In reality, the miscellaneous adsorption mechanisms are implemented while using MOF materials as adsorbents. Besides topology, porous structure, and functionality contributing to the intrinsic adsorption properties of MOFs, there are additional factors, such as recycling/reuse/regeneration and degradation ability of MOF materials and contributing to their potential as advanced adsorbents for water remediation. It is one of the major issues that should be considered to realize their real-world utilization.

Recycling/reuse/regeneration and degradation of MOFs are necessary for industrial applications. The possibility of recycling waste MOFs depends on their chemical and thermal stability, chemical bonding and structure, and the solvents used [287]. The production costs of MOFs depend on various raw materials (e.g., metal salts, linkers, and solvents) and processes (e.g., precipitation, filtration, washing, and drying). Organic linkers are the main cost driver among such consumables. The cost contribution of precursors according to hierarchy is as follows: organic linkers > metal salts > solvents or organic linkers > metal salts for non-aqueous and aqueous synthesis. The high cost of organic solvents could be reduced through recycling of the used solvents.

### 4.1. MOF-Based Composites

Despite apparent advantageous properties of MOFs, most of them have some drawbacks, for example, poor chemical stability which is crucial for the practical utility under industrial conditions [288]. A prospective approach to improve some characteristics of MOF materials, such as mechanical strength, shaping possibility, water sensitivity as well to modify their catalytic and adsorption performance involves combining them with suitable solid supports affording the formation of composites and hybrids [289].

To date, a large variety of MOF-based composites was developed by integrating them with other functional materials, for instance, silica, carbon, organic polymers, nano- and microparticles, supramolecular metallopolymers, enzymes, and biomacromolecules [261,290,291]. Due to a combination of a number of cooperative effects originated from an integration of different kinds of materials, they may find potential applications in gas storage and separation, heterogeneous catalysis, energy storage/conversion, and sensing [289].

In this context, MOF composites with carbonaceous materials are particularly promising because of a diversity of novel functionalities like improved stabilities, template effects etc. in the formed MOF-Carbon composites leading to an enhancement of their functional properties towards specific tasks and thereby expanding their applications fields [292].

### 4.2. MOF-Carbon-Based Composites as Multifunctional and Multipurpose Materials

MOF-Carbon composites can be regarded as the novel class of hierarchically structured matrices that combine the advantages of MOF (well-defined structural and textural characteristics, versatile framework topology and composition, and high porosity) with the intrinsic properties of carbon-based materials, such as high mechanical and elastic strength, excellent chemical and thermal robustness, distinguished electronic and optical properties, low weight, low toxicity and sometimes low cost [293]. These novel functionalities are arising from the synergistic effects (mainly chemical composition, porous system or structure changes) of the two components, namely MOF matrices and carbon matrices. It is a principle to design the advanced MOF-carbon composites [294]. Since most carbon-based materials possess excellent stability towards water/vapor, high temperature, mechanical strength etc., MOF-carbon composites may possess these characteristics. For instance, it was shown that the thermal stability of the GO-MIL-101(Cr) composites increased slightly with increasing GO content, which is similar to the results obtained with UiO-66-GO composites [295].

With respect to their structures, MOF-carbon composites may be divided into two types of materials. The first ones are MOF matrices modified with carbon nanoparticles and the second ones are carbon matrices containing embedded MOF nanoparticles.

### 4.3. Removal of the Toxic Pollutants from Water Using MOF-Based Carbon Composites

Liquid-phase adsorption is an important application of MOF-carbon composites. It has been recognized that the physico-chemical and functional properties of the resulted MOF-Carbon composites associated with liquid adsorption could be enriched, renewed, and enhanced as compared to pristine components by an integration of MOF matrices with carbonaceous materials. To date, there is a number of reports that justify the efficiency of the MOF-carbon composites as environment-friendly adsorbents in the removal of a wide range of pollutants [16].

For instance, the composite GO-MIL-101(Cr) prepared by combining a mesoporous MIL-101(Cr) matrix and GO has shown efficiency in adsorptive denitrogenation of a model fuel containing indole and quinoline due to both favorable mechanisms involving hydrogen bonding between the surface functional groups of GO and the hydrogen attached to the nitrogen atom of indole and appropriate porosity [296].

To date, MOF-carbon composites demonstrated a potential in removal of diverse organic pollutants [286], such as naphthalene [297], tetracycline antibiotics [298], anti-inflammatory drugs (naproxen and ketoprofen) [248], benzoic acid [299], pesticides [300], etc.

Most research reports regarding the removal of organic pollutants from aquatic media on MOF-carbon composites deal with MOF-AC, MOF-GO, and MOF-CNTs systems.

The reusability and regeneration issue are of a prime importance for the practical implementation of the adsorbent. According to relevant literature, the MOF-carbon composites of different types can be easy regenerated by simple washing with organic solvents, as a rule, alcohols, such as methanol or ethanol followed by drying. Then, the regenerated adsorbents of this type can be reused in 5–6 (and probably more) adsorption cycles.

### 4.4. Key Factors, Which Govern Adsorption Performance of MOF-Carbon Composites

Carbon matrix addition in the MOF-carbon composites modify remarkably its surface and porosity in a non-linear way. According to Jhung [281], the adsorbent surface area is one of the most important factors in the removal of organic contaminants, when no special adsorption mechanism, excluding van der Waals interaction, is involved. However, as a rule, an integration of carbon-based materials with MOF components with abundant and diverse functional groups results in enriching interactions between the adsorbate and adsorbent, and thereby in implementation of specific adsorption mechanisms.

### 4.5. Mechanisms Realized Using MOF-Carbon Composites in the Adsorption Processes Related to the Removal of Organic Pollutants from Water

The mechanisms for the adsorption performance of MOF composites may be similar to or different from that of pristine metal-organic frameworks [301]. In this context, it is important to elucidate the difference between these systems related to the adsorption performance based on specific adsorption mechanisms. Simultaneously, it is important to study the synergistic effect of carbon-based materials and MOF matrices on the adsorption performance of the resulted MOF composites in adsorptive removal of organic pollutants from water and answering the question whether the adsorption mechanism can be considered as a sum of adsorption mechanisms realized using separate composite components or it is a particular synergetic mechanism.

As it can be seen from contemporary literature, the MOF-based adsorbents display more diverse adsorption mechanisms than carbon-based materials (even modern ones). Analysis of relevant literature shows that these factors interact with each other. So, it would be interesting to analyze what mechanisms of adsorption of an organic pollutant from water are realized for MOF-Carbon composites and MCs. On the other hand, it is important that an appropriate choice of the MOF structure and changing its content in the composite may result in the change of the adsorption mechanism and realization of a combination of mechanisms.

The adsorption mechanisms implemented for MOF-Carbon composites will be considered for specific materials later on. As to kinetic assessment, most reported experimental results stated that the pseudo-second-order kinetics model could well feature organic pollutant adsorption on MOF-Carbon composites indicating that chemisorption occurred between adsorbates and adsorbents. Moreover, in this case, the adsorption process is referred to a Langmuir isotherm. This phenomenon indicated that chemical adsorption could play an important role in the process of adsorption of organic pollutants on MOF-Carbon composites.

### 4.6. MOF-AC Composites

Currently, MOF-AC composites are extensively studied in organic pollutant removal. Even based on a rather “classical” carbon component, MOF-AC composites can be classified as nano-engineered materials. They show a pronounced efficiency in the organic pollutant adsorption and can be regarded therefore as advanced carbon materials. In particular, MOF-AC systems have been recognized as promising adsorbents for the removal of various dyes, (poly)nitrophenols, and drugs from water [293]. Note, in the case of the MOF-AC composites, they are commonly formed as carbon matrices containing embedded MOF nanoparticles.

An example of the enriched adsorption mechanism realized due to a combination of MOFs and carbon materials is an investigation of the HKUST-1-AC composite prepared by ultrasonically assisted hydrothermal method starting from HKUST-1 (Cu_3_(btc)_2_) metal-organic framework and AC as components [302]. Figure 10 illustrates the preparation procedure for this material.

The AC composite was investigated in the ultrasound-assisted adsorption of dyes in a ternary solution. The Q_m_ values were 133.33 mg/g, 129.87 mg/g, and 65.37 mg/g for CV, DB, QY respectively. These values were remarkably higher than Q_0_, i.e., 59.45 (CV), 57.14 (DB), and 38.80 mg/g (QY) for the reference HKUST-1 material.

The adsorptive removal of studied dyes is dependent on the solution pH value. The maximum adsorption efficiency (R, %) for QY was observed at pH = 3. At higher pH, a hydroxide ion and anionic form of dyes mainly compete with each other for adsorption onto reactive sites of the adsorbent. Based on adsorption measurements, it was suggested that an enhanced adsorption was based on the contribution of various mechanisms, i.e., ion exchange, hydrogen bonding, electrostatic, soft-soft, and dipole-ion interactions.

These authors have used this ultrasonic-assisted way for the preparation of the MOF-5-AC composite using MOF-5 framework (Zn_4_O(bdc)_3_) and AC as components [303] following to similar ultrasound-assisted solvothermal methodology developed for the HKUST-1-AC system [302].

Note that the pristine MOF-5 matrix is an archetypical MOF structure with a high surface area—up to ~3000 m^2^/g and porosity [304]. However, its weak point is a low stability under humid conditions and moderate thermal stability (till ~330 °C). This hinders a practical utilization of the MOF-5 material.

The produced MOF-5-AC material was studied in the ultrasonic-assisted simultaneous removal of FG, EY, and QY from aqueous media. The structures of the studied dyes are given in Figure 11. Q_m_ values obtained for the composite were 21.230, 20.242, and 18.621 mg/g, for FG, EY, and QY, respectively, while the maximum capacities for the pristine MOF-5 matrix were obtained to be 8.5, 5.7, and 4.8 mg/g, respectively. So, the synergetic effect of the MOF-5 and AC components was found in the composite.

Highly dispersed ZIF-8 nanoparticles were crystallized onto porous Zn-containing and nitrogen-enriched carbons derived from adipic acid by an in situ growth method affording ZIF-8-C composites [305]. The surface area and pore volume of these materials are controlled efficiently by changing the Zn content in the carbon matrix. ZIF-8-C composites were studied in the malachite green adsorptive removal from water solutions.

It was found that the adsorption values were determined by the ZIF-8 content in the composite. The highest adsorption capacity (q_e_)—431.48 mg/g—was achieved using the 14-ZIF-8-C system with the maximal ZIF-8 content. This value is higher than the same parameters measured for pristine porous carbon and pure ZIF-8 matrix. The Q_m_ value as high as 3056 mg/g (calculated from the Langmuir model) was obtained for the 14-ZIF-8-C composite at 30 °C.

The adsorption process can be considered as a combination of physisorption and chemisorption. Note, the adsorption of MG by the pristine ZIF-8 matrix is mainly dependent on the electrostatic attraction, π-π stacking interactions, and hydrogen bonding. In particular, both MG and ZIF-8 components have aromatic rings, so, strong π-π stacking interaction between them may be implemented. On the other hand, nitrogen atoms in the 2-methylimidazolate linkers in the ZIF-8 framework and MG molecule can interact via hydrogen bonding [306]. These mechanisms are suggested for the adsorption performance of the ZIF-8-C composite.

An AC-NH_2_-MIL-101(Cr) nanocomposite was synthesized starting from activated carbon AC and the NH_2_-MIL-101(Cr) matrix by a one-pot approach under hydrothermal conditions [307]. In this material, AC particles were incorporated into the pores of the NH_2_-MIL-101(Cr) matrix. The synthesized AC-NH_2_-MIL-101(Cr) composite was studied in the removal of p-nitrophenol (PNP) from water.

In the adsorption experiments, the AC-NH_2_-MIL-101 composite shows an adsorption capacity of ~18.3 mg/g. The removal capacities follow the order: AC-NH_2_-MIL-101(Cr) composite > activated carbon > NH_2_-MIL-101(Cr). For this composite, a number of adsorbent-adsorbate interactions were realized. So, the enriched mechanism for enhanced adsorption performance is related to coordination of NO_2_ groups of PNP and the OMSs (Cr^3+^ ions in the NH_2_-MIL-101(Cr) framework) of the AC-NH_2_-MIL-101(Cr) material, electrostatic interactions between NO_2_ groups of PNP and Cr^3+^ ions, the hydrogen bonding between the functional groups of PNP and AC-NH_2_-MIL-101(Cr) and π-π stacking interaction between aromatic moieties of PNP and NH_2_-MIL-101(Cr) (Figure 12).

Moreover, AC-NH_2_-MIL-101(Cr) shows good stability and reusability in five cycles of the PNP adsorption process.

### 4.7. MOF-GO Composites

GO is an advanced adsorbent for water purification (see previous sections). Further enhancing its adsorption performance is achieved by tuning porosity or dispersion on a suitable support [308]. An efficient way to improve the adsorption performance of GNs is a combination of GO with MOF matrices. In this context, the syntheses of nanocomposites based on MOF and GO is a growing area of research for the development of new nano-engineered materials with different potential applications.

GO is promising for the preparation of composites with MOF materials due to its intrinsic porous structure and high surface area, and diverse ample functional groups, such as carboxyl, epoxy, and hydroxyl groups on its surface [160,285]. In recent years, GO-MOF composites, such as GO-ZIF-8 [21], GO-ZIF-67 [22], GO-MOF-5 [23], GO-HKUST-1 [24], GO-MIL-100 [26], and GO-MIL-53 [27] were produced. They combine the intrinsic properties of both GO layers and MOF crystals [309,310] and find applications in many fields, such as gas separation and storage [311,312], as well as water purification [313]. In particular, MOF-GO composites have shown efficiency in the adsorptive removal of organic pollutants from water [198].

Along with other types of composites, GO-MOF systems are synthesized using a one-step “bottle-around-ship” approach under different conditions [314], for instance, using a sonochemical technique [301]. Figure 13 presents a few examples of this synthesis route for some most studied MOF structures. In the context of a high cost of GO and MOF components, significant efforts are directed towards a decrease of the production cost of the MOF-GO composites [315]. In particular, GO is often used in the composite synthesis as a modifier in the form of particles dispersed on the MOF support.

#### 4.7.1. Electrostatic Interaction, π-π Stacking Interaction, Hydrogen Bonding and Open Metal Sites Complexation

It can be seen from contemporary literature that the metal-organic frameworks belonging to the MIL family are promising adsorbents for organic pollutant removal in a pristine form and as components of composites with carbonaceous materials [270]. MIL frameworks with M^3+^ ions are composed of polycarboxylate linkers and have a pore size that differed in the micro/meso range and diverse topology. Several MIL structures, such as MIL-101(Cr) and MIL-100(Fe), have unsaturated coordination sites or open metal sites (as potential adsorption centers) as inorganic nodes in the framework. The particular physico-chemical characteristics make MIL matrices effective for water remediation via adsorption.

An impact of different GO contents (2, 5, 10, 15, 20 wt.%) in the synthesized GO-MIL-101(Fe) composites with a core-shell structure on their adsorption performance in the removal of MO dye was studied [316]. The introduction of GO particles modified the pore volume and specific surface area in the resulted composites as compared to the pristine MIL-101(Fe) matrix (Fe_3_O(bdc)_3_). Moreover, the thickness of the MIL-101(Fe) shell could be adjusted through ultrasonication [285]. It was found that the Q_m_ values were a function of the GO wt.% content in the composite. The optimal adsorption performance was achieved using the 10% GO-MIL-101(Fe) composite. The Q_m_ value for this system was as high as 186.20 mg/g, while the same parameter measured for the pristine MIL-101(Fe) material was 117.74 mg/g. Further increasing the GO content in the composites (15 and 20 wt.%) resulted in a gradual decrease of the adsorption value, due to pore blocking and shielding the active adsorption sites on the surface.

The proposed adsorption mechanism involved electrostatic interactions between the positive charge on the surface of the GO-MIL-101(Fe) material and the negative charge of the sulfonic acid group of MO, π-π interactions of MO with the composite surface, hydrogen bonding of MO with functional groups (OH and COOH) of GO as well as carboxylic groups of the bdc linkers, and coordination of electronegative MO molecules with open metal sites (Fe^3+^) as inorganic nodes in the MIL-101(Fe) structure, so synergetic adsorption performance for the GO-MIL-101(Fe) composite was observed, which was enhanced as compared to adsorption properties of the pristine MIL-101(Fe) material.

The same strategy was used for the preparation of GO-MIL-101(Cr) starting from the MIL-101(Cr) matrix with the same topology and open metal sites (Cr^3+^) similarly to its isostructural MIL-101(Fe) analog. GO-MOF composites (GO-MIL-101(Cr)) with different GO contents were prepared starting from MIL-101(Cr) and GO components via one-pot synthesis strategy [317]. The porosity of the GO-MIL-101(Cr) composite was increased by deposition of GO particles onto the MIL-101(Cr) matrix up to a certain ratio. The S_sp_ (BET) of GO-MIL-101(Cr) (3%) was about 6.3% larger than that of pristine MIL-101(Cr). Note, the concentrations of acidic functional groups contributing to the adsorption, changed in the order GO > GO(3%)-MIL-101(Cr) ≫ MIL-101. It should be noted that the number of phenolic groups in the GO(3%)-MIL-101(Cr) composite was higher than those of other functional groups.

The synthesized GO-MIL-101(Cr) composites were studied in the adsorption of AIDs, i.e., NAP and KET from water. These materials show improved adsorption performances toward both NAP and KTP relative to the pristine MIL-101(Cr) matrix, pure GO sample, and commercial AC. The GO(3%)-MIL-101(Cr) composite shows a higher Q_m_ values for NAP (2.1 and 1.5 times) than those for commercial AC [318] and pristine MIL-101 matrix [319], respectively, at pH 5.4. The adsorption performances for both NAP and KTP decreased in the order: GO(3%)-MIL-101(Cr) composite > MIL-101 > AC > GO.

Sarker et al. [317] concluded that the enhanced performance of the GO(3%)-MIL-101(Cr) composite in the adsorption of AIDs over a wide range of pH was assisted by an increase of the surface area and introduction of additional ample functional groups due to GO particles. These hydroxyl and carboxylic groups along with carboxylic groups of MIL-101(Cr) contribute to the hydrogen bonding (H-bond donor: MIL-101-GO (3%); H-bond acceptor: AIDs), which is a dominating adsorption mechanism. Note that the adsorption mechanism realized using the pristine MIL-101(Cr) matrix involves electrostatic interactions, π-π-stacking interactions, complexation via unsaturated coordination sites (open Cr^3+^ sites). Therefore, a synergistic effect between MIL-101 and GO in the composite was observed for the AIDs adsorption.

MIL-68 type MOFs are representative structures to illustrate the diversity of possible adsorption mechanisms realized in their composites with GO in the adsorption of organic pollutants. The MIL-68(Al) matrix is assembled from the infinite straight chains of corner-sharing metal-centered octahedral AlO_4_(OH)_2_ units that are connected through hydroxyl groups and terephthalate ligands [320]. The adsorption performance of the MIL-68(Al) material can be further improved by integration with some carbonaceous adsorbents, such as GO.

The MIL-68(Al) material was used for the preparation of the composite with GO (MIL-68(Al)-GO) [321]. Compared to the parent material, this composite has a higher total pore volume and surface area. The MIL-68(Al)-GO system has a sandwich construction with retained structures of the MIL-68(Al) framework and GO.

This composite was studied in the adsorption of MO from water over a wide pH range and showed the maximum q_e_ ~400 mg/g. The adsorption of the planar MO molecule on the MIL-68(Al)-GO material could be realized by π-π stacking interactions due to benzene rings in MO, organic linkers in the MIL-68(Al) structure, and aromatic moiety of GO. Additionally, MIL-68(Al) has been proved to contain μ_2_-OH groups in Al–O–Al units, which can form hydrogen bonds with MO nitrogen or oxygen atoms [322]. So, the adsorption mechanism for the MIL-68(Al) –GO system involves multiple simultaneous interactions, such as electrostatic interactions, π-π-stacking, and hydrogen bonding.

The same mechanism diversity was also realized using NH_2_-MIL-68(Al)—aminomodified analog of the MIL-68(Al) structure—as the component of the composite with GO, RGO-NH_2_-MIL-68(Al) [198]. This RGO-NH_2_-MIL-68(Al) composite was synthesized via a one-pot solvothermal process. It was studied for the adsorptive removal of cationic dyes, i.e., CR from single and binary water solutions.

The characterization results show that the surface area of the RGO-NH_2_-MIL-68(Al) material is significantly enhanced by the introduction of rGO. It was shown that the maximum CR adsorption capacity of RGO-NH_2_-MIL-68(Al) (473.93 mg/g) is 121% and 1337% higher than those measured for the pristine materials, NH_2_-MIL-68(Al) and RGO, respectively. The mass transfer of CR onto the RGO-NH_2_-MIL-68(Al) adsorbent is controlled by the film-diffusion and intra-particle diffusion. The proposed adsorption mechanism for this composite is based on the three-dimensional excitation-emission matrix (3D-EEM) fluorescence spectroscopy and involves the electrostatic interaction (attraction), π-π stacking interaction and hydrogen bonding. So, the synergistic effect in respect to adsorption performance between composite components was shown.

Using another structure belonging to the MIL-68 family, i.e., NH_2_-MIL-68(In) as the component, NH_2_-MIL-68(In)-GO composites (In-MOF@GO) based on GO were prepared via a one-step solvothermal route [323]. The NH_2_-MIL-68(In) metal-organic framework is built from infinite chains of corner-sharing InO_4_(OH)_2_ octahedra linked via 2-aminobenzene-1,4-dicarboxylate linkers [324].

The S_sp_ (BET), V_total_ and V_micro_ values of the composites with a low GO content (i.e., In-MOF@GO-1 and In-MOF@GO-2) are higher than those of the parent MIL-68(In)-NH_2_ matrix. The In-MOF@GO-2 sample had the maximum surface area (679.5 m^2^/g) and total pore volume (0.389 cm^3^/g). A gain in the composite porosity may be explained by additional pores created at the interface between GO particles and NH_2_-MIL-68(In) matrix. However, for the In-MOF@GO-3 composite with the highest GO content, a decrease in all these parameters was observed due to pore blocking in the NH_2_-MIL-68(In) host.

In-MOF@GO composites were studied in the adsorption of rhodamine B from aqueous solutions (Table 3). The optimal adsorption activity was achieved at pH = 6. The adsorption performance of the NH_2_-MIL-68(In)-GO composite was significantly better than that of the parent NH_2_-MIL-68(In) material, which confirmed the synergy between the composite adsorbent components.

Liang et al. [324] suggested that the π-π stacking interactions between GO and RhB assisted in the dye adsorption. Additionally, the negative charges in the GO sheets caused by oxygen-containing functional groups contributed to the additional electrostatic interactions with cationic RhB molecules.

The separation of MOF-C adsorbents in a powder form is a rather hard task preventing their practical application. To conquer this circumstance, some shaping processes conducted by using an extrusion or a pressing method have been explored [325]. However, the pellets or granules produced in this way are sometimes unstable in water, which may result in a decrease of the adsorption capacity.

The organic pollutant adsorption using MOF-GO systems more technologically was studied [326]. Following this strategy, the MIL-68(Al)-GO composites with a different MIL-68(Al): GO mass ratio (10.00, 6.25, 4.17, and 3.33) were prepared in the pellet form by adding a natural cross-linking agent and further shaping in a calcium chloride solution.

The pelletized MIL-68(Al)-GO composite (MIL-68(Al): GO mass ratio = 10) was studied as an adsorbent for aqueous TC removal. Its performance was evaluated by comparison with the parent powder. It was concluded that the mesoporous structure of the MIL-68(Al)-GO composite formed during pelletization contributes significantly to the adsorption performance.

The surface area and total pore volume of the MIL-68(Al)-GO pellets decreased due to the disappearance of micropores. Actually, due to crosslinking with sodium alginate, the micropores in the MIL-68(Al)-GO material were greatly reduced, while additional mesopores were generated during the pelletization process. TC molecules with a diameter of ~12.0–13.0 Å are not easily transferred into the micropores of the powder MIL-68(Al)-GO. In contrast, they could enter into the mesopores of pellets with diameters of 80–150 Å.

The material shows a maximum adsorption capacity of 228 mg/g. The adsorption mechanism for the pellets may be ascribed to the complex interactions of hydrogen bonding, π-π stacking as well as Al-N covalent bonding. In particular, Al-N covalent bonding is realized for Al^3+^ ions in the MIL-68(Al) framework and N-moiety in the TC molecules (Figure 14).

In context of practical applications, a high stability in a wide pH range and reusability after adsorption completion of the MIL-68(Al)-GO pellets was demonstrated.

An example of the utilization of the preliminarily functionalized rGO as a component for the preparation of the composite with a MOF matrix is reported [327]. In this work, the rGO-MIL-68(Al) composites with a different rGO loading were prepared starting from the MIL-68(Al) material and rhamnolipid-functionalized graphene oxide by a one-step solvothermal method starting from the MIL-68(Al) material.

The adsorption performances of the synthesized rGO-MIL-68(Al) composites were studied for the *p*-nitrophenol removal from aqueous solutions. Interesting to note, the PNP adsorption capacities of the rGO-MIL-68(Al) materials changed along with their surface areas, i.e., they first increased and then decreased with increasing the rGO content. The Q_m_ for the composite with the optimal content, i.e., the rGO(15%)-MIL-68(Al) material prepared at the RGO-to-MIL-68(Al) mass ratio of 15%, calculated from the Langmuir model is as high as 332.23 mg/g at 303 K. This value is significantly higher than the values measured for the pristine MIL-68(Al) and rGO components. Furthermore, the PNP uptake rate for this composite is 64% and 123% higher than that for the starting materials, respectively.

A series of the studies was performed using highly stable isoreticular UiO-66,67(Zr) matrices based on Zr_6_O_4_(OH)_4_ clusters. Note, these metal-organic frameworks have Zr4+ open metal sites as inorganic nodes. For instance, GO particles were used for the modification of the adsorption performance of UiO-66(Zr)-type materials functionalized with hydroxyl and carboxylic groups [315]. The GO-UiO-66-(OH)_2_ composite was prepared using the hydroxyl-modified UiO-66-(OH)_2_ matrix and guest GO particles via a hydrothermal method. For reducing the production cost of the adsorbent, the synthesis strategy involves a small amount of GO. The GO introduction improved further the stability of the GO-UiO-66-(OH)_2_ material. Compared with the pristine UiO-66-(OH)_2_ matrix, the addition of GO particles increased the content of functional groups in the resulted system for more rich and stronger composite-adsorbate interactions.

The GO-UiO-66-(OH)_2_ nanocomposite was studied in the adsorptive removal of MB and TCH from water. This adsorbent has shown the adsorption efficiency as high as 99.96% (MB) and 94.88% (TC), respectively.

The high surface area and porosity of the UiO-66-(OH)_2_ matrix assist to the diffusion of MB from the solution to the outer surface of the GO-UiO-66-(OH)_2_ matrix as well as pores and an inner surface. Moreover, the enhanced adsorption of MB on the GO-UiO-66-(OH)_2_ material may be related to the strong chemical bonding between the metal and the MB molecule improving the adsorption efficiency. In particular, abundant open metal sites (Zr^4+^) in the UiO-66-(OH)_2_ matrix facilitate the adsorption of guest molecules with corresponding functional groups. The aromatic ring of the MB molecule can interact with the benzene ring in the organic moiety of the GO-UiO-66-(OH)_2_ composite via π-π interaction. The epoxy and hydroxyl groups in GO can interact with MB functional groups, e.g., -CH_3_ as H acceptor via hydrogen bonding. So, the modification of the UiO-66(Zr) matrix with hydroxyl groups and GO particles and thereby introducing more active adsorption sites improves the removal efficiency for MB and TC. Accordingly, the electrostatic interactions, π-π stacking interactions, and hydrogen bonding play a key role in the adsorption of pollutants on the GO-UiO-66-(OH)_2_ composite.

One more UiO-66(Zr) framework modified with carboxylic groups was used for producing the GO-UiO-66-(COOH)_2_ composite (Figure 15) [328]. This material was used for TC adsorption. The Q_m_ of UiO-66-(COOH)_2_-GO was as high as 164.91 mg/g, while the pristine UiO-66 matrix has shown the Q_m_ around 27.53 mg/g. As in the case of the GO-UiO-66-(OH)_2_ composite, the enhanced adsorption performance of the GO-UiO-66-(COOH)_2_ system is predominantly attributed to increased concentration of adsorption active sites caused by the introduction of GO particles and carboxyl groups in the UiO-66-(COOH)_2_ matrix.

Note, the adsorption value for the GO-UiO-66-(COOH)_2_ system for TC was higher than in the case of the GO-UiO-66-(OH)_2_ composite, so modification of the UiO-66 matrix with carboxylic groups was more efficient than in the case of modification with hydroxyl groups (37.96).

The TC molecule has electron-rich –NH_2_ groups, and therefore it interacts as the electron donor with the benzene ring of UiO-66-(COOH)_2_ via π-π stacking interactions. TC may serve as π-electron acceptor as a result of the presence of the electron-donating group of ketones in TC molecules facilitating the π-π interaction between the benzene ring of TC molecules and the organic ligand of UiO-66-(COOH)_2_.

Accordingly, the proposed adsorption mechanism involves π-π interaction, chemical coordination due to open metal sites (Zr^4+^), and weak electrostatic interaction between UiO-66(COOH)_2_-GO and TC.

The second type of composite was prepared starting from graphene oxide and other zirconium-based UiO-67 metal-organic framework affording a GO-UiO-67 nanocomposite via the solvothermal procedure [329]. The UiO-67 matrix has the same topology as UiO-66, but with larger pores, due to extended 4,4′-biphenyldicarboxylate (bpdc) linkers. The produced composite formed as UiO-67 nanoparticles on the surface of GO sheets.

The UiO-67-GO nanocomposite was studied in the adsorption of organophosphorus pesticides from aqueous solutions. It was demonstrated that abundant Zr-OH or µ_3_-OH groups in inorganic nodes of the UiO-67 nanoparticles play a vital role in capturing glyphosate molecules [330].

The adsorption capacity of glyphosate on the UiO-67-GO composite was as high as 2.855 mmol/g (482.69 mg/g) at pH = 4, which surpasses that of most GO-based adsorbents. The enhanced adsorption performance of this nanomaterial is related to abundant hydroxyl and carboxylic groups in the GO matrix as well as Zr-OH units in the UiO-67-GO structure. In particular, due to the functional groups of the UiO-67 metal-organic framework, which may act as binding sites for phosphorus on the surface and in the volume of the UiO-67-GO nanocomposite, this system shows enhanced adsorption performance for glyphosate compared to the pristine GO matrix. Therefore, it was suggested that the dominant mechanism of glyphosate adsorption suggested from FTIR and XPS characterization was a surface/inner-complexation with functional groups of the UiO-67-GO material (Figure 16).

In the acidic region, the higher concentration of H^+^ enhances the electrostatic interactions between the surface of the UiO-67-GO nanocomposite and the phosphate group of glyphosates, which contribute to the adsorption process.

Preparation of a composite aerogel based on MOF crystals and carbonaceous nanomaterial may result in significant improvement of their adsorption performance for their application in water treatment [277]. For instance, the porous 3D ZIF-67-rGO composite in an aerogel form was prepared via the assembly of ZIF-67 crystallites with a polyhedron shape on the 3D rGO framework [331]. The synthesis procedure for this material is presented in Figure 17. The resulted ZIF-67-rGO composite was studied in the absorption of cationic CV and anionic MO dyes.

The synthesized composite shows the maximum adsorption capacities up to 1714.2 mg/g and 426.3 mg/g for CV and MO, respectively. In particular, loading ZIF-67 on 3D rGO aerogel increases the adsorption capacity of 3D ZIF-67-rGO aerogel to MO. Accordingly, the adsorption capacities towards MO are 119.2 mg/g, 403.3 mg/g, and 417.1 mg/g for the 3D rGO aerogel, ZIF-67 matrix, and 3D ZIF-67-rGO aerogel, respectively. Analogous adsorption improvement was observed for the removal of CV dye.

These enhanced adsorption values towards both cationic dyes compared to the pristine ZIF-67 material and rGO aerogel can be ascribed to π-π interactions and electrostatic interactions between 3D rGO-ZIF-67 aerogel and dye molecules (Figure 18). So, the synergistic interactions of 3D rGO and ZIF-67 crystallites in the rGO-ZIF-67 material with multi-affinity properties could be concluded.

ZIF-GA were synthesized via a counter-diffusion method using graphene sheets and isostructural ZIF-8 and ZIF-67 frameworks with meIm linkers and Zn^2+^ and Co^2+^ ions [332]. ZIF-67-GA and ZIF-8-GA composites have been prepared by a two-step synthesis process. The synthesized hybrid aerogels are organized as a 3D interconnected macroporous framework of Gn sheets decorated with ZIF particles with a uniform dispersion.

The adsorption performance of the ZIF-67-GA and ZIF-8-GA materials was investigated in the removal of dyes from water. Due to cooperative effects between ZIF and GA, the ZIF-GA composites demonstrated an enhanced adsorption performance towards studied dyes, especially for the anionic ones. The ZIF-67-GA nanocomposite had a higher adsorption efficiency than the ZIF-8-GA material and pristine graphene aerogels towards dyes. In particular, the adsorption capacities of ZIF-67-GA were 550.3 and 380.7 mg/g towards MO and RhB, respectively. An enhancement in the retention of anionic dyes in the ZIF-GA systems may be assigned to the strong electrostatic attraction. The differences in the adsorption performance of ZIF-GA towards anionic MO dye and cationic RhB dye are related to the different electrostatic attraction and π-π interaction of ZIF-GA composites to dyes. Note, for different dyes, π-π stacking interactions between graphene sheets and dye molecules may be different [333]. The different size of mesopores in the ZIF materials along with a different size and amount of embedded ZIF particles may also contribute to the adsorption peculiarities.

The –NH moiety in amide-functionalized MOF structures can act as an electron acceptor, whereas the -CO group can act as an electron donor [334]. For instance, amide-functionalized MOF TMU-23 ([Zn_2_(oba)_2_(bpfb)]·(DMF)_5_) was utilized for the synthesis of its composite with GO, i.e., the TMU-23-GO material [335].

This composite was synthesized in a nanoplate form using an ultrasound irradiation technique (Figure 19). In TMU-23-GO synthesis, GO acts as a template to capture Zn^2+^ cations and facilitate the crystal growth on its surface. The GO-TMU-23 nanocomposite exhibits outstanding adsorption properties for the adsorptive removal of MB from water. It exhibits a higher adsorption efficiency (~90%), than pristine GO and TMU-23 materials. In particular, the adsorption rate for MB was much higher using the TMU-23-GO nanocomposite than using pristine TMU-23 and GO components. This MB adsorption efficiency is related to the dominating mechanism based on strong electrostatic or acid-base interactions between the cationic dye and amide groups of the TMU-23 component. Additionally, π-π stacking interactions between the aromatic rings in the MB molecule and TMU-23 as well as physical adsorption of MB molecules on the GO sheet surface contribute also to the adsorption performance of the GO-TMU-23 nanocomposite.

#### 4.7.2. Lewis Acid-Base Interactions: Role of Metal Centers in the Framework

According to ultrasonic ball milling strategy, the HKUST-1(Ni)-GO composite was synthesized starting from the HKUST-1(Ni) material and GO using a one-step in situ mechanochemical method. In the composite structure, the HKUST-1(Ni) nanoparticles were embedded in the GO layers [336]. The mesoporous HKUST-1(Ni)-GO material has a bit higher S_sp_ (BET) value of 69.6 m^2^/g as compared with the pristine mesoporous HKUST-1(Ni) matrix (59.8 m^2^/g). This slight gain in the surface area could be attributed to the new pores created at the interface between HKUST-1(Ni) crystallites and GO.

The adsorption performance of the HKUST-1(Ni)-GO nanocomposite was studied in the Congo red removal from water. The presence of functional groups like carboxylic and hydroxyl groups of GO increased the number of adsorption sites. In particular, cationic dye molecules at acidic pH may be attached to the surface of the HKUST-1(Ni)-GO composite replacing H^+^ ions of carboxylic and hydroxyl groups. Moreover, Ni^2+^ ions as Lewis acid sites in the HKUST-1(Ni) structure contribute to the acid-base interactions and thereby improve the adsorption value.

Note, a suitable temperature for composite drying (over 100 °C) may result in an enhancement of the adsorption capacity of CR over the HKUST-1(Ni)-GO composite. The reason for this adsorption improvement is the removal of coordinated water molecules shielding Ni^2+^ metal sites in the HKUST-1(Ni) framework, so the acid-base interaction between active metal sites (Lewis acid) and –NH_2_ (Lewis base) in CR is implemented in a much stronger way, so the adsorption capacity of the HKUST-1(Ni)-GO composite for CR was ~2489 mg/g, which was higher than the value measured for the pristine HKUST-1(Ni) material (2046 mg/g) and a number of known adsorbents.

#### 4.7.3. Multi-Component Systems Including GO, MOF Crystallites, and Other Components

Hydrogen bonding, hydrophobic interactions, electrostatic interactions, and π-π stacking/interactions. It is interesting to note that GO may serve as a component of more sophisticated three-component composites. For instance, a potential GO-hydrogel for MOF-based composite preparation and exploitation as a three-component-system in TC adsorptive removal from water was studied by Kong et al. [337]. In this work, an alginate hydrogel-GO hybrid material was used as a substrate for ZIF-67 nanoparticles. In the first step of the composite synthesis, an alginate-graphene matrix with rich oxygen-containing functional groups was formed into a hydrogel (AG-Co) through Co^2+^ coordination. The Co^2+^ ions in the AG-Co matrix acted as metal nucleation sites for the in situ growth of ZIF-67 crystals to form alginate-graphene-ZIF-67 composite (AG-ZIF-67).

This strategy allowed one to obtain highly dispersed ZIF-67 particles on alginate-Gn hydrogel for the further adsorption enhancement. The alginate also prevented GO from aggregation in TC removal. In addition, the AG-ZIF-67 material has a three-dimensional aerogel structure cross-linked through sodium alginate and has some wrinkles and defects on the surface. These defects assist to additional contact sites than the smooth surface of the Co-AG precursor in the process of organic pollutant adsorption.

For TC removal, the Q_m_ for AG-Co and AG-ZIF were 105.49 and 456.62 mg/g, respectively. The adsorption sites on the surface of AG-ZIF was nonuniform with multilayer adsorption characteristics due to the different adsorption sites in the AG-ZIF composite, i.e., ZIF crystals and oxygen-containing functional groups of alginic acid and GO. Due to abundant functional groups, the tetracycline adsorption on the AG-ZIF material was administered by the π-π interaction and the cation-π bonding.

In several cases, addition of third or fourth components to the MOF-GO composites can enrich the adsorption mechanism. An example is a preparation of a GO-CD composite (GO-β-CD)), which was decorated by iron oxide affording a Fe_4_O_3_-GO-β-CD system, then HKUST-1(Cu) nanoparticles were embedded in the Fe_4_O_3_-GO-β-CD matrix [338]. The preparation procedure for this material is presented in Figure 20. The characterization results indicated that the Fe_4_O_3_-GO-β-CD system consisted of a thin single layer containing Fe_4_O_3_. The HKUST-1 material was coated on the Fe_4_O_3_-GO-β-CD surface. The obtained HKUST-1-Fe_4_O_3_-GO-β-CD material has shown super-paramagnetism with the magnetization value of 10.47 emu/g and a moderate S_sp_ (BET) value of 250.33 m^2^/g.

The HKUST-1-Fe_4_O_3_-GO-β-CD composite was used for adsorption purification of neonicotinoid pesticide spiked tap water samples. It shows pesticide adsorption uptakes, i.e., imidacloprid (3.11 mg/g), thiamethoxam (2.88 mg/g), acetamiprid (2.96 mg/g), nitenpyram (2.56 mg/g), dinotefuran (1.77 mg/g), and thiacloprid (2.88 mg/g). Its adsorption capacity and rate for neonicotinoid insecticides are facilitated by the presence of hydrophobic inner cavities and supramolecular recognition of Fe_4_O_3_-GO-β-CD.

The studied neonicotinoid insecticides have nitrogen-containing groups, hydrophobic groups, and delocalized π-bonds due to their benzene rings or five-membered heterocycles. Therefore, the adsorption performance of the HKUST-1-Fe_4_O_3_-GO-β-CD composite may be implemented via interactions between functional groups of neonicotinoid insecticides and the HKUST-1 framework. So, due to multicomponent and multifunctional properties of the composite system, the multiple cooperative effects are realized using this adsorbent.

The three-component composite adsorbents, Fe_3_O_4_-Ni_3_(BTC)_2_@GO and Fe_3_O_4_-Co_3_(BTC)_2_@GO were produced using other HKUST-1 materials, i.e., HKUST-1(Ni) (Ni_3_(BTC)_2_) and HKUST-1(Co) (Co_3_(BTC)_2_) metal-organic frameworks as components with a one-step solvothermal method [339]. The proposed design strategy involves decoration of GO using embedded HKUST-1(Ni)/HKUST-1(Co) nanoparticles and Fe_3_O_4_ magnetic nanoparticles. The preparation procedure for these materials is presented in Figure 21. According to TEM results, the 2D structure of GO serves as a support for both HKUST-1(Ni)/HKUST-1(Co) and Fe_3_O_4_ nanoparticles. No Fe_3_O_4_ or HKUST-1 nanoparticles are observed outside the Gn sheets.

Kong et al. [339] suggested that HKUST-1(Ni)/HKUST-1(Co) nanoparticles were formed and grown over layers of GO through chemical bonding between functional groups of GO and metal-organic frameworks. This formation type prevents stacking of the GO layers providing thereby a higher surface available for the adsorption.

The prepared composite nanoadsorbents were studied in the removal of the MB dye and metal ions or anions (Na^+^, Ca^2+^, Mg^2+^, SO_4_^2−^, SiO_3_^2−^) from a brackish water model. The HKUST-1(Ni) composite shows the MB adsorption concentration of 54 mg/g, while the HKUST-1(Co) material demonstrates the adsorption concentration of 67 mg/g with an initial concentration of 100 mg/g. Important to note, composite nanoadsorbents show a higher MB uptake, when compared to the pristine HKUST-1 and GO materials.

This adsorption enhancement is attributed to highly dispersed MOF nanoparticles in the GO matrix, and to preventing of stacking of the graphene layers by the MOF nanoparticles and providing more available adsorption sites. So, the synergistic effects of three different components in the composite provides an efficient adsorption of MB.

### 4.8. MOF-CNT Composites

The next type of functional MOF-carbon based composites is obtained by integration of CNTs and MOF crystallites. CNTs feature an attractive combination of intrinsic structural, chemical, textural, mechanical, and hydrophobic properties [99], which provides their potential as a platform for the development and preparation of high-performance composites with other materials including MOF matrices [285]. These systems find a wide range of applications, especially, acting as adsorbents for water remediation.

#### 4.8.1. π-π Stacking Interactions and Hydrogen Bonding

The MIL-68(Al) MOF was used for the synthesis of a series of MWCNT@MIL-68(Al) composites with different carbon nanotubes loadings changing in the range 0.15–15 wt.% (the composite type 1) [286]. This work shows that the incorporation of MWCNTs in small quantities (below 0.45 wt.%) can improve textural properties of MIL-68(Al). For instance, the introduction of MWCNTs generates a large amount of small micropores in the MIL-68(Al) framework and, therefore, increases the microporosity of its composites with MWCNTs.

Compared with the pristine MIL-68(Al) material, highly dispersed MIL-68(Al) nanocrystals containing MWCNT particles with a homogeneous distribution form MWCNT@MIL-68(Al) composites. From IR spectroscopy study, it was suggested that the functionalized CNT particles coordinate to the framework of MIL-68(Al) through –COOH groups.

The MWCNT@MIL-68(Al) composites were studied in the phenol removal from water. It was shown that the adsorption capacity for phenol from aqueous solutions was up to 341.1 mg/g at the initial concentration of 2000 ppm for the MIL-68(Al) matrix modified with 0.75 wt.% MWCNTs. According to adsorption tests, a removal capacity of phenol on CNT@MIL-68(Al) composites is a function of the MWCNT content. Interesting to note, introducing appropriate amounts of MWCNTs in the MIL-68(Al) matrix increases both the BET surface areas and the total volume of small micropores (<16 E), thereby leading to enhanced adsorption performance of the CNT@MIL-68(Al) composites. On the other hand, the optimal MWCNT loading corresponding to the largest adsorption capacity is dependent on the initial concentration of phenol in the solution.

In particular, the 3.5% CNT@MIL-68(Al) system with the largest cumulative pore volume in the range of pore width less than 1.6 nm exhibits the best performance. Its adsorption capacity is 109.9 mg/g, which is 7.4 times higher than that of the pristine MIL-68(Al) material (13.1 mg/g).

Regarding the industrial concentration range of phenol, the adsorption capacity was up to 341.1 mg/g at the initial concentration of 2000 ppm for the MIL-68(Al) matrix modified with 0.75 wt.% MWCNTs (75% CNT@MIL-68(Al) composite) or up to 188.7% compared with the pristine MIL-68(Al) matrix. This improvement can be explained by the formation of a large amount of small micropores by incorporation of functional CNTs particles. In these small pores, the π-π interactions between the aromatic rings of phenol and the composites can be enhanced to induce the relatively high adsorption capacity, together with the hydrogen bonds between –OH in phenol and –COO^−^ in the adsorbent [340]. These characterization and adsorption results demonstrate the pronounced synergetic effect between the MIL-68(Al) matrix and functional CNTs related to remarkable removal performance.

The MWCNT-NH_2_-MIL-53(Fe) composite was synthesized following to the route 1 involving incorporation of a very small amount of MWCNT particles in the amino-functionalized NH_2_-MIL-53(Fe) matrix by a one-step solvothermal method [341]. The BET specific surface area, pore size and pore volumes of the MWCNT-NH_2_-MIL-53(Fe) system increased after introduction of MWCNT into the NH_2_-MIL-53(Fe) matrix as it was pointed above for other MOF-Carbon composites. In particular, the mesoporosity of the MWCNT-NH_2_-MIL-53(Fe) material significantly increased, thereby assisting the formation of additional active adsorption sites.

The MWCNT-NH_2_-MIL-53(Fe) composite was used in the adsorption of CTH and CTC hydrochloride antibiotics from aqueous solutions. The Q_m_ values of TCH and CTC achieved using the MWCNT-NH_2_-MIL-53(Fe) material were 368.49 and 254.04 mg/g (25 °C), respectively. These values are 1.79 and 8.37 times higher than those of chaff biochar as a traditional adsorbent for antibiotics removal. In particular, the Q_m_ values decreased in the order of MWCNT-NH_2_-MIL-53(Fe) > NH_2_-MIL-53(Fe) > MWCNT > chaff biochar for both TC and CTC.

According to the experimental results, both physisorption and chemisorption participate in the adsorption of TC and CTC on MWCNT-NH_2_-MIL-53(Fe). As for the chemisorption, the enhanced adsorption capacity can be assigned to hydrogen bonding between amino functional groups on MWCNT-NH_2_-MIL-53(Fe) and hydroxyl functional groups on TC or CTC, where H atoms from -OH groups could act as H-bond donors and the N atoms from the -NH_2_ groups serve as H-bond acceptors. However, the π-π stacking interactions between TC and CTC adsorbates and composite adsorbents are suggested as the dominant adsorption mechanism. Furthermore, the adsorption capacity of the MWCNT-NH_2_-MIL-53(Fe) composite was improved by tuning the porous structure, because the physisorption depended primarily on the pore size distribution of the adsorbent. So, the pore-filling effect contributed also in the adsorption enhancement.

The similar strategy was utilized for the incorporation of MWCNTs into ZIF-8 matrix resulting in the MWCNTs-ZIF-8 composite [342]. The preparation procedure for this material is illustrated in Figure 22. The MWCNTs-ZIF-8 adsorbent with a hierarchical porous structure was studied in the simultaneous removal of both radionuclides and organic pollutants from environmental water, U(VI) and humic acids (HA) from aquatic environments. The saturated adsorption amounts showed by the MWCNTs-ZIF-8 composite were 200.77 and 55.68 mg/g for U(VI) and HA at pH = 5.0, respectively. These values are higher than those for traditional adsorbents.

#### 4.8.2. Hydrogen Bonding and Zn-O-P Interaction

A reverse strategy (Composite type 2) was utilized for producing ZIF-8@MWCNT nanocomposite starting with almost the same components, i.e., zeolitic imidazolate framework ZIF-8 nanoparticles and MWCNT with a high content of hydroxyl functional groups [343]. This nanocomposite was obtained via an in-situ growth of ZIF-8 polyhedron nanocrystals on hydroxylated MWCNT as a host matrix. Upon integration with MWCNT, the ZIF-8 crystals tend to decorate within the MWCNT networks. In this way, three types of ZIF-8@MWCNT nanocomposites featuring different nanostructures are obtained by changing the MWCNT content.

The resulted nanocomposites were evaluated in the phosphate adsorptive removal from wastewater. The ZIF-8@MWCNT120 with the optimal MWCNT content in the system (prepared starting from 120 mg of MWCNT introduced in the reaction mixture) shows the Q_m_ of 203.0 mg/g, which is 0.7–3.7 times higher than those of the other common adsorbents as well as pristine ZIF-8 material. The high phosphate adsorptive removal was observed in the real wastewater (residual effluent Conc. = 0.0–0.45 mg/L) with initial concentrations of 0.63–6.32 mg/L.

It was found that the enhanced adsorption performance of the ZIF-8@MWCNT120 composite is attributed to a favorable adsorption mechanism involving hydrogen bonding and Zn-O-P interaction, which was confirmed by theoretical and experimental studies.

#### 4.8.3. π-π Stacking Interactions

The adsorption performance of the ZIF-8@CNT and ZIF-8@GO composites fabricated by embedding ZIF-8 nanoparticles in the CNT and GO matrices according to a one-step RT method was demonstrated [344]. In particular, an effect of GO and CNT host matrices for the composite adsorbent design was comparatively studied. A series of ZIF-8@CNT and ZIF-8@GO nanocomposites with a different loading of ZIF-8 nanoparticles was synthesized. In the case of ZIF-8@CNT and ZIF-8@GO nanocomposites, new pores are formed. This additional porosity may contribute to the adsorption of organic pollutants. Moreover, the nanocomposite formation facilitates a high dispersion of ZIF-8 nanoparticles, increasing of surface area, and tuning proper pore size along with thermal stability improvement. In particular, the thermal stability of the hybrid nanocomposites increases with the modulation of the ZIF-8 content.

The adsorption performances of the obtained ZIF-8@CNT and ZIF-8@GO nanocomposites with different amounts of ZIF-8 nanoparticles were compared in removal of MG in colored wastewater. It was observed that the q_e_ first increased and then decreased along with increasing the ZIF-8 content. The optimal ZIF-8 content was ~80 wt.% for both composite nanomaterials based on GO and CNT. These systems demonstrate enhanced adsorption capacity towards MG in comparison with the pristine ZIF-8 matrix at different pH values (3.5–7.0). The Q_m_ values of 1667, 2034, and 3300 mg/g were found for the ZIF-8, ZIF-8@CNT, and ZIF-8@GO materials (20 °C), respectively. Moreover, the adsorption performance of the composites could be improved at elevated temperatures. The suggested adsorption mechanism involves π-π stacking interactions related to the aromatic imidazole rings in the ZIF-8 framework and aromatic rings of MG.

Thus, a combination of MOF structures with carbon-based matrices, such as GO and CNTs results as a rule in higher and better dispersion –smaller nanoparticles sizes for MOF materials in the formed composites than in the case of pristine MOF materials.

#### 4.8.4. Three-Component Systems Including CNTs, MOF Crystallites, and Inorganic Matrix: Electrostatic Interactions

An efficient strategy for enhancing adsorption performance of MOF-CNTs-based composites is their combination with inorganic materials. For instance, the preparation and adsorption utilization of the SiO_2_@ZIF-67-CNTs composite with a favorable pore structure for organic contaminant adsorption was reported [345]. Note, using the ZIF-67 material for organic pollutant removal is associated with mass transfer limitations due to its microporous structure. So, preparation of the high-performance composites based on ZIF-67 crystallites and other materials, such as mesoporous SiO_2_ microspheres and CNTs is an efficient strategy. The preparation process for SiO_2_@ZIF-67-CNTs composites is presented in Figure 23.

The adsorption performance of the SiO_2_@ZIF-67-CNTs composite was evaluated in the MO removal at low temperature. So, at 5 °C, the saturated adsorption capacity of the composite for MO reached 112 mg/g, which was 1.49 times higher than that of the activated carbon. A high-voltage discharge treatment may increase the adsorption rate and adsorption capacity for the corresponding D-SiO_2_@ZIF-67-CNTs material. Moreover, calcination treatment of the SiO_2_@ZIF-67-CNTs material results in the improvement of its saturated adsorption capacity up to 324 mg/g for the calcined C-SiO_2_@ZIF-67-CNTs composite due to an increase of the specific surface area and hydrophobicity enhancement, so the adsorption capacities of the composites follow the following rank: C-SiO_2_@ZIF-67/CNTs > D-SiO_2_@ZIF-67/CNTs > SiO_2_@ZIF-67/CNTs > AC.

A favorable micron-sized pore structure of the SiO_2_@ZIF-67/CNTs material is accessible for the methyl orange adsorbate. The presence of CNTs increases its specific surface area. The adsorption performance of the composite is improved due to abundant adsorption sites in the (hetero)aromatic moiety in ZIF-67 and CNTs of SiO_2_@ZIF-67/CNTs contributed to the π-π interactions with MO molecules in aqueous solutions. The improving high-voltage discharge treatment increases the surface roughness of the composite, thereby providing additional adsorption sites. After calcination, due to the decomposition of the organic material, more pores are generated in the mesoporous silica microspheres. It was suggested that the electrostatic interactions impact significantly on the improvement of the adsorption rate of the composite, especially after treatment with the high-voltage discharge.

### 4.9. MOF-Biochar Composites: Electrostatic and π–π Stacking Interactions

Biochar is a valuable byproduct from both fast biomass pyrolysis during bio-oil production or slow pyrolysis processes at different temperatures. The utilization of biochar for the organic pollutant removal will be discussed in Section 6 of this review. Biochar may be utilized also for the preparation of the MOF-biochar composites serving as a host matrix for MOF crystallites with a high dispersion.

However, only one unique example of the integration of a MOF material and biochar for efficient organic pollutant removal is reported to date [346]. The MIL-53(Fe) metal-organic framework was crystallized starting from the bdc linker and FeCl_3_ into the preliminarily prepared biochar/Fe_3_O_4_ magnetic hybrid (MBC) with a high surface area (Figure 24). The resulting three-component MIL-53(Fe)-MBC composite adsorbent was used in adsorptive removal of rhodamine B and its photodegradation with or without Cr^6+^ ions.

The Rh B Langmuir adsorption capacity for MIL-53(Fe)/MBC was ~55 mg/g at pH = 6 (25 °C). It was found that electrostatic and π-π stacking interactions play a significant role in the Rh B sorption. Navarathna et al. [346] suggest that biochar serves as a surface dispersant for MIL-53(Fe) to reduce its particle agglomeration. Simultaneously, the biochar component increases the mechanical strength of the composites, and serves as a secondary adsorption site for heavy metals, oxy anions, and organic contaminants. Furthermore, highly-dispersed Fe_3_O_4_ nanoparticles in the MIL-53(Fe)/MBC composite assist in its separation and recovery after adsorption completion.

The examples of organic pollutants adsorption onto MOF-Carbon Composites are listed in the Table 3.

**Table 3 molecules-26-06628-t003:** Surface characteristics, adsorption capacity and mechanisms for the adsorption of organic pollutants using MOF-Carbon composites.

N	Adsorbent	Composition	S_sp_, m^2^/g	Surface Functional Groups	Adsorbate	Adsorption Conditions	Adsorption Capacity, mg/g	Mechanism of Adsorbent/Adsorbate Interactions	Ref
1	HKUST-1-AC	(Cu_3_(btc)_2_)-C composite	No data	No data	Crystal Violet	25 °C, pH = 3	133.33	Electrostatic interactions, ion exchange, H-bonding, soft-soft interactions, dipole-ion interactions	[302]
					Disulfine Blue		129.87		
					Quinoline Yellow		65.37		
2	MOF-5-AC	(Zn_4_O(bdc)_3_)-AC	No data	No data	Fast Green	25 °C pH = 3	21.230	Synergetic effect of MOF-5 matrix and AC-matrix	[303]
					Eosin		20.242		
					Quinine Yellow		18.621		
3	14-ZIF-8-C	(Zn-(meIm)_2_)-C derived from adipic acid	136.5	N	Malachite Green	T = 30 °C, pH = 4	3056	Electrostatic interactions, π-π stacking interactions, H-bonding	[305]
4	AC-NH_2_-MIL-101(Cr)	AC-Cr_3_O(abdc)_3_	1681	NH_2_	p-Nitrophenol	25 °C, pH = 5	18.3	Electrostatic interactions, π-π stacking interactions, H-bonding, open metal sites (Cr^3+^)	[307]
5	GO-MIL-101(Fe)-	GO10%-Fe_3_O(bdc)_3_	888.289 (Langmuir)	OH, COOH, COO^−^	Methyl Orange	25–65 °C, pH = 3–4	186.20	Electrostatic interactions, π-π stacking interactions, H-bonding, open metal sites (Fe^3+^)	[316]
6	GO-MIL-101(Cr)	GO(3%)-Cr_3_O(bdc)_3_	3259	OH, COOH, COO^−^	Naproxene	pH = 5.4	171	H-bonding (dominating), electrostatic interactions, π-π stacking interactions, open metal sites (Cr^3+^)	[317]
					Ketoprophene	pH = 7.0	140		
7	MIL-68(Al)-GO-2	AlOHbdc-GO (4.8 wt.%)	1309	OH, COOH, aromatic moiety, μ_2_-OH	Methyl Orange	pH = 8	400	Electrostatic interactions, H-bonding, π-π stacking interactions	[321]
8	rGO-NH_2_-MIL-68(Al)	AlOHabdc-rGO	1914	OH, COC, COOH, aromatic moiety, NH_2_ and μ_2_-OH	Congo Red	25 °C, pH = 4	473.93	Electrostatic interactions, H-bonding, π-π stacking interactions	[198]
9	NH_2_-MIL-68(In)-GO-2	InOHabdc-GO (5 wt.%)	679.5	OH, COC (epoxy), COOH, aromatic moiety, NH_2_, μ_2_-OH	Rhodamine B	pH = 6	267	π-π stacking interactions (dominating), electrostatic interactions	[323]
10	MIL-68(Al)-GO	AlOHbdc-GO in pellet form; MIL-68(Al): GO mass ratio = 10.00	-	OH, COC, COOH, aromatic moiety, μ_2_-OH,	Tetracycline hydrochloride	20–25 °C, pH = 4–9	228	H-bonding, π-π stacking interactions, Al-N covalent bonding	[326]
11	rGO-MIL-68(Al)	InOHabdc-15%rGO r = rhamnolipid-functionalized	761.97	OH^−^, COC, COOH, aromatic moiety	p-Nitrophenol	30 °C, pH = 5	332.23	H-bonding, π-π stacking interactions	[327]
12	GO-UiO-66-(OH)_2_	GO-Zr_6_O_4_(OH)_4_-bdc-OH_2_	239.5054 (BJH)	OH, COC, COOH	Methylene B	25 °C, pH = 11	96.69	Electrostatic interactions, H-bonding, π-π stacking interactions, open metal sites (Zr^4+^)	[315]
					Tetracycline hydrochloride		37.96		
13	GO-UiO-66-(COOH)_2_	GO (3 wt.%)-Zr_6_O_4_(OH)_4_-bdc-(COOH)_2_	369.96	OH, COC, COOH, COO	Tetracycline hydrochloride	30 °C, pH = 3	164.91 (Q_m_)	π-π stacking interactions (dominating), H-bonding, weak electrostatic interactions, open metal sites (Zr^4+^)	[328]
14	UiO-67-GO	Zr_6_O_4_(OH)_4_-bpdc-GO	-	OH, COOH, Zr-OH	Glyphosate	pH = 4	482.69	Surface/inner-complexation or chemical integration (dominating), electrostatic interaction at low pH values	[329]
15	ZIF-67-rGO aerogel	Co(2-meIm)_2_-rGO aerogel	491	Aromatic moiety, OH, COOH, N	Crystal Violet	pH = 6	1714.2	Electrostatic interactions, π-π stacking interactions	[331]
					Methyl Orange		426.3		
16	ZIF-Ga ZIF-67-GA	Co(2-meIm)_2_-GA aerogel	No data-	Aromatic moiety, N	Methyl Orange	No data	550.3	Electrostatic interactions, π-π stacking interactions	[332]
					Rhodamine B		380.7		
17	TMU-23-GO	[Zn_2_(oba)_2_(bpfb)]·(DMF)_5_ GO (10 wt.%))	No data	Aromatic moiety, NH-, CO	Methylene B	25 °C	≥90% of adsorption efficiency	Electrostatic interactions, π-π stacking interactions, acid-base interactions	[335]
18	HKUST-1(Ni)-GO	Ni_3_(btc)_2_-GO	69.6	OH, COOH	Congo Red	25 °C, pH = 4	2489	Acid-base interactions	[336]
19	AG-ZIF-67	Co(2-meIm)_2_-AG Alginate-graphene hydrogel	138.62	OH, OCO, COOH C=N, C-N, N-N	Tetracycline hydrochloride	30 °C, pH = 6	456.62	π-π stacking interaction, cation-π bonding	[337]
20	HKUST-1-Fe_4_O_3_-GO-β-CD	Cu_3_(btc)_2_-Fe_4_O_3_-GO-β-cyclodextrin	250.33	Aromatic moiety, CO, COO^−^, NH-	Imidacloprid	No data	3.11	H-bonding, hydrophobic interactions, electrostatic interactions, π-π stacking interactions	[338]
					Thiamethoxam		2.88		
					Acetamiprid		2.96		
					Nitenpyram		2.56		
					Dinotefuran		1.77		
					Thiacloprid		2.88		
21	Fe_3_O_4_/MOF(Co, Ni)@GO	Fe_3_O_4_-Ni_3_(BTC)_2_@GO	41.473	O-H, C=O, C=C, C-O, COO^−^	Methylene B		65.78		[339]
		Fe_3_O_4_-Co_3_(BTC)_2_@GO					70.42		
22	CNT@MIL-68(Al)	0.75 wt.%CNT@AlOHbdc	1407	Aromatic moiety, COO^−^	Phenol	25 °C	341.1	π-π stacking interactions, H-bonding	[286]
		3.5% CNT@AlOHbdc					109.9		
23	MWCNT-NH_2_-MIL-53(Fe)	MWCNT-FeOHabdc	125.50	Aromatic moiety, COO^−^, NH_2_-	Tetracycline hydrochloride	25 °C, pH = 3	368.49	π-π stacking interactions (dominating), H-bonding, pore filling effect	[341]
					Chlortetracycline hydrochloride		254.04		
24	MWCNT-ZIF-8	MWCNT-Zn(meIm)_2_	No data		Humic acids	25 °C, pH = 5.0	55.68		[342]
25	ZIF-8@MWCNT120	Zn(meIm)_2_@hydroxylated MWCNT	No data	Aromatic moiety, OH, N	Phosphate	30 °C, pH = 7	203.0 (Q_m_)	H-bonding, Zn-O-P interaction	[343]
26	ZIF-8@CNT	80 wt%Zn(meIm)_2_@CNT	830.3	Aromatic moiety, COOH, OH, COC	Methylene Green	20 °C, pH = 3.5–7.0	2034	π-π stacking interactions	[344]
	ZIF-8@GO	80 wt%Zn(meIm)_2_@GO	1476.4	heteroaromatic moiety, OH			3300		
27	SiO_2_@ZIF-67/CNTs	SiO_2_@Co(meIm)_2_-CNTs	993	(Hetero)aromatic moiety, COOH	Methyl Orange	5 °C	112	Electrostatic interactions, π-π stacking interactions	[345]
	D-SiO2@ZIF-67/CNTs		1005				194		
	C-SiO2@ZIF-67/CNTs		1135				324		
28	MIL-53(Fe)/MBC	FeOHabdc-Fe_3_O_4_-biochar	27.49	Aromatic moiety, COO	Rhodamine B	pH = 6, 25 °C	55	Electrostatic interactions, π-π stacking interactions	[346]

Thus, the relevant contemporary literature shows that MOF-based carbon composites have a number of enhanced properties regarding to their exploitation: (1) improved chemical and thermal stability; (2) modulated and enhanced porosity; (3) processability of MOF-carbon nanocomposites and the possibility to separate them from aquatic solutions after completion of adsorption; (4) MOF-carbon composites of different kinds feature sufficient working stability and can be utilized in consecutive adsorption cycles after regeneration with common solvents without any remarkable adsorption efficiency decrease; (5) diverse and enriched adsorption mechanisms in many cases, which can be different from adsorption mechanism realized for the pristine components of the system; (6) enhanced adsorption performance realized due to cooperative effects and synergism appeared due to intrinsic physico-chemical and adsorption characteristics of the composite components.

It could be concluded that in many cases, textural properties of materials, such as the specific surface area, pore volume, and pore size do not impact significantly on the adsorption capacity of the MOF nanomaterials and MOF-carbon nanocomposites in the adsorption of organic pollutants from aqueous media.

Other factors, such as specific interactions, i.e., electrostatic, acid-base, π-π stacking interactions, hydrogen bonding, complexation with open metal sites have a more pronounced effect on the MOF-carbon nanocomposite performance in the adsorption of organic pollutants in water. So, it could be concluded, that MOF functionalities impact significantly more strongly than MOFs textural properties. This observation or conclusion is also true for the MOF-carbon nanocomposites.

It can be seen clearly from literature examples that using different MOF-carbon nanoadsorbents based on different carbonaceous components, such as AC, GO, rGO, CNTs, or biochar, and the same MOF component, different adsorption mechanisms can be implemented in adsorption of the same pollutants from water. And vice versa, using the different MOF structures and the same carbon component, various interactions can govern the adsorption of organic pollutants, so a careful and judicious choice of the components for the MOF-carbon composite nanoadsorbents should be taken into account for the specific process of organic pollutant removal from water.

## 5. Highly Ordered Porous Carbons Derived from MOF Matrices

### 5.1. Background

As pointed out in previous sections, MOF matrices offer a number of outstanding physico-chemical characteristics, which assist their utilization in the adsorption of organic pollutants from water. Simultaneously, MOF adsorbents feature some limitations, in particular, for applications in an aqueous phase, due to their hydrophilicity and relatively poor stability in water [8,9]. One of the efficient ways to overcome these limitations is fabrication of the functional composites, i.e., MOF-carbon by integration of the MOF nanocrystals with carbon-based materials, such as activated carbons, CNTs, GO, rGO, etc.

One more efficient strategy for increasing the stability, hydrophobicity, and mechanical strength of MOF-based adsorbents for removal of organic pollutants from aquatic media is preparation of porous ordered carbons with a high surface area and diverse surface functionality by controllable pyrolysis of MOF materials [347]. In particular, MOF matrices may serve as a proper template for preparing functional highly porous carbon materials with the particle size changing in the micro-/nano-range [5]. The presence of the organic component, i.e., organic linkers in the metal-organic frameworks allowed the formation of carbon-based materials directly from the MOF materials such as both graphitic and amorphous carbon [290]. Therefore, among other precursors of porous carbon-based matrices, nanoporous MOF matrices are promising sources-templates for the fabrication of highly ordered carbons by pyrolysis [348].

As it was pointed out in numerous literature examples and in previous sections, porous carbons are widely utilized for water remediation as adsorbents due to their excellent hydrophobicity and surface functionalities. Under the controllable thermolysis conditions, volatile molecules including carbon oxides, vapor, etc. can be rapidly released from the carbon precursors, then leaving a porous nanostructured matrix [349].

In this way, MOF matrices can be converted to metal oxide/porous carbon materials including their composites by pyrolysis under an inert atmosphere [50]. Depending on the properties of the structural building units of MOF structures and the carbonization conditions (temperature, time, and atmosphere with or without oxygen), the properties of the resulting materials varied with their structures (graphitic or disordered carbons), doped heteroatoms (metal, O, N, etc.), porosities, and functionalities [350].

An additional advantage of the use of MOF precursors for pyrolysis relates to the formation of composite carbon materials containing highly-dispersed metal/metal oxide [351]. However, as a rule, the porosity of carbonaceous materials derived from MOF precursors is not always high because of the co-presence of metal oxides or metals in the produced carbons [352]. So, an alternate approach involves removal of metal/metal oxide (nano)particles embedded in the carbon matrices.

Accordingly, there are two alternative strategies for the fabrication of highly porous carbons using MOF matrices as templates/precursors, i.e., the preparation of hybrid materials containing metal and metal oxide nanoparticles embedded in the carbon matrix after pyrolysis or removal of the metal species from obtained carbon in a post-carbonization mode by further heating or etching by acids. An appropriate choice and utilization of two strategies for the preparation of the MOF-derived carbons is dependent on the specific adsorption task.

To date, a wide variety of carbon-based functional porous materials with different nanoarchitecture were fabricated by direct carbonization of MOF matrices [353]. These MCs have been recognized as promising functional materials due to their highly-ordered nanoporous structure, a high surface area and tunable chemical and physical properties. Therefore, they are multipurpose materials with potential applications in catalysis, energy storage and conversion including electrode materials, and adsorption technologies and environmental remediation [354]. In particular, several MOF-derived carbons show advanced performances in the adsorptive removal of organic contaminants from different media. An example is purification of fossil fuels using carbon as an adsorbent derived from ZIF-8 as a precursor [355].

Due to a number of unique characteristics, the MCs have been regarded as efficient adsorbents for removal of both organic and inorganic contaminants from water and wastewater [356]. In particular, textural characteristics and intrinsic porosity of MCs are the essential factors that contribute to their adsorption capacity and selectivity [357]. Depending on the adsorbent nature, adsorbate kind and processes parameters, the removal efficiency as high as 99% could be achieved.

The commercial use of an adsorbent relies on its reusability so that it can be used for long time. The reusability of the MCs utilized in adsorption of both organic pollutants and heavy metals was extensively studied. In almost all cases, the regeneration of these adsorbents can be achieved by simple washing with water and/or alcohols, such as ethanol. So, the regenerated MCs may be utilized in four or more adsorption cycles [5].

### 5.2. Factors Determining the Adsorption of Organic Pollutants from Water Solutions: Realization of Specific Adsorption Mechanisms for MCs

In rational design and preparation of advanced carbon-based adsorbents, it is important to outline the differences between the mechanisms realized for the MOF-Carbon composites, MCs, carbon nanoadsorbents, and traditional carbon matrices. So, the examples of mechanisms implemented using MCs for the removal of organic contaminants from water will be considered below.

For most MCs, the experimental results can be plotted adequately with the pseudo-second-order model. Similarly, to MOF-Carbon composites, the adsorption isotherm obeyed in most cases the Langmuir model.

#### 5.2.1. Hydrogen Bonding, π-π Stacking Interactions, and n-π Interactions

The first strategy (see above), which involves utilization of a hybrid carbon material containing metal and metal oxide nanoparticles formed after pyrolysis on the carbon surface has been recognized as promising one for the preparation of an advanced adsorbent designed for the removal of water contaminants. For instance, a few studies report producing the Cu/Cu_2_O/CuO@C composite for adsorptive removal of antibiotics in wastewater [358].

This highly porous nanostructured composite containing Cu and CuO nanoparticles supported on a carbon matrix was prepared by self-sacrificial HKUST-1 template pyrolysis. According to the physico-chemical characterization, the surface of the Cu/Cu_2_O/CuO@C material with a hollow structure contains diverse chemical bonds of abundant functional groups such as O-H bonds, CO bonds of alcoholic groups, OCO bonds of carboxyl groups, CO^2^- bonds of carbonyl/carboxyl groups, and C-OOH carboxylic groups. In this case, due to the presence of the Cu/Cu_2_O species in the produced composite, its S_sp_ (BET) is as low as ~80 m^2^/g.

The produced composite was studied in the adsorption of antibiotics, i.e., CIP and TC. The adsorption values demonstrated by the Cu/Cu_2_O/CuO@C material towards CIP and TC from water were 67.5 and 112.5 mg/g, respectively. In addition, the HKUST-1-dervied porous carbon also showed an adsorption capacity (37.2 mg/g) and removal efficiency (87.6%) towards chloramphenicol antibiotic.

It was found that adsorption proceeds in many steps onto the Cu/Cu_2_O/CuO@C material. The first step of the antibiotic adsorption is characterized by the migration of CIP and TC molecules from the solution to the composite surface via bulk diffusion. The second one can be the mass transfer across the boundary of films. The third one proceeds via intra particle/pore diffusion of CIP and TC antibiotics.

Both studied antibiotic molecules have several aromatic rings, which can provide many π-conjugated systems. According to the Raman spectroscopy results, the Cu/Cu_2_O/ CuO@C composite can contain graphitic carbon with π-electron-rich aromatic rings. XPS characterization indicates the presence of π-π* bonds belonging to aromatic moiety on the Cu/Cu_2_O/CuO@C surface. Accordingly, adsorption of the studied drug molecules may be based on interactions between π-electron systems of the aromatic rings/π-conjugated systems between CIP/TC and adsorbent. So, suggested adsorption mechanisms implemented using a Cu/Cu_2_O/CuO@C nanocomposite involve hydrogen bonding, π-π stacking interactions, and n–π interactions.

#### 5.2.2. Electrostatic Interactions, Hydrogen Bonding, Pore-Filling Mechanism, π-Electron Polarization and Hydrophobic Interactions

In the context of the second approach involving removal of metal species from obtained MC materials, Zn-based MOF matrices are promising precursors for functional carbons with a high surface area and porosity since the produced ZnO upon carbonization of Zn-MOF precursors can be easily removed due to ability of carbons to reduce ZnO species into metallic Zn with a relatively low boiling point (907 °C) [50].

Therefore, the pyrolysis of Zn-based MOF materials and study of the potential applications of the produced carbons draw an increased research interest. An example is the utilization of the ZIF-8 material as a precursor for the preparation of nanoporous carbons through one-step carbonization [52].

The ZIF-8 matrix was selected as a precursor for the MC adsorbent due to its high thermal stability assisting producing carbons with a high specific surface area and pore volume. For this purpose, Zn^2+^ species in the ZIF-8 precursor may be removed through vaporization during pyrolysis. Furthermore, the nitrogen containing MC material is produced due to the presence of 2-methylimidazole organic linkers in the ZIF-8 precursor as a nitrogen source for the resulted highly-ordered carbon matrix. The presence of nitrogen may enhance the adsorption performance of this carbon material.

Note, the physico-chemical properties of MOF-derived carbons can be further tuned by using nitrogen as a dopant. For instance, it was observed that the textural characteristics, i.e., specific surface area and porosity and morphology of carbonaceous materials are also improved [359,360]. Moreover, the adsorption performance of MOF-derived carbons is usually enhanced with nitrogen doping [361].

The most direct way for preparation of nitrogen-doped porous carbon is related to an appropriate choice of a MOF precursor for high-temperature pyrolysis. So, one more reason for the selection of ZIF-8 as a precursor is its nitrogen enriched composition due to its meIm linkers. The direct carbonization of the ZIF-8 material was carried out at different temperatures for producing MC matrices with specific physico-chemical and adsorption properties.

The adsorption performance of the produced carbons denoted by authors as NPC for the removal of CIP was evaluated. Due to the six-membered-ring pore windows (0.34 nm) of ZIF-8-dervied carbon materials (similarly to the ZIF-8 precursor), they showed a high adsorption capacity of CIP. The optimal NPC-700 (NPC carbonized at 700 °C for 2 h) exhibited an enhanced performance in CIP adsorptive removal. The maximum adsorption capacity for CIP based on the Langmuir model was 416.7 mg/g, which was higher as compared to other known carbon materials.

The adsorption tests were carried out under different conditions. It was found that CIP functional groups existed as cationic and zwitterionic species in solution at pH = 6.0 contributing to electrostatic and hydrophobic interactions with the NPC-700 carbon surface. Note, CIP adsorption on carbon is commonly administered by electrostatic interaction between the CIP molecules and the carbon surface. It was suggested that both enhanced electrostatic and hydrophobic interactions played a major role in the CIP adsorption using ZIF-derived nanoporous NPC-700 carbon.

The next example of the utilization of a MOF matrix with Zn^2+^ metal nodes in the framework as a carbon precursor is the preparation of NPC by direct carbonization of a MOF-5 material [357]. The adsorption performance of produced MOF-5-derived carbons as multipurpose adsorbents was investigated with SMZ, BPA, and MO. Their adsorption performance was comparatively studied along with those of other carbon adsorbents, i.e., SWCNTs and commercial PAC.

In the context of the adsorption efficiency, the MOF-5-derived carbon has favorable textural properties, such as a hierarchical micro/meso/macroporous structure and a large S_sp_ (BET) value (1731 m^2^/g), which was much higher than that of PAC (901 m^2^/g) and SWCNTs (585 m^2^/g). Furthermore, its meso/macroporous volume (1.68 cm^3^/g) calculated for 76% of the (V_total_ was higher than the V_total_ values of SWCNTs (1.15 cm^3^/g) and PAC (0.87 cm^3^/g).

Along with this trend, MOF-5-derived carbon shows high maximal adsorption capacities of 625, 757, and 872 mg/g, respectively, towards SMZ, BPA, and MO, 1.02–3.23 times higher than those of SWCNTs and commercial PAC. The adsorption values changed in sequence of MO > BPA > SMZ, despite the lower hydrophobicity of MO dye using MOF-5 derived carbon.

Note, while for slit-shaped pores, the smallest dimension of the pollutant molecule is the determining factor. Accordingly, the pore-filling mechanism may contribute to adsorption of these compounds thanks to their 2D size smaller than ~1.1 nm.

The pH value of the working solution has a crucial impact on adsorption, indicating a pronounced contribution of electrostatic interactions between NPC and studied organic adsorbate molecules, so based on characterization and adsorption studies, a possible cooperative adsorption mechanism involves a pore-filling mechanism, electrostatic interactions, π-electron polarization, and hydrogen bonding between organic molecules or between the adsorbent and contaminant.

The study related to carbonization of another celebrate water-stable Zn-based metal-organic framework, i.e., MOF-74(Zn) [Zn_4_O(dhbdc)_3_], at different pyrolysis temperatures under a nitrogen atmosphere was reported [355]. Note, this MOF structure is the isoreticular analog of the MOF-5 structure having the same metal nodes, i.e., Zn^2+^ ions and network topology [362]. In this case, the first strategy of the MC preparation was utilized, which results in the preparation of zinc oxide nanoparticles incorporated in multifunctional porous carbon composite materials derived from MOF-74 (Zn).

The adsorption performance of the ZnO-C composites prepared in different temperature regimes were studied in the adsorptive removal of RhB from water. The ZnO-C composites obtained at a higher pyrolysis temperature exhibited a better adsorption capability, since these conditions resulted in a higher specific surface area and more active sites in the composites. In particular, the composite with hierarchical pore structures obtained at 1000 °C (ZnO-C1000) with a surface area of 782.971 m^2^/g and pore volume of 0.698 m^3^/g shows the most efficient performance. It was also demonstrated that the ZnO-C composites retained their adsorption capability in a wide range of the solution pH values. Noteworthy, the reported ZnO-C composites show multipurpose potentials as efficient pollutant absorbents as well as electrode materials for supercapacitors.

MOF matrices that are constructed with the incorporation of several biomolecules are denoted as Bio-MOFs. The Zn_8_(adenine)_4_(bpdc)_6_O metal-organic framework denoted as Bio-MOF-1, is a typical one that is composed of Zn and adenine interconnected with dpdc linkers. The Bio-MOF-1 matrix shows a potential as an attractive precursor for functional carbonaceous materials since MOF is composed of N- and O-containing organic moieties and has a high permanent porosity [67,68]. It was used as a precursor for producing porous carbons by carbonization at 1000 °C for different time, namely, 6, 12, and 24 h resulting in Bio-MOF-1-derived carbons (denoted as BMDC-xh, where x represents the pyrolysis time) [50]. According to physico-chemical characterization, the BMDC matrices are highly O- and N-doped porous carbons with a high degree of graphitization.

The amount and kind of functional groups on the BMDC surface may be controlled efficiently by pyrolysis conditions, i.e., time and temperature. The total content of acidic groups, i.e., carboxylic and lactonic groups, increased with the increasing pyrolysis time (BMDC-6 h < BMDC-12 h < BMDC-24 h. On the contrary, the number of phenolic groups did not change with the variation of the pyrolysis time. So, the BMDC-12 h material had a bit higher concentration of phenolic groups than other BMDC matrices. In its turn, the amount of basic groups decreased with the decreasing N content in the BMDCs.

BMDC materials and commercial AC were comparatively studied in the adsorption of bisphenol A from water. The BMDC-12 h adsorbent showed an efficiency ~5-times (in terms of q12h, the adsorbed quantity for 12 h) of that of a commercial AC in BPA capture. Furthermore, BMDC-12 h exhibited the enhanced performance for BPA adsorption as compared to reported common adsorbents.

Based on adsorption tests at different solution pH values, it was suggested that H-bonding was a dominating mechanism due to functional groups present in BPA and functional groups on the BMDC-12 h surface. Hydrophobic interactions and π-π stacking interactions are also involved in BPA adsorption, especially at pH of the working solution >pKa (9.6) of BPA.

#### 5.2.3. Electrostatic Interactions and Hydrophobic Interactions

Other promising metal nodes in MOF-precursors for the preparation of MC matrices are based on La^3+^ ions. The synthesis of a MOF-La material and its further carbonization affording a carbonized lanthanum-based organic framework (denoted as CMOF-La) was reported [363]. MOF-La crystallites retain the original shape of the pristine material. However, lanthanum oxide was produced in a rod form during the carbonization. This resulted in the formation of the ordered parallel arranged channels, which are loaded by metal oxide nanoparticles with rod-like structures. It was found that La oxide nanoparticles are responsible for the CMOF pore structure formation. The CMOF structure with opened channels left after acid treatment in the carbonized MOF and thereby etching La oxide nanoparticles occurred.

CMOF-La shows an efficiency in the adsorption of AR 18 from water. The adsorption capacities are dependent on pH of the solution. Optimal adsorption conditions were obtained at a low pH value of 3. Taking into account the changes in the functional groups in the CMOF before and after dye adsorption confirmed by physico-chemical characterization, electrostatic interactions are suggested as a dominant mechanism for the adsorption of acid red 18 onto carbonized MOF. It was found also that the adsorption of anionic dye onto CMOF-La is driven by hydrophobic interactions between the carbonized metal-organic framework and acid red 18.

#### 5.2.4. Hydrophobic Interactions

An interesting strategy was used [364] for producing the hybrid carbon materials from AC and ZIF-67 crystallites. Trace amounts of ZIF-67 were introduced via impregnation onto commercial AC pellets. Then AC pellets decorated with ZIF-67 nanoparticles were carbonized (800 °C, 6 h) affording N-doped porous carbon. The obtained hybrid carbon material denoted as cal-ZIF-67/AC had highly dispersed Co nanoparticles and N-functionality on its surface. Its S_sp_ and V_total_ values were 833 m^2^/g and 0.56 cm^3^/g, respectively. The cal-ZIF-67/AC hybrid with weak magnetic properties was studied as a multifunctional material in adsorption of RhB and simultaneously as a catalyst to activate PMS for RhB degradation. It was found that the cal-ZIF-67/AC hybrid demonstrated a maximum adsorption capacity of 46.2 mg/g, which was more than twice that of pristine AC. Furthermore, it showed high activity in effective activation of PMS to produce sulfate radicals for the oxidative degradation of RhB.

In terms of nitrogen doping, metal-organic frameworks with azolate-based linkers (metal-azolate frameworks, MAFs, a sub-class of MOF structures) along with ZIF structures belonging to the MAF family, e.g., ZIF-8 and ZIF-67, are promising carbon-sources and templates. S. H. Jhung reported the preparation of carbon materials derived from hydrophobic MAF materials including MAF-6 with Zn^2+^ ions and utilization of them in water remediation by adsorptive removal of PAHs-NAPH, ATC, PRN and BZ from water [356].

The nanoporous Zn, N-containing carbon hybrids were obtained by pyrolysis of MAF-6 precursors for a different time (6–36 h). For example, MC-24 was obtained by the 24-h pyrolysis of MAF-6. It was found that the physico-chemical properties of MC matrices changed depending on the pyrolysis time. It is important to note that the degree of graphitization and the hydrophobicity increase with the increasing pyrolysis time.

AC and pristine MAF-6 material were studied in the adsorption tests as reference samples. All studied adsorbents show adsorption efficiency for all adsorbates in the order BZ < NAPH < ATC < PRN. The carbonaceous materials derived from the MAF-6 precursor showed enhanced adsorption performance for the removal of PAHs and BZ from water compared to the pristine MAF-6 matrix and AC.

The adsorbed quantities achieved in 12 h (q12h) increased in the order MAF-6 < AC < MC-2 < MC-6 < MC-12 < MC-36 < MC-24 (the number corresponds to the pyrolysis time). However, the q12h value based on the surface area (m^2^/g) of the adsorbents showed a slightly different performance order, i.e., MAF-6 < AC < MC-2 < MC-6 < MC-12 < MC-24 < MC-36.

It was suggested that the hydrophobicity of the produced MC adsorbents increased with the increasing pyrolysis time due to a decrease in the total content of heteroatoms (Zn, N, O), so it followed the order MC-36 > MC-24 > MC-12 > MC-2. However, the most prolonged pyrolysis time (36 h) resulted in a porosity decrease for the MC-36 matrix. Accordingly, MC-24 was evaluated as the optimal adsorbent for all PAHs and BZ in terms of adsorption kinetics among the selected adsorbents due to its high surface area and hydrophobicity. The adsorption results correlated with hydrophobicity of the studied polycyclic hydrocarbons and MAF-6 derived adsorbents. Therefore, hydrophobic interactions were suggested as the main mechanism for the adsorption of PAHs and BZ from water over the MC matrices.

#### 5.2.5. Hydrogen Bonding

Another example of the adsorption performance of MAF-6 derived-MDCs prepared at different pyrolysis times is the removal of widely used ASs such as SAC, ACE, and CYC from water [2]. According to Boehm titration tests, the obtained MDCs contain abundant carboxylic, lactonic, and phenolic groups at the surface. In particular, the amounts of carboxylic and lactonic groups increased with the increasing pyrolysis time. On the contrary, the content of phenolic groups increased in the almost opposite order MDC-2 h < MDC-36 h < MDC-24 h < MDC-4 h < MDC-12 h < MDC-6 h.

The MDC material derived from zinc-azolate MAF-6 prepared for 6 h (MDC-6 h) showed a higher adsorption capacity compared to that of commercial AC as well as some carbon materials derived from zinc imidazolate ZIF-8 precursor with Zn^2+^ ions.

Note, in this case, the optimal adsorbent MDC-6 h with a highest content of phenolic groups and relatively low content of carboxylic and lactonic groups was carbonized in a shorter time (6 h) as compared to the previous example relevant to the study of PAHs adsorptive removal using MAF-6 derived carbons. In the latter case, the optimal adsorption time was 24 h. Note, this material has lower porosity as comparted to other MAF-6-derived MDCs.

The adsorbed quantity values qt for the tested adsorbents followed the order MAF-6 < AC < MDC-2 h < MDC-36 h < MDC-24 h < MDC-4 h < MDC-12 h < MDC-6 h. In particular, the saccharin adsorption value for best MDC-6 h material was ~93 mg/g (12 h), while this value for AC reference sample was ~4.7 mg/g under the same conditions.

The enhanced adsorption performance of the MDC-6 h material can be explained by the highest concentration of functional groups, especially, phenolic groups on its surface. Therefore, H-bonding between the surface functional groups of MDC matrices and ASs was suggested as the dominating mechanism for the AS adsorption, where MDC adsorbents and the AS molecules acted as the H-donor and H-acceptor, respectively. π-π Stacking interactions between π-electrons on ASs and the graphene layer of the MDC matrix contribute also to the enhanced adsorption performance.

### 5.3. Bimetallic MOF Matrices as Precursors for NPC

A promising approach for the preparation of the efficient adsorbents is the utilization of bimetallic MOF materials as precursors for pyrolysis to obtain highly ordered carbon matrices. An example is the development of a Zn/Co bi-MOF by SAS to synthesize self-supported N-doped Co-based magnetic porous carbon composites by one-step pyrolysis of the mixed-ligand bimetallic ZIF (CoxZny-JUC-160), where x and y correspond to Co and Zn ratio in the composite [365].

For this purpose, the bimetallic JUC-160 metal-organic framework with Zn^2+^ and Co^2+^ ions as inorganic nodes in the framework was selected as a precursor of the CoxZny-JUC-160 porous carbon. The JUC-160 network is composed of two linkers—2-mbIm and bIm and Zn/Co metal centers with different Co/Zn ratios. The reference monometallic Zn-JUC-160 and Co-JUC-160 materials were studied for comparison.

The reported strategy for the preparation of corresponding porous carbon materials involves the synthesis of a series of monometallic and bimetallic ZIF structures, i.e., Zn-JUC-160, bimetallic CoxZny-JUC-160, and Co@JUC-160, followed by pyrolysis (Figure 25). During pyrolysis, the ZnO species produced via CoxZny-JUC-160 thermal decomposition was reduced by carbon, then formed zinc was evaporated, which resulted in the generation of pores in the carbon material. The obtained N-doped carbon composites had a hierarchical porous structure, well-developed graphitized walls, and uniformly dispersed embedded magnetic Co species.

The carbon composites were comparatively studied in adsorption of ADQ drug from aqueous solutions. The Co/Zn ratio in JUC-160 precursors impacted seriously on their textural properties and adsorption performance. The adsorbed quantities of ADQ at the same equilibrium concentration follows the order: Co@NC-1/4-900 > Co@NC-1/2-900 > Co@NC-1/8-900 >Co@NC > NC. The adsorption capacities of these carbon materials were much higher than the adsorption capacity measured for the parent JUC-160 matrices.

The optimal Co1Zn4-JUC-160 composite (Co@NC-1/4-900) obtained by pyrolysis at 900 °C showed fast adsorbate diffusion. The ADQ adsorption capacity measured for it was as high as 890.23 mg/g. These characteristics achieved using the Co1Zn4-JUC-160 composite were superior as compared to commercial activated carbon and other reported porous adsorbents. Pan et al. [365] suggested that the excellent adsorption capacity of the Co@NC-1/4-900 composite could be ascribed to the synergism of three key factors that affect the adsorption performance—S_sp_ and inner-connected pore structure, metal content and particle size effects. The presence of Co nanoparticles provided also the easy adsorbent separation by a magnet. The proposed adsorption mechanism involves complexation of amodiaqine molecules with Co nanoparticles, hydrogen bonding due to abundant functional groups in ADQ and Co@NC-1/4-900 composite, and π-π stacking interactions.

### 5.4. Preliminary Modification of MOF Precursors with N-Containing Compounds

#### Hydrogen Bonding

The next efficient strategy for further increasing the N content in the MCs is the preliminary modification of metal-organic precursors with specific compounds. For instance, the MAF-6 material was modified by N-enriched melamine resulting in an extra N-containing precursor, which was carbonized affording a nitrogen-doped highly porous carbon materials (denoted as CDM@M-6) in order to achieve better adsorption performance [366]. The porosity and defect concentration of the CDM@M-6 carbons were dependent on the quantity of melamine loaded in the mela@MAF-6 composite.

The CDM@M-6 materials were studied in the adsorptive removal of NIABs, such as DMZ, MNZ, and MZ from water. The adsorption studies revealed a pronounced effect of the melamine content in the MAF-6 material before pyrolysis on the adsorption quantities of NIABs (q12 h), so the CDM(0.25)@M-6 system derived from the melamine@MAF-6 precursor containing 0.25 wt.% of melamine demonstrated an enhanced adsorption performance.

NIAB molecules contain nitro groups. In turn, there are carboxylic and phenolic groups and the nitrogen species on the CDM(0.25)@M-6 surface. The total quantity of acidic and basic functional groups in CDM@M-6 s was relatively higher than those in the non-modified CDM-6 material and AC. Moreover, the total amount of the functional groups (both acidic and basic) on the CDM@M-6 sample increased with the increasing loaded melamine content, which was confirmed by FTIR, XPS, and elemental analysis measurements.

Due to these abundant functional groups, while using CDM (0.25)@M-6 material, the adsorbed quantities (q12 h) were ~1.7, 1.8, and 1.9 times higher for DMZ, MNZ, and MZ, respectively, as compared with the AC reference sample. In particular, the O_m_ values calculated for the CDM (0.25)@M-6 material were as follows: for DMZ—621 mg/g and for MNZ—702 mg/g). Its enhanced performance was ascribed to hydrogen bonding, where CDM@M-6 and DMZ/MNZ/MZ acted as the H-donor and H-acceptor, respectively.

Another approach for the preparation of highly ordered porous carbons with a high concentration of nitrogen is the introduction of IL in the MOF precursor before pyrolysis. In this way, MCs with a high nitrogen content were obtained by the calcination of ZIF-8 precursor after introducing different amounts of an ionic liquid (IL) (IL@MOF) via the ship-in-bottle method (SIB) affording IL@MOF-derived carbons (IMDCs) [367].

It was established by physico-chemical characterization that doped nitrogen exists in three major forms (N-6, N-5, or N-Q (quaternary)) on the surface of the IMDC materials. In particular, the pyridinic (N-6) and pyrrolic (N-5) nitrogen content in the IMDC matrices increased with increasing the introduced IL content in the ZIF-8 precursor (before pyrolysis) or with decreasing the pyrolysis temperature.

IMDC and MDC carbons (without preliminary modification with IL) were studied in the liquid-phase removal of organic contaminants, i.e., ATZ, diuron, and diclofenac in both water and a hydrocarbon. According to adsorption tests, obtained IMDC materials demonstrated an efficient removal of the studied contaminants from both phases.

Interesting to note, IMDC materials showed enhanced performance compared to MDC despite their lower porosity. Therefore, it was concluded that IL loading in ZIF-8 (and hence, increased nitrogen content in the carbonaceous materials) has a decisive impact on the adsorption.

In particular, the IMDC-1000(12%) material obtained by pyrolysis of the ZIF-8 matrix with 12 wt.% of IL was evaluated as the best adsorbent as compared to AC, ZIF-8 precursor, and MDC-1000 material derived from the pristine ZIF-8 matrix. In particular, in the case of the ATZ adsorption, the Q_m_ value of IMDC (208 m^2^/g) was much higher than that for activated carbon (60 m^2^/g) and MDC (168 m^2^/g). The kinetics of adsorption slightly increased using MDC and IMDC materials compared to commercial AC in the case of ATZ. The enhanced adsorption performance of the studied IMDC materials may be explained by the H-bonding mechanism realized due to N-6 and/or N-5 nitrogen species on their surface.

The results of adsorptive denitrogenation and desulfurization of fuel were similar to of the results obtained for water solutions. This justifies the multipurpose adsorption performance of IMDC materials.

It was demonstrated [5] that the preliminary modification of the MOF matrices based on Al^3+^ ions with ionic liquids is an efficient strategy for further improvement of the adsorption performance of the porous carbons derived from them due to increasing nitrogen contents. In this work, nitrogen-doped porous IL@AlPCP carbons were produced by direct carbonization of the AlPCP metal-organic precursors containing different amounts of an ionic liquid.

According to the elemental analysis results and XPS characterization performed for the studied carbon materials, CDIL@AlPCP matrices produced from AlPCP precursors loaded with ionic liquids had a significantly higher nitrogen content than the CD@AlPCP sample derived from the AlPCP material without preliminary modification with IL. The nitrogen contents in CDIL@AlPCP matrices, such as surface N-6 and N-5, i.e., pyridinic nitrogen (N-6), pyrrolic/pyridonic nitrogen (N-5), increased along with the increasing IL loading in the AlPCP precursor. The surface areas and pore volumes of the IL@AlPCP matrices decreased with the increasing amount of loaded IL and accordingly the nitrogen content in the carbon material.

Porous carbons derived from IL@AlPCP and pristine AlPCP (denoted as CDIL@AlPCP and CDAlPCP, respectively) were utilized in the liquid-phase adsorption of a number of pharmaceutical and PPCPs with a phenolic group for both aqueous and non-aqueous media. In particular, the CDIL@AlPCP materials showed enhanced adsorption performance for the removal of PCMX, TCS, and ACP from water.

For example, the adsorption capacity of CDIL@AlPCP was the highest for PCMX as compared with the pristine CDAlPCP matrix and commercial activated carbon. The adsorption efficiency for the studied PPCPs decreased in the following order: CDIL(0.5)@AlPCP > CDIL(1.0)@AlPCP > CDAlPCP > AC. In particular, the optimal CDIL(0.5)@AlPCP adsorbent demonstrated 2.2, 4.0, and 2.1 times the adsorption amount at 12 h (q12 h) for PCMX, TCS, and ACP adsorbates, respectively, as compared with the AC reference sample.

Furthermore, CDIL(0.5)@AlPCP showed the highest adsorption for PCMX removal, compared with many reported adsorbents including MIL-101 metal-organic framework [12], powder AC [368], and hyacinth stem [369].

Note, the adsorption capacity of the CDIL(1.0)@AlPCP material produced by using a higher IL loading in the AlPCP precursor towards studied contaminants as compared to the CDIL(0.5)@AlPCP counterpart decreased. The suggested reasons for this drop were the decrease of the surface area and pore volume.

The total amount of acidic and basic functional groups in the best CDIL(0.5) @AlPCP material was higher than those for other studied carbon materials. Accordingly, the enhanced adsorption performance of the CDIL@AlPCP adsorbents, first at all, CDIL(0.5)@AlPCP material, for both aqueous and non-aqueous phases was explained by the H-bonding mechanism due to phenolic groups in adsorbate molecules and abundant functional groups measured by the Boehm titration method and surface nitrogen species determined by XPS and FTIR in nitrogen-doped carbon adsorbents. For instance, PCMX may act as an H-donor, and ample N and O species of carbon materials may act as an H-acceptor. The hydrophobic interactions and π-π interactions contributed also to the adsorption performance of the CDIL@AlPCP adsorbents.

### 5.5. Using Inorganic Templates for Advanced Hybrid MCs: Electrostatic Interactions

The next step in the optimization of the adsorption properties of advanced carbon-based materials is the preparation of hybrid materials on their basis by combination with inorganic matrices and formation of rather complicated nanoarchitectures. For this goal, it will be interesting to compare the adsorption performance of a carbon adsorbent derived from the ZIF-8 crystallites and hybrid nanomaterial based on it and SiO_2_ nanoparticles.

This approach is illustrated by the preparation of a composite in the form of C@silica core/shell nanoparticles used for the removal of CIP from aqueous solutions [48]. The nanoparticles were prepared by coating a layer of silica gel on ZIF-8 nanoparticles via hydrolysis of TEOS followed by carbonization. The produced C@silica nanoparticles composed of ~85 wt.% of carbon. According to TEM studies, the core/shell C@silica structures were formed with a thickness of the silica layer of 13–28 nm. They had a hierarchical pore structure composed of micro- and mesopores and a surface area (BET) of 594.4 m^2^/g that assists to CIP adsorption from water.

The estimated Q_m_ value on the C@silica nanoparticles for CIP is 516.8 mg/g, which is higher than that of the ZIF-8-derived carbon, i.e., 416.7 mg/g [52] (see above) and 430.6 mg/g [48], as well as those of other adsorbents, such as GO and composites, carbon nanotubes, activated carbons (15.6–379 mg/g).

The Q_m_ value for CIP significantly increased to 1575 mg/g, while Cu(II) species were present in the aqueous solution. The investigation of the reason for the enhanced adsorption capacity of the C@silica core/shell nanoparticles indicates that the silica coating increases a negative charge on the C@silica nanocomposite. So, it was suggested that the stronger electrostatic attraction between CIP and C@silica nanoparticles than that between CIP and ZIF-8-derived carbon resulted in an enhanced adsorption performance of the C@silica nanocomposite towards CIP as compared to the ZIF-8-derived carbon. These results indicate that the silica coating on the carbon-based materials can improve their adsorption performance towards CIP.

### 5.6. Using Wastes for the Preparation of MCs: π-π Stacking Interactions, Cation-π Bonding, Hydrogen Bonding

An elegant way for the “green” production and utilization of MCs for organic contaminants removal was demonstrated [370]. Taking into account that the major drawback for preparation of MCs is the use of benzene-1,4-dicarboxylic acid, which is a rather expensive reagent (H_2_bdc) as an organic ligand for many MOF-precursors, Jhung et al. [370] used H_2_bdc obtained from waste PET bottles. For this purpose, ultrasound-assisted, phase-transfer-catalyzed alkaline hydrolysis of PET was realized under mild conditions. High-purity H_2_bdc was used as an organic ligand in the synthesis of a magnetic porous carbon (α-Fe/Fe_3_C) composite derived from iron-based MOF. The as-prepared α-Fe/Fe_3_C composite comprised α-Fe, Fe_3_C, and graphitic carbon and exhibited a mesoporous structure and superparamagnetic behavior. This composite was studied as an adsorbent for the removal of TCH from aqueous solutions, and an adsorption efficiency and magnetic separation were demonstrated.

Taking into account the molecular structure of TCH and the nearly constant adsorption capacities at pH 5–7 along with XPS characterization results before and after TCH adsorption, a multiple adsorption mechanism was proposed, including π-π EDA interactions, i.e., π-electron donor in the graphite-like structure of the α-Fe/Fe_3_C composite and π-electron acceptors in four aromatic rings of TCH, cation-π bonding, i.e., π-electrons in the α-Fe/Fe_3_C composite and amino groups located in the C4 ring of TCH, and hydrogen bonding interaction, i.e., carboxyl and hydroxyl groups of the α-Fe/Fe_3_C composite and H-acceptors in the amino, hydroxyl, phenol, ketone groups of TCH.

It could be concluded that MCs have a number of properties, which differentiate them from other carbon-based nanoadsorbents, i.e., highly ordered structure, regular porous system, which is in many cases a hierarchical pore system, and rich functionality including metal, metal oxide, and N-, O- heteroatom doping. Due to the favorable porous structure, hydrophobicity, and abundant functional groups as potential adsorption sites, NPC derived from MOF matrices have been recognized as efficient adsorbents for organic pollutant removal due to their intrinsic unique properties. As to the dominating adsorption mechanism, hydrophobic interaction and hydrogen bonding play an important role in this case, especially, using MOF precursors preliminarily modified with N-enriched modifiers. Introducing a third component, such as silica, assists to further adsorption mechanism enrichment, e.g., by implementation of electrostatic interactions.

Furthermore, in order to ensure the rational design and development of the MOF-carbon composite adsorbents and MC adsorbents, it is important to evaluate particular adsorption mechanisms realized using pristine MOF adsorbents and carbon matrices, and to differentiate and compare them with adsorption mechanisms, which are implemented in the case of MOF-based composites with carbon matrices and MC materials.

The examples of organic pollutants adsorption on MOF-derived carbon materials are listed in the Table 4.

## 6. Bioadsorbents

Adsorbents derived from organic matter differ from above nano-adsorbents. However, they are also a topic of our review, because (a) they possess numerous promising properties like the possibility to change effectively their physico-chemical properties by an appropriate choice of the organic precursor, abundant functional groups, eco-friendliness (b) their advanced applications in preparing nano composites with metal organic frameworks [371,372] due to their cost-effectiveness.

Bioadsorbents (also referred to as biosorbents [373,374]) are inactive solid biogenic materials that can be used for surface retention of contaminants (adsorbates) in a solution. Bioadsorbents mainly fall under two categories: (a) microbial biomass like dead bacteria, algae, fungi, and yeast, (b) biomass from forestry, agricultural, industry, and urban origins. Depending on the target adsorbate and the selected adsorbent, bioadsorption occurs through complex mechanisms—surface adsorption, ion exchange, precipitation, diffusion, and complexation [375]. Existing research, as seen in this section, is focused on the prospects of pristine and surface-modified bioadsorbents in the removal of various classes of contaminants—heavy metals, particulates, organics, pharmaceuticals, and dye contaminants—from wastewater and aqueous solutions. The surface modification for improving the adsorption efficiency includes chemical pretreatment of biomass through hydrolysis, oxidation, or sulfonation [98]. The renewable nature and cost-effectiveness make bioadsorbents attractive compared to activated carbon [98]. However, their commercial application is restricted by various factors.

Concerning the adsorption effectiveness, bioadsorbents have limitations such as (a) non-specificity and poor selectivity, (b) requirement of expensive surface modification techniques, (c) lower adsorption capacity, (d) difficulty of regeneration, reuse, and recycling [375,376]. Reuse of any adsorbents after regeneration through desorption or decomposition of adsorbates ensures their economic viability. For instance, commercial activated carbon, depending on its type and target adsorbate, can be regenerated 6–12 times by thermal, chemical, microbial, and electrochemical methods [377,378,379,380,381]. However, biosorbents are mainly limited to chemical methods and about two to six cycles of repeated use [382] with a lower regenerative efficiency per cycle. This is mainly due to the weak mechanical and thermal resistance of parent biomass. Similarly, the lack of stability of exhausted biosorbents renders their disposal problematic. For landfilling and/or incineration, they would need to comply with strict disposal and emission guidelines to reduce terrestrial, aquatic, or aerial pollution. Valorization as fertilizers or for soil improvement is also limited due to the long-term carbon degradation in soils contributing to the active carbon cycle. Other hindrances to commercial adoption involve (a) expensive immobilization (for use in columns without pressure drops) of sorbents based on microbial biomass, (b) lack of scale-up investigations with real industrial effluents and wastewater containing a mixture of pollutants [383]. Biochar as an adsorbent can overcome or offset many of these shortcomings while retaining the benefits of biosorbents.

### 6.1. Biochar

The concept of biochar has its roots in the indigenous Amazonians’ Terra preta [384]—anthropogenic soil modifiers prepared in-situ by smoldering of agricultural wastes—they were used as soil improvers [385,386]. In modern times, biochar is prepared as the carbonaceous residues of thermochemical processes. These include (a) pyrolysis—slow devolatilization of a biogenic feedstock in an inert environment such as N_2_, (b) thermal dehydration of biomass between 200–300 °C, (c) hydrothermal carbonization—thermal treatment of moisture-rich biomass under pressure, (d) gasification—thermal decomposition of biomass in the presence of sub-stoichiometric agents like CO_2_, H_2_O, or O_2_. Among them, slow pyrolysis is the usually preferred biochar production route [387]. Biochar is different from activated carbon in the sense that it does not require physical or chemical activation processes and uses a renewable biogenic feedstock. However, this boundary tends to blur as an increasing number of studies have demonstrated the potential for activated carbon preparation from renewable biomass [388,389,390,391,392,393] and the prospects for biochar activation processes to modify its surface properties. Biochar has garnered an accelerated research focus due to its applications in adsorption of heavy metals [394,395] and organic pollutants (see Appendix A) from aquatic systems. It is a multifunctional material with other application areas including soil enrichment, carbon sinks, hydrogen storage mediums, catalysts, and fuel cells [396,397]. In non-English speaking countries like Germany, for example, the term biochar does not have an equivalent translation. Here, “biokohle”—roughly meaning biogenic charcoal—is the residue that is produced from the hydrothermal carbonization of biomass [398]. The residue derived from plant matter through the conventional slow pyrolysis process is termed “Pflanzenkohle”. This definition of “pflanzenkohle” loosely translates to “char made from plants” and abides by the guidelines of the European Biochar Certificate (EBC) [399]. This differentiation is relevant for the approval of the use and application of biochar.

### 6.2. Biochar from Refuse Biomass

Biochar can be produced from fresh and refuse biomass. Fresh feedstock includes wood and plant-based biomass that are purpose grown for production. The latter consists of agricultural waste, industrial wastes, and urban organic wastes (UOW), which have received increasing attention in environmental research (Appendix A). This is because (a) it is a renewable biogenic raw material and does not consume resources (land, energy, or man-hours) for production and (b) it presents a valorization pathway (with reduced GHG emissions, carbon sequestration, and lower risk of organic pollutants) for an otherwise discarded material thereby supplementing the regional waste management efforts.

Biochar derived from sewage sludge, agricultural wastes, and anaerobic digestates, tends to contain more minerals and ash, such as alkali and alkaline earth metals (AAEM), transition metal oxides, inorganic carbonates, and silica [400]. Such mineral and ash-rich biochar (MAB) can be favorable in certain adsorption processes. Pristine biochar from conventional lignocellulosic material can also be synthetically transformed to MAB for certain applications—magnetization with iron salts for wastewater treatment, nutrient enrichment for catalysis (during pyrolysis), and soil application. However, the term MAB is not quantitatively defined in the literature.

### 6.3. Defining Mineral and Ash-Rich Biochar

According to the International Biochar Initiative (IBI) [401], biochar must have an H/C_organic_ molar ratio ≤0.7 for an acceptable degree of graphitization and fused aromatic rings, which contribute to its porous nature. Above this limit, the biomass is not considered thermochemically converted. High-ash biochar is a material that contains more than 50% of ash (dry basis). On the other hand, European Biochar Commission (EBC) has set hard limits of ≤0.7 for H/C and ≤0.4 for O/C molar ratios [399]. Due to application restrictions, EBC also limits the feedstock to plant-derived matter [402] like wood chips, barks, vegetables, seeds, sawdust, fruits, oilseed residues, shells etc. However, the plethora of existing research (see Appendix A) has proven the prospective uses of MAB in adsorption and carbon sequestration even though concrete guidelines for their field application do not exist. The first step towards such application regulations is the question—can MAB be quantitatively defined?

The H/C atomic ratio represents the aromaticity of the matrix for MAB as the total wt.% of H is not much affected by ash and minerals. It is also inversely related [403] to the highest treatment temperature—the most influential factor in determining biochar properties such as pH, aromatic clusters, carbon recalcitrance, and surface functional groups. The H/C ratio can be indirectly correlated with the carbonization of biochar [404]. Thus, the limit of ≤0.7 set forward by IBI and EBC can persist for MAB. However, MAB may have high concentrations of minerals and ash in the form of oxides, phosphates, carbonates, and silicates. Thus, its O/C ratio will exceed the limit of 0.4 as O becomes dependent on the ash and mineral content. Furthermore, direct O measurement in the lab is expensive and not yet standardized [405,406]. Thus, it must be indirectly measured as in Equation (13):(13)O (%)=100−(C+H+N+S+A+HM)
where C, H, N, S, and A are carbon, hydrogen, nitrogen, sulphur, and ash percentages (dry basis) respectively; HM is the percent of hygroscopic moisture. Hence, the O/C atomic ratio must be supplanted. A suitable replacement candidate is the A/C ratio. The mineral and ash fraction that constitutes the ash content influences numerous biochar properties. Determination of the ash content is also quick and standardized. It also adjusts in cases where inorganic recalcitrant carbon is present in the form of carbonates. A/C is the ratio of the ash and mineral fraction (intrinsic or enriched) to the total concentration of carbon in MAB. However, defining the limit of the A/C ratio depends on the intended biochar application. For instance, a MAB with higher concentrations of AAEM may be beneficial for certain types of plant growth in soils [407,408]. But, this will reduce the net carbon sequestered per unit weight of biochar produced.

From the sampled literature data in Appendix A, for biochars with the H/C ≤ 0.7, A/C ratio have a sample mean and sample standard deviation (s_x_) of 1.18 and 2.04, respectively. Hence, 2s_x_ (i.e., s_x_~4.1) can be considered as a general A/C limit for MAB. Then, MAB can be defined through a modified form of the van Krevelen diagram (Figure 26) with the H/C molar ratio as the ordinate and the A/C ratio as the abscissa. Biochar in the region enclosed by H/C ≤ 0.7 and A/C = 4.1 may be defined as MAB. For biochar application, as in adsorption, future studies can also model their variables of interest (e.g., pH, adsorption efficiency or pore surface area) as a response surface dependent on H/C and A/C as these ratios comprise the most important parameters affecting MABs’ physico-chemical characteristics.

### 6.4. Factors Determining Adsorption Potential

The physio-chemical characteristics of biochar are primarily influenced by independent variables such as lignocellulose in parent feedstock, process parameters (highest heating temperature (HTT), heating rate, residence time), mineral and ash fraction (MAF). They determine the properties such as cation exchange capacity (CEC), surface charge, aromaticity, organic functional groups, and pH that govern adsorption mechanisms. The process of adsorption with biochar also depends on the adsorbate. Heavy metals and inorganic pollutant adsorption are driven by surface adsorption; ion exchange; precipitation; complexation with –O functional groups and phenolic groups; electron donor and acceptor (EDA) interactions with aromatic structures; electrostatic interaction between the biochar surface and metals [409]. The predominant mechanism for organic pollutant adsorption is partitioning hydrophobic interactions and volumetric sorption (pore filling) [410]. In short, different physico-chemical factors can be influential during the adsorption depending on the adsorbate and the mechanism involved. Adsorption of certain pollutants like Pb and Cd is improved in biochar functionalized with iron oxides (magnetic biochar). They also exhibit better thermal regeneration (increase in the number of regeneration cycles and extent of pollutant degradation) after tetracycline removal in wastewater [410].

#### 6.4.1. Highest Treatment Temperature (HTT)

The term HTT can be defined as the peak temperature at which the pyrolysis reaction is performed. It has been shown to have the most influence on biochar properties and adsorption of contaminants compared to other process variables such as heating rate and residence time [407,411]. HTT affects properties like the ash content, fixed carbon, volatile matter, aromatic condensation, pH, CEC, surface area, and porosity. It tends to increase the surface area (mesopores and micropores) of biochar. The surface area and porosity are influential factors for adsorption of anionic pollutants like NO_3_^−^ and PO_4_^3−^ [412,413] and 2,4-D, in cases where physical adsorption is the main mechanism. Macropores (0.3 to 30 μm) developed in biochar also act as habitats for microbes when they are deposited in soil [414]. However, for a certain feedstock, the surface area and porosity can start to reduce when HTT is raised beyond a certain temperature. This thermal shrinkage is attributed to the collapse of the porous structure or clogging by tars [415] from softening and melting at high temperatures. Thermal shrinkage always depends on the type of parent feedstock. Similarly, high HTT can also lead to lowering of the O/C molar ratio of the non-ash portion of biochar. This lack of oxygen functional groups can reduce the sorption of metals through complexation [410].

#### 6.4.2. Polarity

Carboxylic, amino, and hydroxyl groups present in biochar can enhance metal sorption. The functional groups present in biochar also result in its surface polarity, which is denoted as a polarity index, the (N+O)/C molar ratio. The polarity of biochar is dictated by its O containing acidic functional groups. These groups, owing to devolatilization, are eliminated with HTT and are mostly lost after 600 °C. Hence, biochar polarity is inversely proportional to its HTT. The polar functional groups prevent the formation of water clusters through hydrogen bonding, which in turn hinders the sorption of hydrophobic molecules at the hydrophobic sites on the biochar matrix [416]. As a result, low polarities at high HTT support the sorption of hydrophobic contaminants like trichloroethylene and sulfamethoxazole. However, higher polarities favor the selective mechanism of adsorption of polar organic contaminants [411,417]. Thus, the influence of functional groups on the adsorption mechanism is only substantial in biochars produced below 600 °C [415].

#### 6.4.3. Carbonization

Carbonization at higher HTTs also influences the adsorption mechanism. Binding of –N functional groups (in biochars derived from nitrogen-rich feedstock) to aromatic rings can grant them better ion selectivity [418]. The aromatic carbon in biochar becomes an electron acceptor at HTT ≤ 500 °C and an electron donor at HTT ≥ 500 °C [409]. Thus, π-electrons of the biochar aromatic C matrix contribute to the EDA interactions that are responsible for the sorption of aromatic compounds. Biochar produced at higher HTTs also reduces the soluble fraction of heavy metals, suppressing their bioavailability in soils [419]. Carbonization of biochar increases the electrostatic interaction between the negatively charged biochar matrix and the anionic moieties containing heavy metals like Cd, As(V), and Cr(VI), thereby increasing their respective removal efficiencies. Reduction in organic functional groups and higher carbonization at higher HTTs consequentially increases the ash and mineral content in the resultant biochar.

#### 6.4.4. Mineral and Ash Fraction (MAF)

MAF in pristine biochar can be increased (a) at higher HTT and (b) by doping/enrichment processes [420]. Being feedstock-dependent, it is also inherently present in biochar derived from industrial wastes, sewage sludge, poultry litter, diary manure, and waste biosolids compared to those from traditional lignocellulosic feedstock as seen in Appendix A. Such biochar tends to demonstrate higher yields during pyrolysis due to recalcitrant SiO_2_ and inorganic carbonates.

Non-water soluble AAEM cations—Ca^2+^, Mg^2+^, K^+^, Na^+^—present in MAB improve adsorption of heavy metals and phosphates [421] through cation exchange. While anions—OH^−^, CO_3_^2−^, PO_4_^3−^ and SO_4_^2−^—in MAB produced at higher temperatures (~750 °C) support adsorption of heavy metals through precipitation [422]. A higher ash content also leads to formation of mesopores, which enables sorption of smaller organic molecules [404]. However, a higher silica content, such as those present in sewage sludge-derived MAB, lowers the total available porous area. They can also block the pores already developed in the carbon matrix. Thus, they hinder the adsorption potential of MAB for certain adsorbates. An example of such biochar is shown in Figure 27.

In such cases, chemical deashing and/or washing with water can improve the performance of sorption sites [423]. Similarly, deashing can also improve the chemical activation of biochar [422] by allowing better contact between reagents and the biochar matrix. The main mechanism of sorption in MAB is surface precipitation. The ash content is shown to have a positive correlation with surface polarity, which can enable better removal of organic pollutants [404]. Similarly, MAB [424,425] that is rich in phosphorus, an important macronutrient, can help in phosphorus enrichment of arable soils—about 5.7 billion ha of arable soils are deficient in plant-available P. Recycling P from waste biomass is also known to transform lead in soil to pyromorphite and Pb-phosphates, which immobilizes bioaccessible Pb. The effects of MAB on the crop yield depend on the pyrolytic conditions and the soil pH value [407]. The presence of minerals, on the other hand, can make them less recalcitrant in the atmosphere as compared to regular biochar.

The MAF in biochar can also be increased through doping/enrichment and chemical activation. Activation, in general, can be categorized into physical and chemical activation. It aims to increase the surface area and pore volume, minimize volatile carbon, and enhance adsorption capabilities [426]. Steam and CO_2_ activation are the common physical procedures [ibid]. It also improves the cation exchange capacity by exposing the AAEM content of the biochar matrix. However, it does not alter the chemical structure of the biochar matrix. Steam activation of biochar derived at lower HTT can preserve oxygenated functional groups without sacrificing the micropore volume [427]. Chemical activation can be done via agents such as KOH, NaCl, H_3_PO_4_, ZnCl_2_. KOH inserts metallic K into the carbon lattice and enlarges the biochar porosity [426]. H_3_PO_4_ introduces –P functionalities into the biochar matrix. Mineral enrichment/doping [428] is the process of impregnating the carbon lattice of the biochar with N, P, S, or metal oxides to improve its surface functionalities or for specialized applications like magnetic biochar for wastewater treatment. Biochar doped with metal oxides is also referred to as biochar-based composites [429]. Doping does not require temperatures as high as those used in the activation methods, and the enriched foreign compounds become a part of the biochar matrix unlike in chemical activation where a portion of a reagent is recovered and/or gets emitted as non-condensable vapors [ibid]. However, another review paper classified doping/enrichment as also an activation method [396]. Furthermore, it is worth noting that after activation, by definition, biochar becomes activated carbon. Thus, the boundary between activated biochar and activated carbon is fuzzy. Nevertheless, some studies based on the widespread industrial use, consider coal or non-renewable sources to be the precursors of activated carbon to delineate the differences between activated biochar and activated carbon [430,431].

Biochar enrichment with iron oxides (hematite/maghemite/magnetite) can improve the porosity of its matrix and iron oxide can act as a source of electrostatic adsorption sites increasing the cation exchange of biochar [409]. Some studies have reported a reduced sorption capacity of biochar due to pore clogging and structural collapse [418]. However, impregnation with metal oxides can improve the sorption of pollutants—like PO_4_^3−^—with pH-dependent binding mechanisms [429]. These differences in the effects of incorporation of iron salts can depend on the type of enrichment—in situ (co-pyrolysis) or post-processing (chemical co-precipitation). Doping with CaCO_3_ in rice straw magnetic biochar has shown higher Cd(II) sorption due to a charge differential from the increased negative charge on the biochar surface [432]. Calcium carbonate is otherwise naturally present in certain types of MAB derived from sludges and digestates. Therefore, further investigations into sorption using such MAB are required.

Another process that increases the net MAF in biochar is aging—a series of biochemical reactions in biochar when it is mixed with soil. Effects of aging depend on soil and biochar type. It is shown to reduce the biochar sorption potential for organic pollutants because of (a) pore blocking by sorption of dissolved organic matter (DOM) from soils, (b) increase in hydrophilicity due to inhibition by soil minerals [420]. Instances of increased adsorption have also been noted such as (a) sorption of Cd increased during aging due to the introduction of carboxylic groups that created more binding locations on the biochar matrix, (b) on manure-derived biochar (produced at 700, sorption of phenanthrene increased because of the DOM in soil binding with biochar’s Ca^2+^ and Mg^2+^ cations increasing the efficiency of sorption sites on the exposed DOC [433].

#### 6.4.5. pH

As seen from Appendix A, the pH value of biochar is usually alkaline and depends on the type of parent feedstock. It also increases with HTT owing to an increase in the content of carbonates (from the ash content) and the absence of acidic functional groups [415]. An increase in pH leads to higher sorption capacity for metal cations and organic contaminants such as dyes. This arises from the deprotonation of functional groups and better interaction between negatively charged sites on the biochar surface and positively charged dyes, respectively [434]. However, in some cases, where the pH increase is caused by higher HTT, the sorption capacity can decrease due to the collapse of the porous structure [435]. In short, the impact of pH varies with the target adsorbate and the biochar properties. The pH value of biochar is a dynamic property, since it can vary due to the sorption of contaminants like heavy metals during the adsorption process [436]. HTT, MAF, and solution pH determine the adsorption mechanism, kinetics, and removal efficiency. Due to the pyrolysis complexity and feedstock heterogeneity, especially in MABs, these factors and their interrelations must be established on a case-by-case basis for each type of the adsorbent and adsorbate.

### 6.5. Adsorption Mechanism

Biochar reveals a significant ion exchange capacity, which is lower or absent in activated carbon. This beneficial cation exchange property is due to the residual carboxylic acid functionalities on its turbostratic carbon network.

According to [437], a few biochar applications were related to wastewater treatment, and none of them have purely mentioned heavy metal isolation from liquid streams. This section will be reviewing the biochar adsorption mechanism (e.g., kinetics and isotherms), concerning the removal of heavy metals from aqueous solutions. The biochar adsorption mechanisms for removing heavy metals and organic compounds are shown in Appendix A. An overview of various materials-derived biochars with the production characteristics and biochar adsorption applications is shown in Appendix A.

According to [438], the key components affecting the biochar adsorption capacity in aqueous solutions are input factors of the biochar production process (e.g., temperature of pyrolysis, feedstock materials) and operating parameters of the metal adsorption process (e.g., pH, temperature, biochar dose, and initial solute concentration). Various biochar adsorption mechanisms are led by its different surface functional groups and electrical properties, as well as its organic structure [439].

The study of biochar adsorption mechanism has mainly been focused on three widely employed kinetic models: pseudo-first order (Equation (14)), pseudo-second order (Equation (15)) and intra-particle diffusion (Equation (16)) to illustrate a well-defined adsorption process [440], as described below:(14)$$ln\left(q_{e}-q_{t}\right)=lnq_{e}-k_{1}t\;$$
(15)$$\frac{t}{q_{t}}=\frac{1}{k_{2}q_{e}^{2}}+\frac{t}{q_{e}}\;$$
(16)$$q_{t}=k_{p}t^{\frac{1}{2}}+C_{i}\;$$
where t is the contact time (min); $q_{t},\;q_{e}$ are the amount of an adsorbate at a given period of time *t* and at equilibrium (mg/g), $k_{1},\;k_{2}\;\left(\frac{g}{\mathrm{mg}\times \min}\right),\;k_{p}\left(\frac{\mathrm{mg}}{g\times \min^{\frac{1}{2}}}\right):$ are the adsorption rate constants of pseudo-first order (min^−1^) for three kinetic models, respectively, $C_{i}:$ thickness constant of the boundary layer. The adsorption process can be defined according to the regression coefficient values of the kinetic models. For instance, a higher value of the regression coefficient from the pseudo-second order compared to that of the pseudo-first order suggests a chemisorption process. Kinetic models that incorporate the interdependencies of inorganic and organic pollutants on the adsorbents require further investigations [410]. This can aid in understanding of the adsorption mechanisms of MAB and biochar used for treatment of leachate water.

To conclude this section, the reason for classifying biochar as advanced type of carbonaceous adsorbents are as follows: (a) its extremely environmentally benign nature, (b) low cost of production of MAB and (c) abundant functional groups contributing to the realization of different adsorption mechanisms. The main drawback is their relatively lower adsorption potential compared to nanomaterials. However, biochar can be used for synthesis of composites with other carbon-based nanoadsorbents considered in this review and MOF crystallites to improve the selectivity and to lower the cost of synthesis of the latter. Such composites also provide a potential pathway for disposal of spent adsorbents as soil improvers after further pyrolysis.

## 7. Conclusions and Future Outlook

This review summarizes and discusses the contemporary literature on modern carbon-based adsorbents, such as GO, CNTs, biochars, MOF-carbon composites and MCs obtained by controllable pyrolysis of MOF precursors. To the best of our knowledge, the relevant reviews deal separately with different kinds of carbonaceous (nano)adsorbents and biochars. The core distinction of this work as compared with prior literature relates to an attempt to reveal peculiarities and compare the adsorption performances to answer the question “why these carbon-based materials can be defined as advanced adsorbents?” and reveal their advantages over traditional adsorbents like activated carbons. Such consideration of these adsorbents may stimulate the design of advanced adsorbents. This review also outlines the “structure-adsorption properties” relationship for the discussed nanoadsorbents.

The analysis of contemporary literature shows that the recent progress and developments in water pollution remediation are mainly based on adsorption processes as well as combination of the adsorption method with other traditional techniques. The efficient adsorptive removal of organic and inorganic pollutants involves to a significant extent the utilization of modern carbon-based adsorbents. All these materials have a highly ordered structure (excluding biochar), intrinsic high porosity, and surface/pore functionality, which can be further modified by introducing special functional groups or through the preparation of their functional composites with other types of materials including inorganic solids, metal oxides, MOF crystallites and heteroatom dopants and agents, such as melamine and ionic liquids. These features allow one to utilize them for numerous special tasks in the adsorptive removal of a wide spectrum of organic and inorganic pollutants. It is important to note that these carbonaceous nanomaterials show multifunctionality and multipurpose utilization in several environmental application fields, such as fuel denitrogenation and desulfurization, so their development offers new promising opportunities for removal of both organic and inorganic pollutants from aquatic media. The main conclusions from this review can be summarized under three fronts: (a) synthesis strategies of advanced carbon-based nanoadsorbents, (b) effectiveness of their composites and modification/doping methods, and (c) biochar which can be a potential composite partner for MOFs.

The general synthesis strategies for each type of carbonaceous adsorbents are presented with a special emphasis of the impact of utilized preparation procedure on their specific physico-chemical characteristics, i.e., structural, textural, compositional and adsorption properties.

An appropriate choice of synthesis technique including process parameters (temperature, time) along with pretreatment/activation/modification mode may result in a serious improvement of their adsorption performance towards both organic and inorganic pollutants in aqueous media. In several cases, an appropriate temperature regime for calcination of the prepared carbon-based materials may result in a serious improvement of their porosity and hydrophobicity.The different strategies for the improvement of the textural (specific surface area, porosity), morphological characteristics as well as pore surface functionality and thereby their adsorption performance are presented. The most effective strategies for this purpose are pore surface modification by oxidation, hydroxylation, grafting appropriate functional groups, doping heteroatoms (N, S, O) or metal species as well as preparation of their functional composites by integration of different kinds of (nano)adsorbents or classical carbon or inorganic solids like silica.Appropriate processing, shaping/pelletization, preparation of carbonaceous (nano)adsorbents, e.g., CNTs, GO and MOF-Carbon composites, in the form of aerogel or hydrogels are promising to increase the separability, mechanical strength, and stability of these adsorbents in aqueous media. Additionally, using binding agents, such as sodium and calcium alginates may introduce additional heteroatoms in resulting materials and thereby enrich the possible adsorbate-adsorbent interactions.

The cornerstone of the rational design and development of an adsorbent of choice and optimal adsorption process is essential to insight in the possible adsorption mechanisms that involve many interactions between adsorbates and adsorbents and to manipulate with these interactions in a predictable way. So, the special emphasis in this review is placed on the analysis of different adsorption mechanisms or their combination realized using them in water purification by removal of organic and inorganic pollutants. The specific adsorbate-adsorbent interactions and possible adsorption mechanisms are considered for each type of presented carbonaceous nanoadsorbents along with different strategies to administer and enrich them, such as pore surface functionalization, loading of the modifying/doping agent, fabrication of composites, etc., to enhance the adsorption performance. For instance, modification of MOF components with amino, hydroxyl or carboxylic groups may result in drastic enhancement of the adsorption capacity towards dyes and pharmaceuticals for their composites with carbon-based materials. So, a strong correlation between physico-chemical characteristics of a particular kind of carbon-based (nano)materials and their adsorption properties can be concluded.

The effectiveness of the adsorption performance of the discussed carbon-based (nano)adsorbents is illustrated by numerous examples of removal of almost all known POPs and inorganic contaminants. The comparison of the adsorption performances of the considered carbon-based (nano)adsorbents towards different organic pollutants with rather classical materials such as activated carbons is presented. In many cases, they demonstrate higher adsorption efficiency (up to ~99%), adsorption capacity and favorable kinetics as compared to conventional adsorbents.

An appropriate choice of specific carbon-based (nano)adsorbents allows one to resolve almost all problems related to water purification due to involving both physisorption and chemisorption processes, trigger a specific mechanism, or more often, their combination in a controllable way.

Noteworthy, the relationship between selected kinds of the modern carbon-based nanoadsorbents and biochar lies in the possibility to integrate them in functional composites. The possibility of realization of synergetic adsorption performance implemented due to using bi- or three-component composites based on considered carbon (nano)materials is also discussed here.

As a rule, these composites show significantly enhanced pollutant uptake and adsorption efficiency (higher than 99%) as compared to pristine composite components and common adsorbents like activated carbons, so this strategy provides the synergetic adsorption performance due to several cooperative effects provided by intrinsic porosity, functionality, and adsorption properties of composite components.Using the functional composites of carbon-based nanoadsorbents with other materials allows one to involve both adsorptive removal and other related processes, such as (photo)catalytic degradation in several purification cases. For instance, a combination of adsorption and photodegradation may be recommended for the removal of dyes such as methyl orange from wastewater using selected MOF-carbon composites, such as MOF-MWCNTs and MOF-biochar systems.Generally speaking, each type of considered carbon-based nanoadsorbent has several unique properties, which can be further modified, so in each case, an appropriate choice of carbon-based nanomaterial is governed by the needs for specific tasks related to organic pollutant removal.

In this context, an integrating approach is recommended, which involves an improvement of physico-chemical characteristics and adsorption properties of the modified materials or composites. For instance, embedding of an appropriate content of GO particles in the MOF matrices can improve the surface area, porosity by generation of new pores on the interface of the components, dispersion by preventing aggregation of GO particles and enrich the functionality of the resulting GO-MOF composites by a combination of functional groups of pristine composite components.

In some cases, a favorable interaction of adsorbates is demonstrated in the literature. An example is a simultaneous removal of U(VI) and humic acids using MOF-MWCNTs composites. In this case, coexistence of humic acids assists in enhanced U(VI) removal from environmental water. However, the interdependence of organic and inorganic impurities in the adsorption processes has not yet been investigated and understood in detail.

In this context, MOF matrices show a special potential as composite components or precursors of nanoporous carbon hybrids and composites doped with heteroatoms (N, S) and/or metal nanoparticles or oxides, due to their extremely high porosity, specific surface area, and rich and versatile functionality that allows one to enrich the adsorption mechanism and modulate the nanoadsorbent hydrophobicity/hydrophilicity. Moreover, using MOF materials as composite components with other carbon materials or precursors for MCs allows one to accomplish the water purification in a green, ecological, and economical way by utilization of plastics (such as PET) wastes for MOF synthesis. The next possibility is pyrolysis of the spent MOF-carbon nanoadsorbent, for instance, a ZIF-8-MWCNTs composite [441] studied as an electrode, followed by utilization of carbonaceous C@C hybrids for adsorptive removal of hazardous pollutants distributed in water in a new way or other environmental application fields, such as catalysis and energy storage/conversion.

The reason for classifying biochar as advanced type of carbonaceous adsorbents is its extremely environmentally benign nature and abundant functional groups contributing to the realization of different adsorption mechanisms.

Based on existing biochar standards and literature review, mineral and ash rich biochar (MAB) has been defined as those with H/C ≤ 0.7 and Ash/C ≤ 4.1. Mineral and ash fraction along with the highest treatment temperature in pyrolysis are the most influential factors in determining biochar adsorption mechanisms. Valorization of MAB derived from mineral and ash-rich biomasses like urban organic wastes as adsorbents for organic and inorganic pollutants is recommended for further investigation and critical research.Another area of current and future research is appropriate functionalization or integration of biochar with other types of adsorptive materials, like MOF crystallites to enhance their adsorption performance (order, textural and functional properties), lower production costs and create recyclability as soil improvers. Moreover, biochar along with other carbon-based nanoadsorbents considered in this review may be regarded as a multifunctional and multi-purpose material in several ecological applications, such as soil amelioration.

In this context, further investigation in the direction of (a) tuning and optimization of physico-chemical characteristics and adsorption properties of biochar for appropriate functionalization and/or integration with other types of adsorptive materials, like MOF crystallites, and (b) using the effect of heteroatoms inherited from MAB in MAB-MOF matrices are warranted.

Thus, it could be concluded that the rational design and synthesis of novel carbon-based nanomaterials of different kinds as well as their functional composites and hybrids paves the way to successful development of modern adsorbents based on them with advanced adsorption efficiency as high as 99.96% achieved mostly in 12 h towards a wide spectrum of organic pollutants including their bi-, tri- and ternary mixtures or simultaneous removal of organic and inorganic pollutants in drinking and leachate waters.

## Data Availability

Not applicable.

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
