# Peer review of "Modern Carbon–Based Materials for Adsorptive Removal of Organic and Inorganic Pollutants from Water and Wastewater"

_molecules, 2021, doi:10.3390/molecules26216628_

Round 1
Reviewer 1 Report
The submitted review entitled " Modern Carbon–Based Materials for Adsorptive Removal of Organic and Inorganic Pollutants from Water and Wastewater " has discussed an interesting issue for using modern carbon-based materials for adsorptive removal of water purification. The authors have done a lot of work, and this manuscript gives a very detailed summary. However, a major revision is required before publication.
1/Abstract: Include your recommendations and future prospects.
2/Please be sure that your abstract and your Conclusions section not only summarize the key findings of your work but also explain the specific ways in which this work fundamentally advances the field relative to prior literature.
3/The significance of this study should be more emphasize in the introduction.
4/It is suggested to present the structure of the article at the end of the introduction. Also there are some problems in the structure of the manuscript:
1) There is no need for “2.1. Adsorption isotherms”, due to there is only one aspect.
2) “3.1.5 π-π. Interactions” and 3.1.6 to 3.1.8 should be under “3.1.4. Mechanism of adsorption of organic contaminants onto CNTs” as 3.1.4.1, 3.1.4.2, …….
3) “4.6.1. Electrostatic interactions, hydrogen bonding, π-π interactions, open metal sites complexation, soft-soft and dipole-ion interaction” should be divided into 4.6.1, 4.6.2, …..
5/A flowchart should be added to the article to show the research methodology.
6/Line number is very important for the review, so please add.
7/In my opinion, this manuscript is too long, and it needs to be simplified and highly summarized as much as possible.
8/The quality of the tables needs to be further improved, too many contents made it a little disordered, especially the table from page 72-77.
Author Response
September 03, 2021
MDPI Molecules
Manuscript ID: molecules-1301684
Title: "Modern Carbon-Based Materials for Adsorptive Removal of Organic and Inorganic Pollutants from Water and Wastewater"
Author(s): Vera I. Isaeva*, Marina Vedenyapina*, Alexandra Kurmysheva, Dirk Weichgrebe, Rahul Ramesh Nair, Ngoc Phuong Thanh Nguyen, Leonid M. Kustov*
Dear reviewers,
Thank you very much for your helpful and valuable comments and advices. According to your instructive remarks and suggestions, we have done a significant revision of our manuscript. We added also the Supplementary Material file with Tables i, ii, iii, which are Tables 5, 6, 7 in the first submitted version of the manuscript. The Revised manuscript was supplemented by new references 155, 163, 276, 397, flowchart (Scheme 1), Scheme 2 (synthesis of GO) and Figure 2 with structures of selected pharmaceuticals (as organic pollutants) as well as Content, Nomenclature list and Acronym list. Below, we indicate how the referee’s comments have been taken into account in the novel revised version of the manuscript.
Reviewer Comments to Author:
Reviewer 1
The submitted review entitled "Modern Carbon–Based Materials for Adsorptive Removal of Organic and Inorganic Pollutants from Water and Wastewater" has discussed an interesting issue for using modern carbon-based materials for adsorptive removal of water purification. The authors have done a lot of work, and this manuscript gives a very detailed summary. However, a major revision is required before publication.
1/Abstract: Include your recommendations and future prospects.
Response: According to this suggestion, some extracted recommendations and short future outlook have been added to the Abstract of the Revised manuscript.
2/Please be sure that your abstract and your Conclusions section not only summarize the key findings of your work but also explain the specific ways in which this work fundamentally advances the field relative to prior literature.
Response: According to this suggestion, the core distinctions of the present review relative to the precedent works are highlighted in Abstract and Conclusion Sections of the Revised manuscript.
3/The significance of this study should be more emphasize in the introduction.
Response: According to this suggestion, the contribution of this review in the relevant topic is outlined in the Introduction section of the Revised manuscript.
4/It is suggested to present the structure of the article at the end of the introduction. Also there are some problems in the structure of the manuscript:
Response: The structure of the Revised manuscript was corrected according to this suggestion.
1) There is no need for “2.1. Adsorption isotherms”, due to there is only one aspect.
Response: The subtitle 2.1. was removed. Only title for Section 2 is kept in the Revised manuscript.
2) “3.1.5 π-π. Interactions” and 3.1.6 to 3.1.8 should be under “3.1.4. Mechanism of adsorption of organic contaminants onto CNTs” as 3.1.4.1, 3.1.4.2, …….
Response: The subtitles for Section «3.1.4. Mechanism of adsorption of organic contaminants onto CNTs» were renumbered according to this Reviewer’s comment. Analogously, the subtitles were renumbered for Section «3.2.3. Mechanisms of the organic pollutants adsorption onto GO».
3) “4.6.1. Electrostatic interactions, hydrogen bonding, π-π interactions, open metal sites complexation, soft-soft and dipole-ion interaction” should be divided into 4.6.1, 4.6.2, …..
Response: According to analyzed literature, it is really difficult to highlight the dominating adsorption mechanism in case of the MOF-AC composites. As a rule, the adsorption is based on a combination of different adsorption mechanisms involving various interactions. So, the title of Section 4.6.1 was removed in the Revised manuscript.
5/A flowchart should be added to the article to show the research methodology.
Response: The relevant flow chart was added to the Introduction Section as Scheme 1 in the Revised manuscript.
6/Line number is very important for the review, so please add.
Response: Sorry for this inconvenience. Line numbering was inserted in the Revised manuscript.
7/In my opinion, this manuscript is too long, and it needs to be simplified and highly summarized as much as possible.
Response: Thank you for this instructive comment. According to this Reviewer suggestion, some relevant corrections regarding shortening were made in Sections 3, 4 and 5 of the Revised manuscript. Additionally, Tables 5-7 were moved from the manuscript to the Supplementary Information. However, according to suggestions of referees 2, 3, 4, 5, some additional fragments with relevant discussions were added to the Revised Manuscript.
8/The quality of the tables needs to be further improved, too many contents made it a little disordered, especially the table from page 72-77.
Response: According to this Reviewer’s comment, Tables were corrected. Tables 5-7 from the manuscript to the Supplementary Information as Tables i, ii, iii.
Reviewer 2 Report
Review for molecules-1301684
I have only read the first three sections of the manuscript. On page 9 the number of comments was closing in on 60! These comments can be traced as numbered items #i on the attached pdf of these first pages that also include additional questions marks or suggestions for linguistic changes.
After the detailed comments I will draw a conclusion and make a recommendation.
#1 water contamination means water that comes into the environment? There should be examples for the organic pollutants as well, as for the inorganic pollutants.
#2 metal ions are no organic pollutants
#3 there is a distinct difference in the kind of pollution between rural and town environments, between industrial areas and natural environments, between first world and third world countries. This could be detailed here. Also with respect to the technological and infrastructural options to clean water.
#4 is underground meaning groundwater? Please specify a bit more.
#5 it is not wastes, it is the organic products or what substances that even may enter the environment after having passed through human or animal bodies (like antibiotics or other medications).
#6 highly hydrophobic? How is the degree of hydrophobicity defined. The most hydrophobic substances will accumulate at water/air interfaces and are then not really dissolved. They will tend to any kind of surface as well to lower surface energies.
#7 it is not high content only, it can be traces that are very harmful.
#8 what are the conventional methods, give examples
#9 if you use alumina, it is already a solid particle and cannot be formed anymore
#10 flocculants are additions that cause flocculation. I think it is typically not solid particles. It is rather solutes, like Al or Fe that will first be solutes before they form solids. It is not precisely described here.
#11 what is the diffrerence between coagulation and flocculation? I think they are connected. First you form the solids and then these solids aggregate and will sediment or can be taken off in some other way. So it is even connected to sedimentation. I know this as a suite of processes. One distinct process will not be useful.
#12 The combinations of methods is not mentioned. This has to be the future given the diversity of contaminants.
#13 what are relatively mild conditions here?
#14 uptake depends on pH and many other things as the authors claim later. So how can you define something like an adsorption capacity. It is a term that is often used and gives the impression of describing the capability of an adsorbent to bind something. However, it is never known what conditions this refers to and it can be entirely different if you just change the pH.
#15 in water cleaning if this is the general method, there cannot be target pollutants, but everything that has the potential to cause damage has to be removed (see #12).
#16 it is not clear how this all is done or expected to be operational. It could be discontinuous (batch-type) but also in a column type arrangement. It should be discussed at some point.
#17 please describe shortly what you understand by carbonaceous adsorbents. For some it could include carbonates, but this is probably not the case here.
#18 I got lost here. Why would you classify it this way. Chemsorption could be covalent as well. Ionic interactions could be associated to electrostatics, but this is rather physical adsorption. Where does the H-bonding fall in. Classify the mechanisms which you later discuss in detail here.
#19 this is ambiguous, one might understand that alumina and the rest are also biosorbents. Then mesporous is not the same class of material as alumina or zeolites.
#20 you discuss solids and these cannot have a pH. Only in water or solution will they cause a certain pH.
#21 why not you AC after you have defined it.
#22 what are these unsatisfactory adsorption characteristics etc.? without more detail the reader is lost, because before AC has been advocated.
#23 please give more detailed examples for the nano-engineered adsorbents and the classical adsorbents. Also say something about cost of making them and availability.
#24 shape? Nanotubes, rods and wires are nanoparticles. So you could say nanoparticulate adsorbents… also nanocomposites are not in the same class. It is not clear whether they would be mixtures of tubes, rods and wires for example or if they have a composite chemical structure.
#25 in this introduction I would have expected something about the role of pH. What is the range of pH for which reasonable cleaning operations can be carried out? Then there is pH effect on uptake of many solutes. Also water composition may play a role (complexation keeping metal ions from being adsorbed for example). Finally temperature plays a role and the trends could be introduced in this introduction.
#26 These kinds of models cannot treat multicomponent systems in a straightforward ways. The cannot intrinsically handle variation of pH and effects of charge. Surface complexation models exist to allow all this. Why not advocate those?
#27 I am not sure whether the graphene oxide is really inherently hydrophobic.
#28 which types are you talking about? It only becomes clear later.
#29 why not put the simple one before the complex one?
#30 explain how the MWCNTs can keep their structure? i.e. why do the tubes not simply separate to form SWCNTs?
#31 many surfaces will be negative (as stated later also by the authors) and many organic pollutants are rather negative than positive as well. So overall the surface charge will not help adsorption.
#32 can surface charge have properties?
#33 what about release of components of such sorbents? Often the additions might become pollutants.
#34 header says organic, now we included inorganic
#35 how? More amino than carboxylate? pH? Overall instead of cumulative?
#36 again this depends on pH and nothing is said about ranges of charge and pH.
#37 dispersion and particle size is something different. Good dispersion may be obtained for any size as long as sedimentation is avoided.
#38 Two of them are no acids to me.
#39 again, what conditions? Is it continuous or batch-type? What are the quantities.
#40 what if such more loosely bound components of functionalized sorbents are released and cause harm?
#41 again a new component. There are many ILs and many contain not so nice components. My idea would be to keep anything potentially harmful away. I am not sure abou the imidazole for example in this context.
#42 order for what? Adsorption of what?
#43 what adsorption values? Maximum uptake, Langmuir constants?
#44 it could be anything. But typically it is a mixture of initial interaction and subsequent transport properties.
#45 in the real world waters, there will be multicomponent solutions with competition for adsorption sites. There might also be hindrance of bigger molecules blocking pores or reactions in the bulk solution that change the system from the single component lab investigation.
#46 EDA is not defined. Later I could guess what it means.
#47 I would guess both are at play.
#48 not clear if this refers to solute or surface or both
#49 IR spectra in the H-bonding region can be messed up by water bonds. I do not know to what extent this community takes this into account. I would hope they do.
#50 H-bonding is typically on the weak side. So how tightly bound is a pollutant and can it be easily released? It raises again the question about the operation (batch vs flow through). Also in multicomponent systems strong adsorption of one component may impede the H-bonding of another.
#51 potentials not forces. Maybe forces between adsorbents or between solutes and adsorbents.
#52 This is what I mentioned before. With the negative charge there is repulsion concerning anions and many organic solutes are anions.
#53 what are all these compounds (give strutures or at least say to which class of pollutant they belong)?
#54 Be careful to state the conditions in more detail, like temperature. Also structures could be given for the molecules, so the reader can get an idea about functional groups.
#55 this is often a misunderstanding or a wrong conclusion. Electrostatics may help, but often it is just a part of the story. And again without knowing a bit more beyond the name of the compound it is difficult to follow.
#56 qe is often in moles/kg. so it is not concentration, please check.
#57 it suggests. There is no proof.
#58 reference to table 1 is too late here I guess.
Also table 1 could include columns for
- pH and concentration ranges in all cases (you might exlude cases where it is not specified)
- overall charge of the sorbent at the operational pH
- uptake in moles/m2 as well for better comparison
- removal in percentage
I guess those numbers give much more information than mg/g. With the specific surface area it is easy to see that the per mass is misleading given that adsorption is a surface phenomenon.
With so much problematic wording and a text that is hard to follow and not so nice to read also because it is not really synthesizing information, I decided not to continue reading on page 9. The authors should have a look at articles in Chemical Reviews for example to get an idea about how to design such reviews.
Many of the statements are so general, lacking examples or ranges that the reader does not learn anything.
I cannot recommend it for publication in its present shape.

Author Response
September 03, 2021
MDPI Molecules
Manuscript ID: molecules-1301684
Title: "Modern Carbon-Based Materials for Adsorptive Removal of Organic and Inorganic Pollutants from Water and Wastewater"
Author(s): Vera I. Isaeva*, Marina Vedenyapina*, Alexandra Kurmysheva, Dirk Weichgrebe, Rahul Ramesh Nair, Ngoc Phuong Thanh Nguyen, Leonid M. Kustov*
Dear reviewers,
Thank you very much for your helpful and valuable comments and advices. According to your instructive remarks and suggestions, we have done a significant revision of our manuscript. We added also the Supplementary Material file with Tables i, ii, iii, which are Tables 5, 6, 7 in the first submitted version of the manuscript. The Revised manuscript was supplemented by new references 155, 163, 276, 397, flowchart (Scheme 1), Scheme 2 (synthesis of GO) and Figure 2 with structures of selected pharmaceuticals (as organic pollutants) as well as Content, Nomenclature list and Acronym list. Below, we indicate how the referee’s comments have been taken into account in the novel revised version of the manuscript.
Reviewer Comments to Author:
Reviewer 2
I have only read the first three sections of the manuscript. On page 9 the number of comments was closing in on 60! These comments can be traced as numbered items #i on the attached pdf of these first pages that also include additional questions marks or suggestions for linguistic changes.
Response: The authors are grateful to the Referee for the detailed analysis of the manuscript and valuable and instructive comments.
After the detailed comments I will draw a conclusion and make a recommendation.
#1 water contamination means water that comes into the environment? There should be examples for the organic pollutants as well, as for the inorganic pollutants.
Response: Yes, the water contamination means the presence of the pollutants of different genesis in water resources as part of the environment, drinking water, leachate water, etc. The examples of common organic and inorganic pollutants are given in the next paragraph of the Revised manuscript.
#2 metal ions are not organic pollutants
Response: Sorry for this error. The incorrect collocation “The common organic pollutants…” was replaced by “The common pollutants…” in the relevant phrase of the Revised manuscript.
#3 there is a distinct difference in the kind of pollution between rural and town environments, between industrial areas and natural environments, between first world and third world countries. This could be detailed here. Also with respect to the technological and infrastructural options to clean water.
Response: The authors are grateful to Referee for this important suggestion. We agree that these issues are of prime importance in order to gain more deep insight into the problem of water remediation in a global scale. Some short corrections are added to the Revised manuscript. However, this topic deserves a separate review with a special emphasis of the formulated problems and careful analysis of the wastes of different genesis.
#4 is underground meaning groundwater? Please specify a bit more.
Response: Thank you for this comment. Accordingly, “underground water” was replaced by “ground water” accompanied by a short specification in the Revised manuscript.
#5 it is not wastes, it is the organic products or what substances that even may enter the environment after having passed through human or animal bodies (like antibiotics or other medications).
Response: According to this suggestion, this collocation was replaced by “The diverse organic and inorganic compounds and species…”.
#6 highly hydrophobic? How is the degree of hydrophobicity defined. The most hydrophobic substances will accumulate at water/air interfaces and are then not really dissolved. They will tend to any kind of surface as well to lower surface energies.
Response: According to this comment, the incorrect term “highly” was removed from the phrase in the Revised manuscript: “Most of the organic pollutants are hydrophobic, therefore, they accumulate in water and then penetrate the tissues of various aquatic organisms, as well as humans [12].”
#7 it is not high content only, it can be traces that are very harmful.
Response: We agree completely with this important comment. Accordingly, the relevant phrase was replaced by a more correct one in the Revised manuscript: “So, even trace concentrations of toxic pollutants in water may lead to serious health problems”.
#8 what are the conventional methods, give examples
Response: The examples of the conventional methods, e.g., membrane separation processes, coagulation and flocculation, reagent purification, etc., were added to the relevant phrase of the Revised manuscript: “Conventional methods, such as membrane separation process, coagulation and flocculation, reagent purification, etc. demonstrate certain drawbacks, such as, time-consuming, laborious, and quite expensive means and even producing undesirable (more harmful) by-products.”
#9 if you use alumina, it is already a solid particle and cannot be formed anymore
Response: According to this suggestion, the relevant phrase was replaced by more correct one in the Revised Manuscript: “In turn, the coagulation method for removal of non-settleable particles is based on the formation of suspended solid particles with coagulation chemicals such as aluminum and iron coagulants, for instance, ferric and aliminum sulfates [20]. These particles gather together to form larger particles in the flocculation process using polyaluminum chloride, biopolymeric pectin, polyacrylamide, etc., [21]. Then, these larger particles are removed through sedimentation process.”
#10 flocculants are additions that cause flocculation. I think it is typically not solid particles. It is rather solutes, like Al or Fe that will first be solutes before they form solids. It is not precisely described here.
Response: Thank you for this valuable comment. According to this suggestion, the more precise description for the flocculation technique was added to the Revised manuscript.
#11 what is the difference between coagulation and flocculation? I think they are connected. First you form the solids and then these solids aggregate and will sediment or can be taken off in some other way. So it is even connected to sedimentation. I know this as a suite of processes. One distinct process will not be useful.
Response: The difference between coagulation and flocculation is outlined in the Revised manuscript. We agree completely that coagulation-flocculation-sedimentation are commonly combined in one removal process. The relevant corrections are added to the Revised manuscript. See, please the answer to the comment 9.
#12 The combinations of methods is not mentioned. This has to be the future given the diversity of contaminants.
Response: Thank you for this valuable comment. Accordingly, an importance of the combination of removal methods is emphasized in the Revised manuscript.
#13 what are relatively mild conditions here?
Response: The most adsorption processes in the liquid phase are realized mostly at room temperature. The corrections were made in the relevant part of the Revised manuscript: “This process involves mild operating conditions by implementing mostly at room temperature, and robust cost-energy efficiency.”
#14 uptake depends on pH and many other things as the authors claim later. So how can you define something like an adsorption capacity. It is a term that is often used and gives the impression of describing the capability of an adsorbent to bind something. However, it is never known what conditions this refers to and it can be entirely different if you just change the pH.
Response: We agree that the term «adsorption capacity» is incorrect in this context. The relevant phrase was corrected “Efficient adsorptive removal of hazardous materials relies on different factors including rapid uptake and easy regeneration of the used adsorbents [25]”.
#15 in water cleaning if this is the general method, there cannot be target pollutants, but everything that has the potential to cause damage has to be removed (see #12).
Response: According to this comment, the relevant phrase was replaced by the more correct one “The important properties of the adsorbents of choice are high porosity, large specific surface area, chemical stability, processability and regenerability [26]”.
#16 it is not clear how this all is done or expected to be operational. It could be discontinuous (batch-type) but also in a column type arrangement. It should be discussed at some point.
Response: The relevant description of the key process parameters for column and batch adsorption design are added to the Revised manuscript. The research paper discussed in this review deals with the batch adsorption process.
#17 please describe shortly what you understand by carbonaceous adsorbents. For some it could include carbonates, but this is probably not the case here.
Response: The erroneous term “carbonaceous” was removed from the relevant phrase of the Revised manuscript: “Among these characteristics, the pore geometry, favorable pore structure and surface chemistry are key factors affecting the adsorption characteristics of adsorbents.”
#18 I got lost here. Why would you classify it this way. Chemsorption could be covalent as well. Ionic interactions could be associated to electrostatics, but this is rather physical adsorption. Where does the H-bonding fall in. Classify the mechanisms which you later discuss in detail here.
Response: The classification of the adsorption mechanisms according to this comment was added to the Revised manuscript: “Usually, the adsorption mechanism involves intermolecular forces of attraction, such as chemisorption, e.g., chemical bonding, acid-base interactions, coordination interactions and physisorption, e.g., electrostatic interactions, hydrogen bonding, van der Waals forces, π-π interactions and diffusion [30].”
#19 this is ambiguous, one might understand that alumina and the rest are also biosorbents. Then mesporous is not the same class of material as alumina or zeolites.
Response: According to this suggestion, the relevant phrase was replaced by the more correct one in the Revised manuscript: “Currently, various materials are used as adsorbents for the removal of organic and inorganic pollutants [31], such as AC [32,33], polymeric matrices [34], agro-industrial wastes [35] which are regarded as low-cost adsorbents and biosorbents, e.g., biomass and chitosan [36, 37] as well as inorganic matrices including alumina [38], zeolites [39], soil/clays [40, 41], mesoporous silicas [42], and hybrid nanoporous solids, i.e., MOFs [8].”
#20 you discuss solids and these cannot have a pH. Only in water or solution will they cause a certain pH.
Response: The erroneous collocation “neutral pH” was removed from the relevant phrase.
#21 why not you AC after you have defined it.
Response: According to this comment, abbreviation AC is used in the relevant phrase of the Revised manuscript. Additionally, the Nomenclature list and Acronym list were added to the Revised manuscript.
#22 what are these unsatisfactory adsorption characteristics etc.? without more detail the reader is lost, because before AC has been advocated.
Response: This phrase was corrected: “However, AC has also a number of disadvantages such as irregular porous structure and complicated porous networks, as well as poor selectivity and ineffectiveness in the removal of some emerging organic pollutants from water due to the unfavorable kinetics [47] and lack of complete removal and low adsorption capacity at low concentrations of pollutants [48, 49].” in the Revised manuscript.
#23 please give more detailed examples for the nano-engineered adsorbents and the classical adsorbents. Also say something about cost of making them and availability.
Response: According to this comment, the Revised manuscript was supplemented by providing more information regarding nanoadsorbents: “Nanoadsorbents can be divided in the following groups: metallic nanoparticles, nanostructured mixed oxides, carbonaceous nanomaterials and silicon nanomaterials including silicon nanotubes, silicon nanoparticles, silicon nanosheets as well as nanoclays, polymer-based nanomaterials, nanofibers, aerogels, and their composites with biochar [54].”
#24 shape? Nanotubes, rods and wires are nanoparticles. So you could say nanoparticulate adsorbents… also nanocomposites are not in the same class. It is not clear whether they would be mixtures of tubes, rods and wires for example or if they have a composite chemical structure.
Response: According to this comment, this phrase was replaced by a more correct one “Nanoadsorbents according to their morphology can be classified as nanotubes, nanorods, nanowires and nanosheets [56, 57]”.
#25 in this introduction I would have expected something about the role of pH. What is the range of pH for which reasonable cleaning operations can be carried out? Then there is pH effect on uptake of many solutes. Also water composition may play a role (complexation keeping metal ions from being adsorbed for example). Finally temperature plays a role and the trends could be introduced in this introduction.
Response: Thank you for this valuable suggestion. The pH and temperature effects in the adsorption processes are discussed in the next sections of this Review using relevant examples. Some examples of the role of the water composition are given in Section 4 of the Revised manuscript.
#26 These kinds of models cannot treat multicomponent systems in a straightforward ways. The cannot intrinsically handle variation of pH and effects of charge. Surface complexation models exist to allow all this. Why not advocate those?
Response: Thank you for this valuable suggestion. The relevant paragraph regarding SCMs was added to Section 2 of the Revised manuscript.
#27 I am not sure whether the graphene oxide is really inherently hydrophobic.
Response: The incorrect term “inherent hydrophobic” was removed from the relevant phrase of the Revised manuscript: “In addition, the surfaces of these materials can be functionalized to target specific pollutants via chemical or electrical interactions [55].”
#28 which types are you talking about? It only becomes clear later.
Response: According to this comment, the relevant phrase was removed from the Revised manuscript.
#29 why not put the simple one before the complex one?
Response: Thank you for this comment. The parts of Figure 1 were swapped.
#30 explain how the MWCNTs can keep their structure? i.e. why do the tubes not simply separate to form SWCNTs?
Response: The relevant fragment, which explains the integrity of the structure of MWCNTs, is inserted in the Revised manuscript (Section 3.1.1).
#31 many surfaces will be negative (as stated later also by the authors) and many organic pollutants are rather negative than positive as well. So overall the surface charge will not help adsorption.
Response: We agree with this comment. However, in some special cases, the surface charge can contribute to the adsorption. It is reported, for instance in [Y. Pi, X. Li, Q. Xia, J. Wu, Y. Li, J. Xiao, Z. Li. Chem. Eng. J., 2018, 337, 351; DOI: 10.1016/j.cej.2017.12.092].
#32 can surface charge have properties?
Response: Sorry for this misprint. The relevant phrase was replaced by most correct one in the Revised manuscript: “The chemical structure, chemical composition including functionalization, distribution of dimensions (uniformity of sample and aspect ratios), physicochemical properties of the surface, agglomeration state and presence of extraneous metallic nanoparticles (from the manufacturing processes) were identified as the most critical parameters of CNTs [53].”
#33 what about release of components of such sorbents? Often the additions might become pollutants.
Response: Thank you for this important suggestion. We agree completely that a release of some species deposited on CNTs during the preparation process is an important issue in terms of additional contamination of aquatic media due to usage of the adsorbents themselves. A relevant fragment was added to Section 3.1.1 in the Revised manuscript. It is briefly pointed out in the Revised Manuscript. Actually, this problem deserves a special consideration, which is not included in this work.
#34 header says organic, now we included inorganic
Response: The erroneous word “inorganic” was removed from the title for Section 3.1.2 in the Revised manuscript.
#35 how? More amino than carboxylate? pH? Overall instead of cumulative?
Response: This information is a suggestion of the authors of [96]. The relevant phrase was replaced by the more correct one in the Revised manuscript. “Moreover, CNTs have micropores and mesopores and different functional groups such as –COOH, –OH, and/or –NH2 [96].”
#36 again this depends on pH and nothing is said about ranges of charge and pH.
Response: Actually, the surface charge is critically dependent on the pH value. The pH effect is discussed by presenting examples of adsorption of organic pollutants using CNTs. Accordingly, the relevant phrase was replaced by more correct one in this context: “The modification techniques alter the characteristics of the CNTs surface (including functionality) and porosity, dispersion and hydrophobicity [99].”.
#37 dispersion and particle size is something different. Good dispersion may be obtained for any size as long as sedimentation is avoided.
Response: According to this comment, the collocation “particle size” was removed from the relevant phrase (see, please, answer to comment 36) in the Revised manuscript.
#38 Two of them are no acids to me.
Response: Sorry for this misprint. The relevant phrase was replaced by the more correct one in the Revised manuscript: “In particular, acid treatment of CNTs is carried out by using different oxidants, including HNO3, KMnO4, H2O2, H2SO4 and HCl.”
#39 again, what conditions? Is it continuous or batch-type? What are the quantities.
Response: Thank you for this important comment. The majority of the research papers discussed in this review deal with model adsorption processes that require small quantities of the adsorbents under batch conditions.
#40 what if such more loosely bound components of functionalized sorbents are released and cause harm?
Response: Thank you for this important comment. Accordingly, the following fragment was added to the Revised manuscript: “However, a possible release of the modifying agent, which could contribute to the additional water contamination, is an important problem in this case. So, a covalent modification of the CNT surface is preferable from this point of view.”.
#41 again a new component. There are many ILs and many contain not so nice components. My idea would be to keep anything potentially harmful away. I am not sure abou the imidazole for example in this context.
Response: We agree with this important suggestion. Accordingly, the following phrase was added to the Revised manuscript. “However, the potential release of ILs from the CNT surface should be evaluated to avoid the potential additional contamination with these modifying agents.”
#42 order for what? Adsorption of what?
Response: According to this comment, the relevant phrase was replaced by the more correct one in the Revised manuscript: “The following order of the Om values was observed for CNTs and IL@CNT composites as follows: CNT+KET/SMZ < CNT@ILs+SMZ < CNT@ILs+KET.” Qm is a maximum adsorption capacity.
#43 what adsorption values? Maximum uptake, Langmuir constants?
Response: According to this suggestion, the relevant phrase was replaced by the more correct one: “The Qm values of the different CNT materials for specific organic pollutants, impact of activation/modification of the adsorption efficiency, dominating adsorption mechanism and effect of conditions of the removal process are listed in Table 1.”.
#44 it could be anything. But typically it is a mixture of initial interaction and subsequent transport properties.
Response: Thank you for this important comment. Accordingly, this phrase was replaced by the more correct one: “The adsorption removal of organic pollutants from water on CNTs may be realized as physisorption and/or chemisorption, associated with their transport properties contributing to the intra-particle diffusion [108].”.
#45 in the real world waters, there will be multicomponent solutions with competition for adsorption sites. There might also be hindrance of bigger molecules blocking pores or reactions in the bulk solution that change the system from the single component lab investigation.
Response: Thank you for this instructive comment. Accordingly, the Revised manuscript was supplemented by the following fragment: “Especially, revealing the adsorption mechanism is difficult for the real water probes, which are usually multicomponent systems. So, interactions of adsorbates and their competition for the adsorption sites associated with pore blocking by the bulkier pollutant molecules contribute to the adsorption process.”.
#46 EDA is not defined. Later I could guess what it means.
Response: EDA is Electron Donor-Acceptor. The relevant definition was added to the Acronym list in the Revised manuscript.
#47 I would guess both are at play.
Response: According to this important comment, the relevant phrase was replaced by the more correct one in the Revised manuscript: “The next possibility is an n-π EDA interaction (probably, in a combination with π-π interactions), which involves a direct interaction of a lone electron pair of nitrogen and oxygen in the amino group, hydroxyl group, O- or N-heteroatoms as n-electron donors with π-electron-acceptor sites on the CNTs surface.”
#48 not clear if this refers to solute or surface or both
Response: According to this comment, the relevant phrase was replaced by more correct one: “Oxygen-containing functional groups on the CNT surface may interact with -ОН, –NH, and –NH2 groups of organic molecules via hydrogen bonding that contributes to the organic pollutant adsorption on CNTs [122, 123].”
#49 IR spectra in the H-bonding region can be messed up by water bonds. I do not know to what extent this community takes this into account. I would hope they do.
Response: Thank you for this important comment. The authors [121] consider the results of IR study as reliable ones without any superposition or shielding.
#50 H-bonding is typically on the weak side. So how tightly bound is a pollutant and can it be easily released? It raises again the question about the operation (batch vs flow through). Also in multicomponent systems strong adsorption of one component may impede the H-bonding of another.
Response: Thank you for this important comment. The authors [124] studied a model one-component system and suggest that H-bonding is a more possible dominating mechanism of acetaminophen adsorption. Accordingly, the relevant phrase was replaced by the more correct one in the Revised manuscript: “It was suggested [124] that hydrogen bonding is a dominating mechanism for the adsorption of acetaminophen on МWCNT.”
#51 potentials not forces. Maybe forces between adsorbents or between solutes and adsorbents.
Response: Authors agree completely with this important comment. The relevant phrase was replaced by a more correct one: “The appearance of a positive and negative charge on the CNT surface due to deviance from pH point of zero charge creates the electric potential on the adsorbent”.
#52 This is what I mentioned before. With the negative charge there is repulsion concerning anions and many organic solutes are anions.
Response: Authors agree completely with this valuable comment.
#53 what are all these compounds (give strutures or at least say to which class of pollutant they belong)?
Response: The structures of some selected pharmaceuticals like ibuprophen are given in Figure 2 of the Revised manuscript.
#54 Be careful to state the conditions in more detail, like temperature. Also structures could be given for the molecules, so the reader can get an idea about functional groups.
Response: The adsorption conditions for diethyl phthalate, oxytetracycline and сiprofloxacin are included in the relevant paragraphs of the Revised manuscript. The structures of some selected organic pollutants including сiprofloxacin are given in Figure 2 of the Revised manuscript.
#55 this is often a misunderstanding or a wrong conclusion. Electrostatics may help, but often it is just a part of the story. And again without knowing a bit more beyond the name of the compound it is difficult to follow.
Response: Thank you for the valuable suggestion. Accordingly, the relevant phrase was replaced by more correct one: “According to [132], electrostatic interaction is suggested as a dominating mechanism in this adsorption process.”
#56 qe is often in moles/kg. so it is not concentration, please check.
Response: Thank you for this comment. Accordingly, this phrase was removed from Section 3.1.4.4 of the Revised manuscript. The relevant description for these parameters, i.e., qe - a quantity of the adsorbate adsorbed at the equilibrium (mg/g), and Ce – equilibrium concentration of an organic pollutant in solution, mg/L, are given in the Nomenclature list.
#57 it suggests. There is no proof.
Response: Accordingly, the relevant phrases were replaced by more correct ones: “It was suggested [138] that hydrophobic interactions play a decisive role in chlorophenol adsorption on SWCNTs. A linear correlation between logKow and logKd indicates that the adsorption ability correlates positively with hydrophobicity.”
#58 reference to table 1 is too late here I guess.
Response: The reference to Table 1 is first mentioned in the Section 3.1.4.
Also table 1 could include columns for pH and concentration ranges in all cases (you might exlude cases where it is not specified), overall charge of the sorbent at the operational pH, uptake in moles/m2 as well for better comparison, removal in percentage.
I guess those numbers give much more information than mg/g. With the specific surface area it is easy to see that the per mass is misleading given that adsorption is a surface phenomenon.
Response: Table 1 presents the range of рН values (if it is mentioned in the cited work) and adsorption capacity, mostly, maximum uptake (if it is mentioned in the cited work). These parameters are presented in most analyzed papers.
The overall charge of the sorbent at the operational pH is not given in majority of the considered works.
As to the uptake in moles/m2 and removal in percentage, we have presented the adsorption capacity units, which were given by the authors. These units were not recalculated.
Reviewer 3 Report
Dear Authors,
Dear Editor,
I read the manuscript and I provide the following report:
The manuscript is comprehensive and mostly cover the topic assumed by the title. in my opinion, the Carbon-based materials are well describing the uses for pharmaceuticals and pesticides removal but only marginally for inorganic pollutants, such as heavy metals (many other works such as: DOI: 10.3390/ma13071687 can be used). Also, in my opinion something related to the potential resorption in environmental conditions should be considered because the potential toxicity of these synthetic materials have to be considered. For this I could recommend: https://doi.org/10.1016/j.chemosphere.2020.127885 but again, you have to consider other references.
The manuscript has enough consistency and could be even considered in this form but, I am confident that both authors and editors will agree that my comments are just devoted to improve the quality of the manuscript.
Best regards,
R1
Author Response
September 03, 2021
MDPI Molecules
Manuscript ID: molecules-1301684
Title: "Modern Carbon-Based Materials for Adsorptive Removal of Organic and Inorganic Pollutants from Water and Wastewater"
Author(s): Vera I. Isaeva*, Marina Vedenyapina*, Alexandra Kurmysheva, Dirk Weichgrebe, Rahul Ramesh Nair, Ngoc Phuong Thanh Nguyen, Leonid M. Kustov*
Dear reviewers,
Thank you very much for your helpful and valuable comments and advices. According to your instructive remarks and suggestions, we have done a significant revision of our manuscript. We added also the Supplementary Material file with Tables i, ii, iii, which are Tables 5, 6, 7 in the first submitted version of the manuscript. The Revised manuscript was supplemented by new references 155, 163, 276, 397, flowchart (Scheme 1), Scheme 2 (synthesis of GO) and Figure 2 with structures of selected pharmaceuticals (as organic pollutants) as well as Content, Nomenclature list and Acronym list. Below, we indicate how the referee’s comments have been taken into account in the novel revised version of the manuscript.
Reviewer Comments to Author:
Reviewer 3
I read the manuscript and I provide the following report:
The manuscript is comprehensive and mostly cover the topic assumed by the title. in my opinion, the Carbon-based materials are well describing the uses for pharmaceuticals and pesticides removal but only marginally for inorganic pollutants, such as heavy metals (many other works such as: DOI: 10.3390/ma13071687 can be used). Also, in my opinion something related to the potential resorption in environmental conditions should be considered because the potential toxicity of these synthetic materials have to be considered. For this I could recommend: https://doi.org/10.1016/j.chemosphere.2020.127885 but again, you have to consider other references.
Response: Authors are grateful for these instructive suggestions. Some discussion of the potential toxicity of the considered carbon-based nano-adsorbents has been added to the Revised manuscript. See, please, Introduction, sections 1, 3 and 4. The citations of these important papers with an appropriate discussion as Refs. 155 and 163 along with one more Ref. 276 have been added to the Revised manuscript.
The manuscript has enough consistency and could be even considered in this form but, I am confident that both authors and editors will agree that my comments are just devoted to improve the quality of the manuscript.
Reviewer 4 Report
Article entitled Modern Carbon–Based Materials for Adsorptive Removal of Organic and Inorganic Pollutants from Water and Wastewater written by Vera I. Isaeva, Marina D. Vedenyapina, Alexandra Yu. Kurmysheva, Dirk Weichgrebe, Rahul Ramesh Nair, Ngoc Phuong Thanh Nguyen and Leonid M. Kustov and submitted to Molecules journal deals with an important issue of water and wastewater treatment.
The article is interesting and could be considered for publication in Molecules journal. As English is not my native language, I am not able to assess language correctness. However, while reading, I found some statements missing, confusing or unclear. Below I enclose the list of my comments.
All abbreviations should be explained at first appearance.
There are some typos in the text, e.g. with units on a page 3 /indexes/. Text should be corrected.
Paragraph 2.1. some other models including BET are missing. I think BET should be mentioned, it is commonly determined for sorbents. It is even mentioned in table 2 or 3.
There are some repetitions eg: “The adsorption mechanism includes π-π interactions between π systems on CNTs”, “CNT walls through hydrogen bonds, Van-der-Waals forces, π-π-stacking interactions,”, 3.1.5 etc.
From the beginning of the article, the Authors write about the removal of pollutants that the use of modified carbon-based materials is a great alternative to other treatment methods. However, it should be clearly stated over and over again that adsorption as such has a strong, powerful disadvantage – only micropollutants can be removed in this way.
Very little space is devoted to the sorbents’ preparation methods.
The title of the article is misleading - The authors discuss the removal of organic compounds. The content should be supplemented with inorganic compounds or the title should be corrected (it is only mentioned in case of organic sorbents chapter 6).
When discussing individual materials, the authors focus on the removal mechanisms, wanting to characterize the material almost exclusively by the composition and adsorption capacity. What about other parameters?
Table on pages 71 – 77: description is missing. I suggest placing it horizontally, not vertically (as eg. tables 2 and 3).
Chapter 6 clearly differs from the previous ones in terms of approach and description. I propose to unify or delete this chapter.
The authors completely ignore the issue of the applicability of the materials described. The key in practical application are the costs of the material and the method of its regeneration / disposal after use. The article should be supplemented with such considerations. Here again we can see the difference between chapter 6 and the rest of the work - in the case of "bio" sorbents their way of producing or modifying them is much easier, compared to the sorbents described in earlier chapters. There is also no discussion on this topic.
Based on my comments and general impression I suggest major revision.
Author Response
September 03, 2021
MDPI Molecules
Manuscript ID: molecules-1301684
Title: "Modern Carbon-Based Materials for Adsorptive Removal of Organic and Inorganic Pollutants from Water and Wastewater"
Author(s): Vera I. Isaeva*, Marina Vedenyapina*, Alexandra Kurmysheva, Dirk Weichgrebe, Rahul Ramesh Nair, Ngoc Phuong Thanh Nguyen, Leonid M. Kustov*
Dear reviewers,
Thank you very much for your helpful and valuable comments and advices. According to your instructive remarks and suggestions, we have done a significant revision of our manuscript. We added also the Supplementary Material file with Tables i, ii, iii, which are Tables 5, 6, 7 in the first submitted version of the manuscript. The Revised manuscript was supplemented by new references 155, 163, 276, 397, flowchart (Scheme 1), Scheme 2 (synthesis of GO) and Figure 2 with structures of selected pharmaceuticals (as organic pollutants) as well as Content, Nomenclature list and Acronym list. Below, we indicate how the referee’s comments have been taken into account in the novel revised version of the manuscript.
Reviewer Comments to Author:
Reviewer 4
Article entitled Modern Carbon–Based Materials for Adsorptive Removal of Organic and Inorganic Pollutants from Water and Wastewater written by Vera I. Isaeva, Marina D. Vedenyapina, Alexandra Yu. Kurmysheva, Dirk Weichgrebe, Rahul Ramesh Nair, Ngoc Phuong Thanh Nguyen and Leonid M. Kustov and submitted to Molecules journal deals with an important issue of water and wastewater treatment.
The article is interesting and could be considered for publication in Molecules journal. As English is not my native language, I am not able to assess language correctness. However, while reading, I found some statements missing, confusing or unclear. Below I enclose the list of my comments.
All abbreviations should be explained at first appearance.
Response: The Acronym list was added to the Revised Manuscript.
There are some typos in the text, e.g. with units on a page 3 /indexes/. Text should be corrected.
Response: Sorry for the misprints. The Revised manuscript was carefully checked and corrected.
Paragraph 2.1. some other models including BET are missing. I think BET should be mentioned, it is commonly determined for sorbents. It is even mentioned in table 2 or 3.
Response: The brief presentation of the Surface Complexation Models (SCMs) is added to section 2 of the Revised manuscript.
There are some repetitions eg: “The adsorption mechanism includes π-π interactions between π systems on CNTs”, “CNT walls through hydrogen bonds, Van-der-Waals forces, π-π-stacking interactions,”, 3.1.5 etc.
Response: The Revised manuscript was carefully checked in order to avoid the repetitions.
From the beginning of the article, the Authors write about the removal of pollutants that the use of modified carbon-based materials is a great alternative to other treatment methods. However, it should be clearly stated over and over again that adsorption as such has a strong, powerful disadvantage – only micropollutants can be removed in this way.
Response: Authors agree completely with this important suggestion. An appropriate discussion of this disadvantage along with a strategy to overcome it, i.e., the combination with other removal techniques is added to Section 1, Introduction and Conclusion of the Revised manuscript.
Very little space is devoted to the sorbents’ preparation methods.
Response: A number of additional fragments relevant to the preparation methods along with Scheme 2 (Preparation of GO) and Refs. 155 and 276 were added to Sections 3, 4 and 5 in the Revised manuscript.
The title of the article is misleading - The authors discuss the removal of organic compounds. The content should be supplemented with inorganic compounds or the title should be corrected (it is only mentioned in case of organic sorbents chapter 6).
Response: A brief discussion of heavy metal adsorbents along with Ref. 155 was added to Introduction, Sections 1, 3, 4 and 5 of the Revised Manuscript.
When discussing individual materials, the authors focus on the removal mechanisms, wanting to characterize the material almost exclusively by the composition and adsorption capacity. What about other parameters?
Response: The authors appreciate this important suggestion. The main focus of this review is elucidation of the adsorption mechanism and structure-adsorption properties (capacity) relationship for the considered carbon-based adsorbents. The discussion of other important parameters, such as removal efficiency, adsorption kinetics, adsorption isotherms and adsorbent reusability/regenerability, has been added to Sections 4 and 5 of the Revised manuscript.
Table on pages 71 – 77: description is missing. I suggest placing it horizontally, not vertically (as eg. tables 2 and 3).
Response: An appropriate discussion of Tables 5-7, which are Tables i, ii, iii, was added to Chapter 6 of the Revised manuscript. Since these tables are large, they have been moved to the Supplementary Material as Tables i, ii and iii.
Chapter 6 clearly differs from the previous ones in terms of approach and description. I propose to unify or delete this chapter.
Response: Chapter 6 was revised according to the reviewer’s suggestion.
The authors completely ignore the issue of the applicability of the materials described. The key in practical application are the costs of the material and the method of its regeneration / disposal after use. The article should be supplemented with such considerations. Here again we can see the difference between chapter 6 and the rest of the work - in the case of "bio" sorbents their way of producing or modifying them is much easier, compared to the sorbents described in earlier chapters. There is also no discussion on this topic.
Response: The authors are grateful to the referee for this valuable suggestion. Some discussion of the mentioned important issues is added to Introduction, Sections 1, 4, 5 and Conclusions of the Revised manuscript. Moreover, new relevant references 163, 276 were added to the Revised manuscript.
Reviewer 5 Report
The manuscript was revised, the topic was focused on the preparation and modification of carbon-based systems for the adsorption. Adsorption mechanisms are investigated, few revisions are required and reported below: - please try to reduce acronyms from the abstract - please add numbers for the equations - add a nomenclature list with acronyms and parameters - for table 1 add information about the choice of adsorbate - revise table 2 and table 3 - I suggest to reduce the materials presented using a supplementary material sectionAuthor Response
September 03, 2021
MDPI Molecules
Manuscript ID: molecules-1301684
Title: "Modern Carbon-Based Materials for Adsorptive Removal of Organic and Inorganic Pollutants from Water and Wastewater"
Author(s): Vera I. Isaeva*, Marina Vedenyapina*, Alexandra Kurmysheva, Dirk Weichgrebe, Rahul Ramesh Nair, Ngoc Phuong Thanh Nguyen, Leonid M. Kustov*
Dear reviewers,
Thank you very much for your helpful and valuable comments and advices. According to your instructive remarks and suggestions, we have done a significant revision of our manuscript. We added also the Supplementary Material file with Tables i, ii, iii, which are Tables 5, 6, 7 in the first submitted version of the manuscript. The Revised manuscript was supplemented by new references 155, 163, 276, 397, flowchart (Scheme 1), Scheme 2 (synthesis of GO) and Figure 2 with structures of selected pharmaceuticals (as organic pollutants) as well as Content, Nomenclature list and Acronym list. Below, we indicate how the referee’s comments have been taken into account in the novel revised version of the manuscript.
Reviewer Comments to Author:
Reviewer 5
The manuscript was revised, the topic was focused on the preparation and modification of carbon-based systems for the adsorption. Adsorption mechanisms are investigated, few revisions are required and reported below: - please try to reduce acronyms from the abstract - please add numbers for the equations - add a nomenclature list with acronyms and parameters - for table 1 add information about the choice of adsorbate - revise table 2 and table 3 - I suggest to reduce the materials presented using a supplementary material section
Response: Thank you for instructive comments. Accordingly, some acronyms were removed from the Abstract in the Revised manuscript.
The equations were numbered in the Revised manuscript.
The Nomenclature list and Acronym list were added to the Revised manuscript.
Tables 1, 2, 3 and 4 were corrected in the Revised manuscript.
Tables 5, 6, 7 were moved to Supplementary Information as Tables i, ii, iii.
Round 2
Reviewer 1 Report
I agree to accept the article in present form,though it can be further revised.
Author Response
Dear Reviewer,
Thank you for your valuable comments. The review was further revised and edited
Leonid Kustov
Reviewer 2 Report
molecules-1301684
I am still not at all convinced that this will be publishable. The language needs shaping, and there are also shortcomings in many other respects. I include a scanned copy of the first part with handwritten comments and numbered items, to which I will refer in the following.
#1 I think it should be Cr(VI) or CrO4-2; there is no Cr6+ ion.
#2 there is at least one reference required here, ideally several. This is a recurring mark.
#3 this high surface area has to be internal, i.e. in the pores. Maybe a relation between the external and internal area should be given. It is probably you can find an example for this.
#4 this seems weird to me. At low concentrations usually the uptake is high in fractional terms. The term capacity is for me always related to occupation/maximum possible uptake. Instead there is the interplay between maximum possible uptake (from a physical point of view in terms of available space) and affinity (from a chemical point of view), as comes up e.g. in the Langmuir isotherm. The term capacity blurs this distinction.
#5 is it reuse? These passages are very difficult to read, also due to the formatting. It would be nice to also have one clean copy, where the tracking is eliminated.
Scheme 1: there are typos in there. It should by synergism for example.
#6 the Freundlich isotherm does not at all consider layers. So how do you want to relate it to it. It would be better to say something like “unlimited uptake” or so. It is confusing as it is.
#7 you should point out that the modern models have advantages of course. Langmuir and Freundlich parameters all depend on pH, temperature, or solution components, and are affected by the charging properties. Therefore, comparison between different systems make little sense. SCMs can give you some stability constant at infinite dilution and the use of a consistent model for the double layer will allow to have data base. Right now it is measuring and producing parameters that at most will confuse future researchers also in the sense that starters will believe these parameters have a physical or chemical meaning.
#8 what conductivity, heat, electrical?
#9 are the sheets also hexagonal or only the arrangement of the atoms on the sheets? I think there is one hexagonal in excess.
#10 I got the impression that thickness and diameter are confused, also it has to be clear if the dimensions refer to one sheet or the particles.
#11 critical in what sense? I got lost here.
#12 application to what?
#13 there seems to be some repetition here.
#14 not clear to what extent HCl is an oxidant in this context.
#15 this is again completely unclear. I guess assisting in dispersion in water means that the composites are better dispersable than the bare CNTs. It is not clear how the composites are composed. Magnetite bound to the CNTs? And the use of the word high somehow suggests they had high dispersability already.
#16 not sure what the relevance of the ionic liquids is. If these are then in contact with water, what is going to happen?
#17 how did the authors (#2 references missing) determine this? Was it an interpretation based on the macroscopic data or spectroscopy?
#18 this is section as far as I can see. Would it be section 3.3.? why not just give the number.
#19 I think a scheme could help here, showing the ranges. In general for a review some schemes/figures are in general inserted to make things clear. The text is not so illustrative for people consulting a review, where the beginner should also be able to get some picture of the topic.
#20 measurements of electrokinetic data? There is often too much detail and often too little. The balance is not ok for a review.
I stopped here, also because I did not even mark all the issues I could have raised in this first part. As pointed out before, I think the authors need to make much more of an effort for such a kind of review.

Author Response
October 14, 2021
MDPI Molecules
Manuscript ID: molecules-1301684
Title: "Modern Carbon-Based Materials for Adsorptive Removal of Organic and Inorganic Pollutants from Water and Wastewater"
Author(s): Vera I. Isaeva*, Marina Vedenyapina*, Alexandra Kurmysheva, Dirk Weichgrebe, Rahul Ramesh Nair, Ngoc Phuong Thanh Nguyen, Leonid M. Kustov*
Dear reviewer,
The authors are grateful for detailed review analysis and priceless comments and suggestion. According to your instructive remarks we have done a further revision of our manuscript. Below, we indicate how the referee’s comments have been taken into account in the novel revised version of the manuscript.
Reviewer 2
I am still not at all convinced that this will be publishable. The language needs shaping, and there are also shortcomings in many other respects. I include a scanned copy of the first part with handwritten comments and numbered items, to which I will refer in the following.
#1 I think it should be Cr(VI) or CrO4-2; there is no Cr6+ ion.
According to this instructive comment, “Cr6+ ions” were replaced by “Cr(VI)” in the Revised manuscript.
#2 there is at least one reference required here, ideally several. This is a recurring mark.
Thank you for this instructive comment. The novel relevant Reference 92 was inserted in the Revised manuscript.
#3 this high surface area has to be internal, i.e. in the pores. Maybe a relation between the external and internal area should be given. It is probably you can find an example for this.
Thanks to the important instructive comment. Yes, the authors mean SSA. Accordingly, the relevant phrase was replaced by a more correct one in the Revised manuscript “AC has a micropore-dominant structure with a high external and internal or microporous surface area (the cumulative surface of the micropore walls) [46].
#4 this seems weird to me. At low concentrations usually the uptake is high in fractional terms. The term capacity is for me always related to occupation/maximum possible uptake. Instead there is the interplay between maximum possible uptake (from a physical point of view in terms of available space) and affinity (from a chemical point of view), as comes up e.g. in the Langmuir isotherm. The term capacity blurs this distinction.
According to this remark, the relevant phrase was replaced by a more appropriate one: “However, AC has also a number of disadvantages such as irregular porous structure and tangled porous networks, as well as poor selectivity and ineffectiveness in the removal of some emerging organic pollutants from water due to unfavorable kinetics [47] and lack of complete removal at low concentrations of pollutants [48, 49].”
#5 is it reuse? These passages are very difficult to read, also due to the formatting. It would be nice to also have one clean copy, where the tracking is eliminated.
Scheme 1: there are typos in there. It should by synergism for example.
Sorry for misprints and inappropriate formatting. Accordingly, the Revised manuscript was carefully corrected.
#6 the Freundlich isotherm does not at all consider layers. So how do you want to relate it to it. It would be better to say something like “unlimited uptake” or so. It is confusing as it is.
Thank you for the comment and suggestion. The Freundlich isotherm assumes a heterogeneous surface with different affinities that has multilayer sorption. References are cited from:
- 1. Gerente, C., Lee, V.K.C., Le Cloirec, P., and McKay, G. (2007). Application of chitosan for the removal of metals from wastewaters by adsorption – Mechanisms and models review. Critical Reviews in Environmental Science and Technology 37, 41-127, doi:10.1080/10643380600729089.
- 2. Inyan, M.I, Gao, B., Yao, Y., Xue, Y., Zimmerman, A., Mosa, a., Pullammanappallil, P., Ok, Y.S., Cao, X. (2016). A review of biochar as a low-cost adsorbent for aqueous heavy metal removal. Critical Reviews in Environmental Science and Technology 46, 406-433, doi:10.1080/10643389.2015.1096880.
- 3. Sun, W., Selim, H.M. (2020). Chapter Two – Fate and transport of molybdenum in soils: Kinetic modeling. Advances in Agronomy, academic Press, Vol. 164, 51-92, doi:10.1016/bs.agron.2020.06.002.
However, the recommended term “unlimited uptake” will be used in the revised manuscript to remove confusion, as below:
“…, while the Freundlich model (Eq. 2) was developed from unlimited uptake adsorption with an uneven-heat-distribution surface” (line 446-447).
#7 you should point out that the modern models have advantages of course. Langmuir and Freundlich parameters all depend on pH, temperature, or solution components, and are affected by the charging properties. Therefore, comparison between different systems make little sense. SCMs can give you some stability constant at infinite dilution and the use of a consistent model for the double layer will allow to have data base. Right now it is measuring and producing parameters that at most will confuse future researchers also in the sense that starters will believe these parameters have a physical or chemical meaning.
Thank you for this valuable comment. The relevant confusing fragment was removed from the Revised manuscript.
#8 what conductivity, heat, electrical?
According to this comment, the collocation “high conductivity” was removed from the relevant phrase in the Revised manuscript.
#9 are the sheets also hexagonal or only the arrangement of the atoms on the sheets? I think there is one hexagonal in excess.
Sorry for this inappropriate definition. According to this important comment, the relevant phrase was replaced by the correct one: “CNTs are composed of one or several graphene sheets of hexagonally arranged carbon atoms with sp2 hybridization rolled into a tube forming SWCNTs with a diameter of ~ 1 nm or MWCNTs with diameters of 10-100 nm and a distance of 0.342-0.375 nm between sheets, respectively (Figure 1) [84, 85].” Figure 1 was replaced by the novel more illustrative one in the Revised manuscript.
#10 I got the impression that thickness and diameter are confused, also it has to be clear if the dimensions refer to one sheet or the particles.
Thanks for the important comment. According to the literature, the diameters of SWCNTs and MWCNTS are presented in the relevant phrase. In order to avoid confusing between dimensions, Figure 1 was replaced by the new one with appropriate comments along with new Reference 90 in the Revised manuscript. The next phrase was also corrected as follows: “A thickness of SWCNTs is one carbon atom thick.”
11 critical in what sense? I got lost here.
The confusing definition “critical” was replaced by an appropriate “important” one.
#12 application to what?
The relevant phrase was replaced by an appropriate one “This possibility expands the application fields of CNTs including adsorption in the liquid phase”.
#13 there seems to be some repetition here.
According to this instructive comment, the relevant phrase was removed from the Revised manuscript.
#14 not clear to what extent HCl is an oxidant in this context.
Sorry for this mistake. According to this comment, “HCl” was removed from the relevant phrase of the Revised manuscript.
#15 this is again completely unclear. I guess assisting in dispersion in water means that the composites are better dispersable than the bare CNTs. It is not clear how the composites are composed. Magnetite bound to the CNTs? And the use of the word high somehow suggests they had high dispersability already.
According to this important comment, the relevant fragment was corrected as follows in the Revised manuscript:
“Another possibility of CNT modification is the preparation of the functional composites on their basis. In particular, CNTs tend to aggregate in water because of their hydrophobicity and high length-to-diameter ratio [113]. A simple combination of CNTs with magnetite affording modified magnetic CNTs assists their dispersibility in water as well as reusability and recovery after adsorption [59].”
The preparation way of the magnetite/CNTs composite is illustrated on the novel Figure 3 along with adding novel Reference 114 in the Revised manuscript.
#16 not sure what the relevance of the ionic liquids is. If these are then in contact with water, what is going to happen?
The authors are grateful to Referee for this important comment. Obviously, in IL@CNT composites, the potential release of ILs from the CNT surface should be evaluated. However, some reported results show that IL@CNT adsorbents demonstrate good recovery and recyclability [115], confirming no IL leaching during adsorption in aqueous media.
#17 how did the authors (#2 references missing) determine this? Was it an interpretation based on the macroscopic data or spectroscopy?
The authors are grateful for this valuable comment. In the cited work (Ref. 106), kinetic, isotherm and computational studies were carried out to determine the adsorption efficiency and mechanism of pharmaceuticals adsorption on IL@CNT composites. The novel Reference 117 related to in silico analysis of IL@CNT composites was added to the Revised manuscript.
#18 this is section as far as I can see. Would it be section 3.3.? why not just give the number.
Sorry, for this misprint. This is Section 4 in the Revised manuscript.
#19 I think a scheme could help here, showing the ranges. In general for a review some schemes/figures are in general inserted to make things clear. The text is not so illustrative for people consulting a review, where the beginner should also be able to get some picture of the topic.
According to this important suggestion, the novel Figure 5 illustrating different types of adsorption mechanisms for organic pollutant removal on CNTs was added to the Revised manuscript.
#20 measurements of electrokinetic data? There is often too much detail and often too little. The balance is not ok for a review.
Thank you for this instructive comment. According to it, the relevant phrase was supplemented as follows: “According to the specific surface charge determination by the measurement of streaming potential, SWCNT and MWCNT samples were negatively charged at pH values exceeding 3 (PZC of both samples was 3) [139] (Table 1, entry 3).”
I stopped here, also because I did not even mark all the issues I could have raised in this first part. As pointed out before, I think the authors need to make much more of an effort for such a kind of review.
Reviewer 4 Report
This is my second review of this article. The authors answered all my questions. The suggested corrections have been applied. The revised version is better the original one. I suggest to accept this article as it stands.
Author Response
Dear Reviewer,
Thank you for your valuable comments and suggestions.
Best regards
Leonid Kustov